# Humans and neural networks show similar patterns of transfer and interference during continual learning

Eleanor Holton [1,2] ✉, Lukas Braun [1,3], Jessica AF Thompson [1], Jan Grohn [1] & Christopher Summerfield [1]

In artificial neural networks, acquiring new knowledge often interferes with existing knowledge. Here, although it is commonly claimed that humans overcome this challenge, we find surprisingly similar patterns of interference across both types of learner. When learning sequential rule-based tasks (A–B–A), both learners benefit more from prior knowledge when the tasks are similar—but as a result, they also exhibit greater interference when retested on task A. In networks, this arises from reusing previously learned representations, which accelerates new learning at the cost of overwriting prior knowledge. In humans, we also observe individual differences: one group ('lumpers') shows more interference alongside better transfer, while another ('splitters') avoids interference at the cost of worse transfer. These behavioural profiles are mirrored in neural networks trained in the rich (lumper) or lazy (splitter) regimes, encouraging overlapping or distinct representations respectively. Together, these findings reveal shared computational trade-offs between transferring knowledge and avoiding interference in humans and artificial neural networks.

Continual learning is the ability to acquire multiple tasks in succession. Learning tasks in sequence is challenging because new task acquisition may cause existing knowledge to be overwritten, a phenomenon called catastrophic interference. Artificial neural networks (ANNs) trained with gradient descent are particularly prone to catastrophic interference[1–3]. Their difficulties with continual learning are often counterpointed with those of humans, who seem to be capable of accumulating and retaining knowledge across the lifespan. The computational basis of continual learning in humans is a topic of active investigation[4–7], and a consensus has emerged that task learning in humans and linear ANNs may rely on fundamentally different mechanisms[7]. Here, we describe work that challenges this assumption.

Recent work has shown that catastrophic interference can be counterintuitively worse when successive tasks are more similar to each other[8–10]. When faced with similar tasks, ANNs tend to adapt existing representations, rather than forming new ones. This allows for

better transfer (where learning one task accelerates learning of others), but existing representations are corrupted, provoking heightened interference. By contrast, when dissimilar tasks are encountered in succession, ANNs adopt a different strategy, which involves forming entirely new representations. This means that learning proceeds more slowly, but networks suffer less from interference[8,10,11]. In other words, higher catastrophic interference can be a cost that accompanies the benefits of transfer.

Although catastrophic forgetting in ANNs is often contrasted with successful continual learning in biological systems, there is good reason to believe they might rely on common principles of generalization and interference. In psychology, the term 'retroactive interference' refers to the phenomenon where new learning interferes with previous knowledge, analogous to catastrophic interference[12–20]. Cases of retroactive interference, for example, in sequential recall tasks, have also been proposed to depend on task similarity[20–25]. To take an intuitive

[1]Department of Experimental Psychology, University of Oxford, Oxford, UK. [2]Princeton Neuroscience Institute, Princeton University, Princeton, NJ, USA. [3]Allen Institute for Neural Dynamics, Allen Institute, Seattle, WA, USA. ✉e-mail: eleanor.holton@princeton.edu

example, language learners will find it easier to learn Italian after learning a similar Romance language (for instance, French compared with Korean), but may begin misapplying Italian words in French as a result. In dual-task paradigms, where participants must perform two tasks simultaneously, it is well established that cross-task interference is higher for tasks with shared structure[26,27], argued to be an intrinsic cost of sharing neural representations across tasks[28–30]. While these studies suggest there may be similar trade-offs occurring in humans too, as far as we are aware, no previous studies have systematically compared how patterns of catastrophic interference relate to transfer during continual learning in humans and ANNs.

Here, we directly compare humans and linear ANNs performing the same continual learning task, with a view to examining whether transfer and interference are governed by analogous computational principles. To investigate the fundamental computational principles that govern transfer and interference during continual learning, we adopted a minimalist modelling approach using layerwise linear neural networks. We trained both classes of learners on two sequential tasks (task A and task B) and then retested their knowledge of the first task (task A). First, we studied the effects of task similarity by varying the relationship between two task rules across three different groups of subjects (Same, Near and Far rule conditions). For both humans and ANNs, more similar tasks led to faster learning of task B (transfer), while more dissimilar tasks resulted in lower interference from task B when retested on task A. In ANNs, by analysing the hidden layer representations, we were able to show the precise computational principles that govern this effect. Consistent with previous work[10], we found that networks encode similar tasks in shared subspaces, which leads to interference; when they are sufficiently different however, networks encode tasks in separate, non-overlapping subspaces, which eliminates catastrophic interference.

Alongside these phenomena, we observed substantial individual differences consistent with a computational trade-off between the benefits of transfer and the avoidance of interference. Writing to a friend, the naturalist Charles Darwin described two groups of taxonomists: those who preferred to divide the botanical world into as many different species as captured their unique properties, and those who focused on commonalities, preferring to merge across the differences. Reflecting on these groups, he wrote, 'It is good to have hair-splitters & lumpers'[31]. In our study, we found a similar divergence in how people structured new information. Some participants reused the same rule across all stimuli ('lumpers'), which allowed them to learn faster in the second task, while incurring more interference when retested on the original task. These participants were also better at generalizing to unseen stimuli within a task, by applying their knowledge of the shared task rule. Meanwhile, other participants were able to avoid interference, but at the cost of worse transfer to new tasks and poor generalization within a task ('splitters'). Intriguingly, this group was better at recalling unique properties of the stimuli. These findings suggest that a tendency to focus on generalization of shared features versus individuation of unique features may reflect a meaningful axis of variation in human learning, although further work is needed to determine the stability of these tendencies across contexts.

We sought to understand these individual differences using our modelling framework. We drew upon recent work in machine learning revealing that networks can solve the same task using fundamentally different representations. In so-called rich networks, inputs are encoded in representations which reflect the low-dimensional structure of the task. By contrast, so-called lazy networks rely on high-dimensional, discriminable projections of the inputs, which form a basis for flexible downstream computations but often generalize poorly[32–37]. This transition from the 'rich' to 'lazy' regime can be driven by the scale of the initial weights[34,35,37]. We found that we could fully account for the individual differences in human learning by assuming a mixture of rich and lazy task solutions that favour generalization or individuation.

Together, these results point to key parallels between the trade-offs governing transfer and interference in humans and ANNs in continual learning settings. In both learning systems, learners who benefit most from generalizing shared structure also demonstrate the highest costs of interference. This balance is influenced both by external variables such as task similarity and by differences in the initial learning strategies.

## Results

Humans and twinned linear ANNs (collectively, 'learners') learned two successive tasks (task A followed by task B) and were then retested on task A (Fig. 1). Each task required learners to map six discrete inputs (plants) onto positions on a ring (locations on a planet) in two distinct contexts (the seasons of summer and winter; Fig. 1a). Within each of the two tasks, a consistent angle referred to as the 'task rule' defined the relationship between summer and winter locations for any plant. For example, within task A, each plant's winter location might always be 120° clockwise from their summer location (rule A = 120°; Fig. 1b). Learners were always probed on a plant's summer location first, and then its winter location after viewing feedback, allowing inference about the rule that linked the seasons. Notably, for one of the six stimuli, learners never received feedback on the winter location, allowing us to measure generalization of the rule within a task.

After completing task A, learners were trained on task B, where they learned to map a new set of six stimuli to their corresponding summer and winter locations on the ring. Participants received no indication that a new task had begun, aside from the fact that the task B stimuli were novel. Learners were divided into three groups, corresponding to three levels of similarity between the rule in task A (rule A) and the rule in task B (rule B; Fig. 1c). Depending on the condition, rule B was either identical to rule A (Same), shifted by 30° (Near), or shifted by 180° (Far). For example, if the relationship between the seasons in task A was 120°, it remained 120° for the new task B stimuli in the Same group, shifted to either 90° or 150° in the Near group, and changed to 300° in the Far group. The rules themselves were matched across conditions (Supplementary Fig. 1). After training on task B, learners were retested on the locations of task A stimuli—this time receiving feedback only about their summer responses. This allowed us to investigate their performance of rule A at retest without feedback, by analysing winter responses.

### Defining transfer and interference

In theory, learners could apply their knowledge of rule A to the novel stimuli in task B. This would manifest as using rule A to assign a winter location to a task B stimulus after receiving feedback about its summer location. Consequently, if learners apply prior knowledge, the more similar the task B rule, the better we expect initial performance on task B. Accordingly, we evaluate transfer in both humans and networks as the difference between the average winter accuracy for task A stimuli during their final presentation and the average winter accuracy for task B stimuli at their first presentation (Fig. 1d). Because we expect transfer to decrease with decreasing rule similarity, we expect the lowest transfer in the Far group.

Conversely, we predicted that successful transfer would come at the cost of greater interference from the new task. If Near group participants benefit more from transfer, this interference would manifest as greater use of rule B when retested on task A stimuli. Because no new rule learning occurs during retest, we could formally quantify interference as the probability of using rule B on return to task A. To measure this, we fit a mixture of von Mises distributions[38] centred on rule A and rule B to learners' rule responses (the offset between the winter response given, and the feedback for summer on the immediately previous trial). Higher interference corresponds to a higher probability weight of responding with rule B during retest of task A (Fig. 1d). As such, we measure interference from rule B in the Near

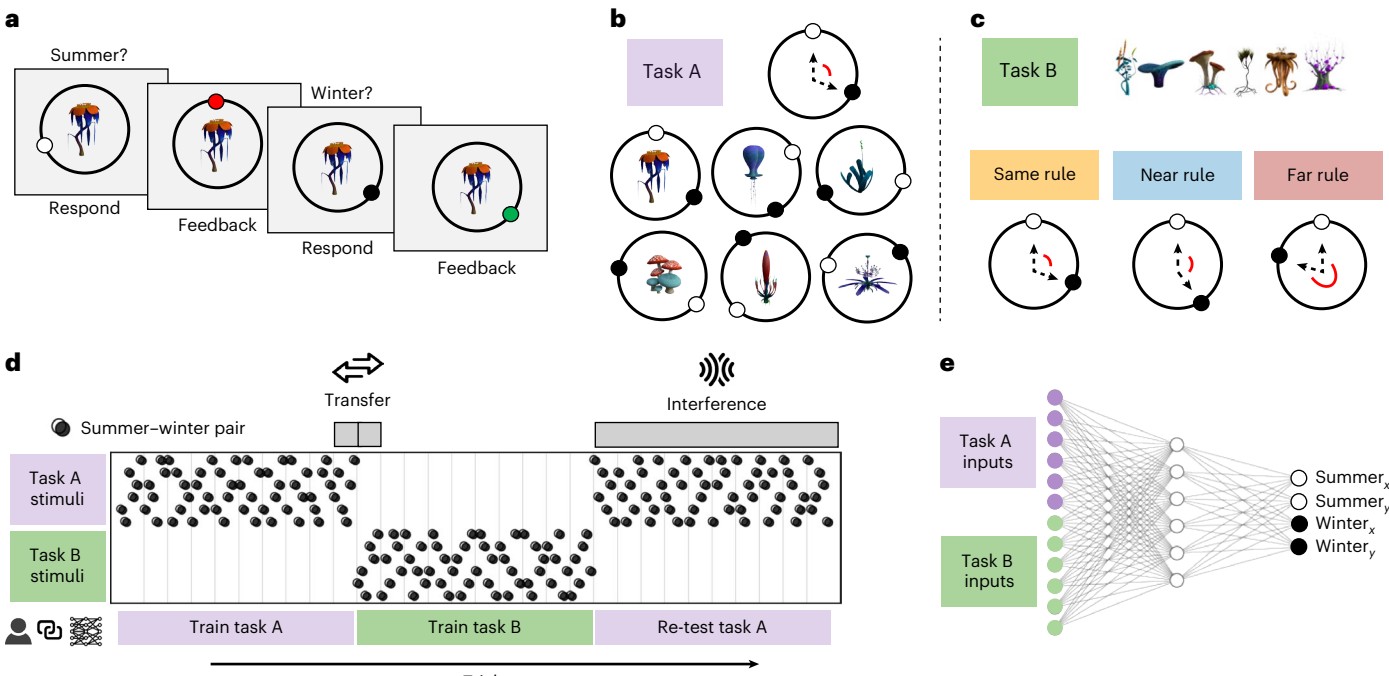

**Fig. 1 | Task design. a,** The task consisted of mapping plant stimuli to their locations on a circular dial, across two contexts (summer and winter). Participants always responded with the probed plant's summer location first, and then its winter location, receiving feedback after each response during training. **b,** Within task A, the relationship between each plant's location in summer (white circle) and winter (black circle) corresponded to a fixed angular rule (for example, 120° clockwise) that was randomized across participants. **c,** In task B, all participants learned to map a new set of stimuli to their respective summer and winter locations. However, the rule defining the relationship between seasons differed across groups of participants. In the Same condition, the seasons for task B were related by the same rule previously learned in task A; in the Near condition, the rule shifted by 30°; in the Far condition, the rule shifted by 180°.

**d,** All learners were trained on task A (120 trials), then task B (120 trials), and then retested on task A without feedback for winter. Transfer is defined as the change in winter accuracy from the final block (that is, one full stimuli cycle) of task A, to the first block of task B. If participants learn the rule, transfer should be better when the task B rule is more similar. Interference is defined as the probability of updating to the task B rule during retest of task A. For each participant, we trained a twinned neural network on the same stimuli sequence order and task rules. **e,** Networks consisted of feed-forward two-layer ANNs trained to associate sets of unique inputs (one-hot vectors; separate sets for each task) with the Cartesian coordinates of the winter and summer locations. Interference and transfer icons from OnlineWebFonts under a Creative Commons license CC BY 4.0.

and Far conditions where the rules change between tasks, but not in the Same condition where by definition the rules remain constant throughout. Parameter recoveries and model validation can be found in Supplementary Section 4.

## ANN studies

To enable direct comparisons between humans and models, each human participant was paired with a twinned neural network that followed the exact same trial schedule—receiving the same ordering of stimuli and feedback as that participant. In the ANN experiments, we used two-layer feed-forward linear networks (Fig. 1e), allowing us to study how the representations supporting rule learning emerge through gradient descent[32,39]. During the task A and task B learning phases, networks were trained to map one-hot vectors representing the discrete input stimuli onto Cartesian coordinates for the winter and summer locations on the ring. Crucially, network weights were not reset between tasks to allow us to study continual learning. Similarity between task A and task B was manipulated identically as for humans, by varying the rule relating the target coordinates in summer and winter. To mirror the continuous, fully informative feedback received by humans, we trained networks with trial-wise gradient updates, using a mean squared error (MSE) loss. During retest of task A, model weights were updated after summer trials, but not winter trials, analogous to participants receiving feedback only for their summer responses. We chose two-layer linear networks as the simplest architecture capable of learning transferable shared structure in this task. By contrast, single-layer regression models trained on unique (one-hot) inputs cannot share weights across stimuli

and are therefore incapable of transfer. Because the task is linearly solvable, linear networks are the most parsimonious choice for studying the representational dynamics supporting transfer and interference. However, we also confirm in supplementary analyses (Supplementary Section 2.1 and Supplementary Fig. 5) that the key behavioural effects also hold in ReLU networks, supporting the robustness of our findings beyond linear networks.

## ANNs show higher transfer at the cost of greater interference when learning similar tasks

All ANNs achieved near-zero training loss on both task A and task B by the end of their respective training phases across all conditions ($\mathcal{L} < 10^{-3}$; Fig. 2a–c).

However, we observed that transfer and interference differed across levels of task similarity (Same, Near and Far rule conditions). First, we focus on transfer. In Fig. 3e, we show that initial accuracy for winter responses in task B is unimpaired in the Same condition, declines moderately in the Near condition and drops the most in the Far condition (Same > Far: $t(202) = 605.79$, $P < 0.001$, $d = 85.25$, 95% confidence interval (CI) 0.94–0.94; Near > Far: $t(200) = 444.95$, $P < 0.001$, $d = 62.93$, 95% CI 0.80–0.81; Same > Near: $t(202) = 79.43$, $P < 0.001$, $d = 11.18$, 95% CI 0.13–0.14). This shows that, despite the novel inputs during task B (one-hot vectors that were not seen during task A), networks can capitalize on their prior training, showing greater transfer when the task rules are more similar.

Next, we examined interference, measured as the probability of incorrectly applying rule B upon return to task A. After training on both

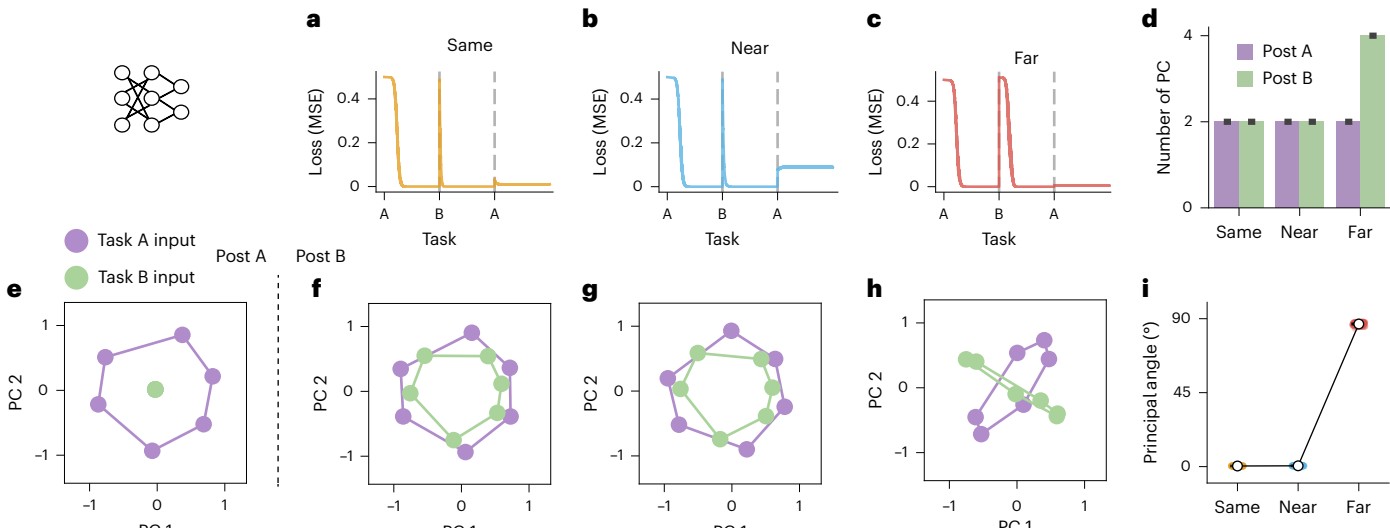

**Fig. 2 | Transfer and Interference in ANNs.** ANNs were trained on participant-matched trial sequences (one network per participant). **a**–**c**, Learning curves for network in the three conditions. Each network is trained sequentially on task A followed by task B (full supervision with mean-squared error loss), and then retested on task A. During retest, model weights are not updated after winter trials (analogous to participants receiving feedback only for summer but not winter stimuli). **a** shows networks trained in the same condition (tasks with identical rules), **b** shows networks trained in the near condition (tasks with similar rules) and **c** shows networks trained in the far condition (tasks with opposite rules). Dashed lines show task change points, showing the introduction of task B stimuli and the return to task A stimuli, respectively. **d**, The number of principal components needed to capture 99% variance of the activity at the network's hidden layer when exposed to all inputs. This is shown split by condition, both after training on only task A (purple) and after training on task B as well (green). **e**, Visualization of the two-dimensional representation of task stimuli at the network's hidden layer, after training on task A stimuli. PCA (with two components) was performed on the network's hidden layer activity when exposed to all inputs. **f**, Visualization of hidden layer stimuli representations after training on task B in the Same condition. **g**, The same as **f** after training the network to perform task B in the Near condition. **h**, The same as **f** after training the network to perform task B in the Far condition (see Supplementary Fig. 2 for additional visualizations of subspaces). **i**, Principal angles between task subspaces in the Same, Near and Far conditions. PCA (n = 2 components) was performed on ANN hidden layer activity for stimuli from task A versus task B, and the angle between subspaces computed. Larger angles indicate greater orthogonality between subspaces.

tasks, networks in the Near condition applied rule B to task A stimuli, showing truly catastrophic forgetting of the initial rule (Fig. 3j). By contrast, networks trained in the Far condition showed no interference: they were able to successfully return to using rule A.

Why would networks trained in different similarity conditions show such different patterns of interference? An initial clue comes from observing the learning curves as the networks were trained on task B. In the Same and Near groups, task B training resulted in rapid exponentially decreasing loss (Fig. 2a,b), as learning unfolds along an already established subspace. By contrast, loss curves in the Far group exhibited an initial plateau (Fig. 2c), appearing qualitatively similar to the curves observed when the networks were initially learning task A. One possibility is that this plateau occurs during the weight modifications that allow for learning to unfold in a new subspace.

To test this hypothesis, we examined the dimensionality of hidden representations over the course of learning, using principal component analysis (PCA). The dimensionality did not change as a result of learning task B in the Same and Near conditions (that is, the same number of components could explain 99% of variance in the hidden layer representation of all inputs). By contrast, the dimensionality doubled after learning task B in the Far condition, supporting the idea that a new subspace was formed (Fig. 2d). Indeed, visualization of the hidden representations in these networks implied that in the Same or Near condition the network reused the same subspace across the two tasks (Fig. 2e–g). However, networks trained on highly dissimilar rules in the Far condition learned the new task in a separate, orthogonal subspace (Fig. 2h).

Finally, we formally quantified the relationship between the subspaces each network used to represent the two tasks. We measured the principal angle between the two-dimensional subspaces encoding task A stimuli and task B stimuli after each network was fully trained[40–42]

(Fig. 2i). In networks learning the Same or Near tasks, this angle was 0°, indicating use of the same subspace across tasks. By contrast, the principal angle was 90° in networks trained in the Far condition, indicating use of an orthogonal subspace for the new task. This explains the slower learning of task B but preserved performance of task A at retest.

## Human studies
Next, we looked at whether humans showed similar patterns of transfer and interference as a function of task similarity. For the human experiments, we recruited separate groups of healthy online participants for each condition, across independent discovery and replication studies (discovery sample: Near, N = 50, Far, N = 50, Same, N = 52; replication sample: Near, N = 51, Far, N = 51, Same, N = 52).

## Humans also show higher transfer but more interference when learning similar tasks
Human participants were able to attain high accuracy in this study across all conditions (see Supplementary Fig. 8 for winter and summer accuracy over the course of learning; average winter accuracy in final block of task A; Same: mean (M) = 0.81, s.e.m. = 0.12, Near: M = 0.85, s.e.m. = 0.10, Far: M = 0.82, s.e.m. = 0.11; average winter accuracy in final block of task B; Same: M = 0.84, s.e.m. = 0.13, Near: M = 0.87, s.e.m. = 0.14, Far: M = 0.82, s.e.m. = 0.16). However, patterns of transfer and interference followed the same trends observed in neural networks, depending on task similarity.

First, we examined transfer among human participants. When introduced to the new stimuli in task B, participants in the Same condition were able to infer the correct winter locations by reapplying the previously learned rule, shown by their response errors clustering around zero (Fig. 3a). A similar pattern is observed in the Near condition, although response errors are systematically biased toward the

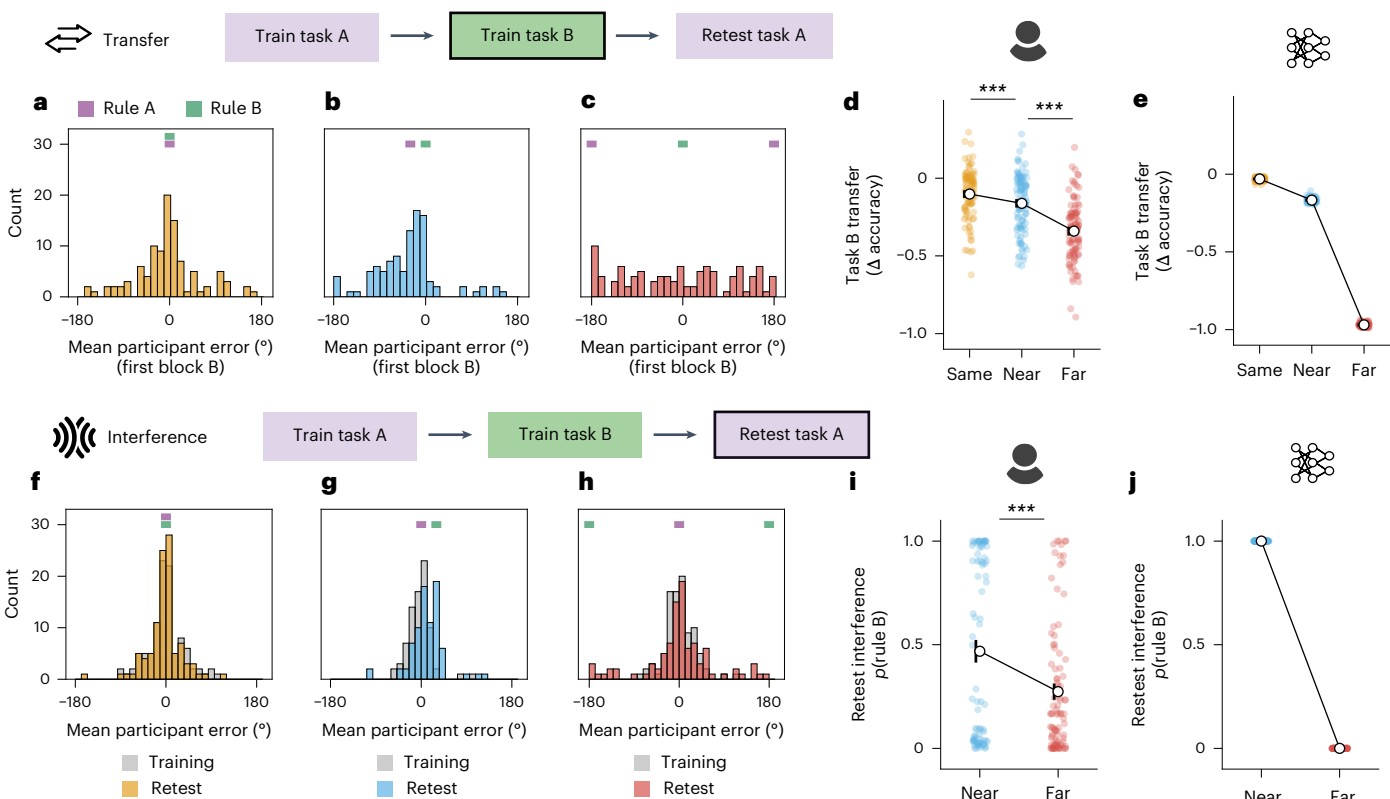

**Fig. 3 | Patterns of transfer and interference. a–e**, To study transfer, we examine the point when participants switch from task A to task B, encountering new stimuli. **a**, Histograms of the mean winter error across participants during the first block of task B in the Same condition (where the rule that links winter to summer stays the same). Purple and green notches mark expected error if applying the task A or task B rule, respectively. **b**, The same as **a** for the Near condition, where the rule shifts by 30°. While Near participants were randomly allocated to a task B rule +30° or −30° from the task A rule, here we flip the sign of errors for −30° participants for consistency, to visualize the biasing influence of the task A rule. **c**, The same as **a** for Far participants, who experience a new task B rule that is 180° from their previous task A rule. **d**, Transfer is defined as the cost of switching to the new task—the change in accuracy between the final winter responses for the six task A stimuli and the first winter responses for the six task B stimuli. Circles indicate mean, error bars show s.e.m. across the participant sample (Same $N$ = 103, Near $N$ = 101, Far $N$ = 101), and colours correspond to condition. $P$ values correspond to results of one-sided $t$-tests (Same > Far: $t$(202) = 9.48, $P$ < 0.001, $d$ = 1.33, 95% CI 0.19–0.29; Near > Far: $t$(200) = 6.63, $P$ < 0.001, $d$ = 0.93, 95% CI 0.13–0.23). ***$P$ < 0.001. **e**, Transfer in

ANNs trained on participant-matched schedules. **f–j**, To study interference, we examine how participants perform when returning to task A after completing task B. **f**, Histograms of retest error on task A (coloured) overlaid on task A training error (grey) in the Same condition. Little change suggests minimal interference. **g**, The same as **f** for the Near condition. A shift towards rule B indicates interference from the recently learned task. **h**, The same as **f** for the Far condition. In this case, very few participants respond using rule B during retest. **i**, Interference is quantified as the probability of using rule B when retested on task A, modelled using a von Mises mixture where 1 indicates full use of rule B and 0 indicates use of rule A. Circles indicate mean, error bars show s.e.m. across participants (Near $N$ = 80, Far $N$ = 94; participants who failed to learn task B were excluded), and colours correspond to condition. $P$ values correspond to results of one-sided $t$-test (Near > Far: $t$(172) = 3.44, $P$ < 0.001, $d$ = 0.53, 95% CI 0.08–0.31). ***$P$ < 0.001. **j**, Interference in ANNs trained on participant-matched schedules. Colours correspond to training condition. Note that human data are aggregated across the discovery and replication samples (for data plotted by sample, see Supplementary Fig. 6). Interference and transfer icons from OnlineWebFonts under a Creative Commons license CC BY 4.0.

previously learned rule (Fig. 3b). By contrast, errors in the Far condition are widely distributed, indicating that participants were unable to reuse their previous rule, instead learning the new task from scratch (Fig. 3c). Similar to ANNs, human participants in the Same and Near groups therefore showed greater transfer to task B than in the Far condition (Fig. 3d; one-way analysis of variance for effect of condition on transfer; discovery sample: $F$(2, 148) = 18.69, $P$ < 0.001, $\eta^2$ = 0.20; replication sample: $F$(2, 151) = 29.34, $P$ < 0.001, $\eta^2$ = 0.28. Δ accuracy in the Far condition was significantly lower than the Near and Same condition; Far < Same one-sided $t$-test: $t$(99) = 6.12, $P$ < 0.001, $d$ = 1.23, 95% CI 0.15–0.29 (discovery sample); $t$(101) = 7.23, $P$ < 0.001, $d$ = 1.43, 95% CI 0.19–0.33 (replication sample); Far < Near one-sided $t$-test: $t$(98) = 3.85, $P$ < 0.001, $d$ = 0.78, 95% CI 0.07–0.23 (discovery sample); $t$(100) = 5.52, $P$ < 0.001, $d$ = 1.10, 95% CI 0.13–0.28 (replication sample). This shows that participants were able to successfully infer the task rules, and benefit from transfer to task B when rules remained similar. Importantly, the pattern of switch costs that we observe is better

explained by participants transferring their previous rule to the new task B stimuli, rather than alternative behavioural strategies such as responding randomly or repeating their summer location feedback (Supplementary Fig. 9).

Next, we measured interference from task B when participants were retested on task A. Our theory concerns interference occurring as a result of new learning, so participants who failed to learn task B were excluded from interference analyses (14% participants excluded). Participants in the Same condition showed response errors tightly clustered around zero, reflecting consistent use of the original rule, which remained unchanged throughout task B (Fig. 3f). However, many participants in the Near condition shifted towards applying rule B at retest (Fig. 3g), while participants in the Far condition largely maintained rule A rather than shifting to rule B (Fig. 3h). Quantifying this formally, we found that Near group participants showed higher interference after learning task B compared with those in the Far condition—in other words, they were more likely to misapply rule B during retest (Fig. 3i;

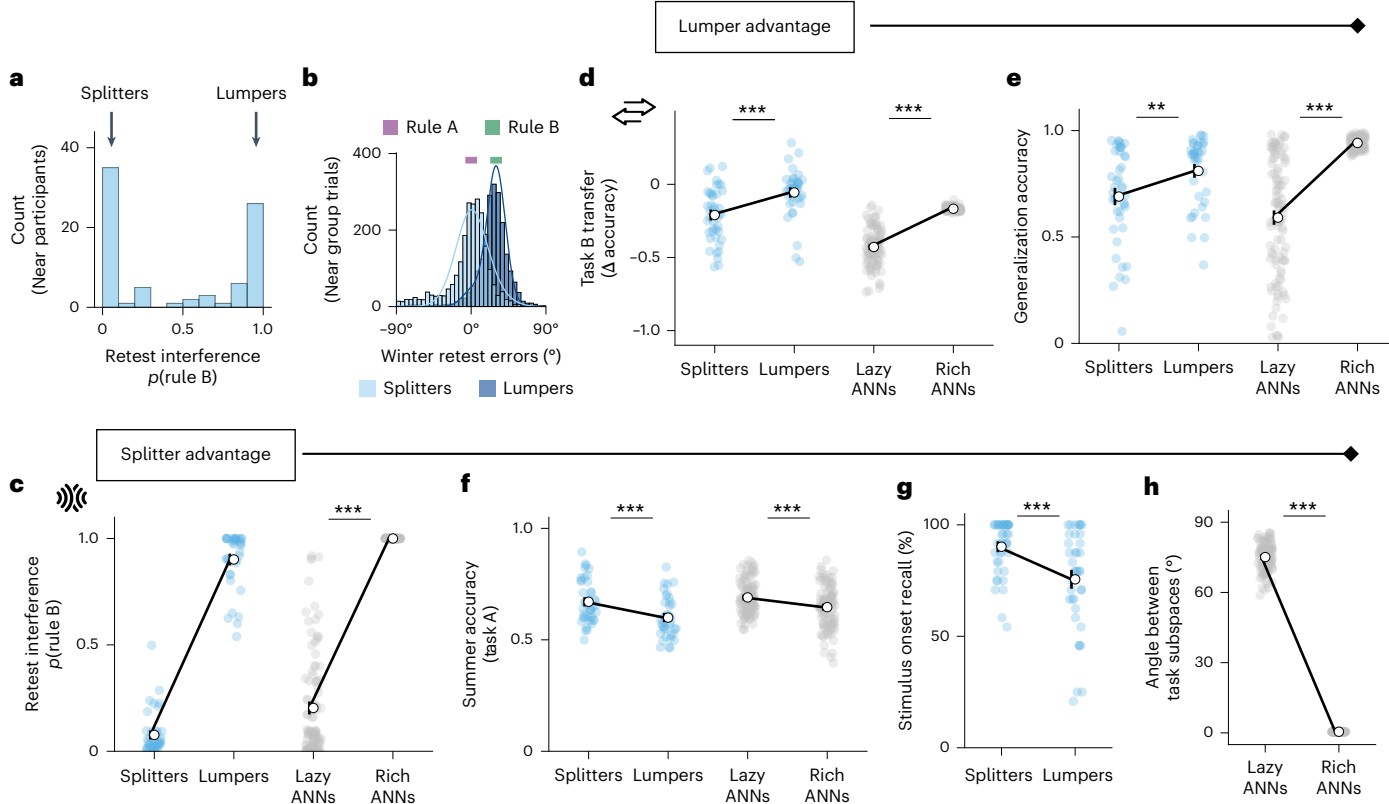

**Fig. 4 | Individual differences in transfer and interference. a**, In the Near condition, interference at retest is bimodally distributed. Participants in the Near group were classified into splitters (those with low interference from task B) and lumpers (those with high interference from task B). **b**, A histogram of all winter retest errors for splitters (light blue) and lumpers (dark blue). Lines show the posterior model fits, computed using the average concentration ($\kappa$) and mixture weight ($\pi$) parameters across participants in each group. **c**–**h**, On the left (in blue), we plot behavioural data from the splitters and lumpers. On the right (in grey), we plot data from ANNs trained under a lazy learning regime (forming unstructured, high-dimensional task solutions), versus trained under a rich learning regime (forming structured, low-dimensional task solutions). In each plot, circles show mean metrics in each group, dots show individual data points and error bars show s.e.m. (splitters: $N = 42$, lumpers: $N = 38$). $P$ values correspond to results of two-sided $t$-tests. **c**, Interference among splitters and lumpers is plotted for illustrative purposes only (because this metric determines the classification), for comparison with interference in lazy and rich ANNs (ANNs: $t(200) = 32.8$, $P < 0.001$, $d = 4.64$, 95% CI 0.75–0.84). **d**, Transfer performance in the groups, as defined throughout as the change in winter accuracy between the final exposure to task A stimuli and the first exposure to task B stimuli (humans: $t(78) = 3.95$, $P < 0.001$, $d = 0.89$, 95% CI 0.08–0.23; ANNs: $t(200) = 20.97$, $P < 0.001$, $d = 2.97$, 95% CI 0.24–0.29). **e**, Generalization accuracy is the average winter accuracy for the test stimulus in task A, for which feedback about winter is withheld throughout. Because participants only receive feedback about its

summer location, they must infer the correct winter location by generalizing their knowledge of the task A rule (humans: $t(78) = 2.74$, $P = 0.008$, $d = 0.62$, 95% CI 0.03–0.21; ANNs: $t(200) = 12.72$, $P < 0.001$, $d = 1.80$, 95% CI 0.30–0.40). **f**, Average accuracy for summer responses, which must be remembered for each stimulus separately (in contrast to winter responses, which can be inferred by applying the rule to the summer feedback). This requires participants to discriminate the unique stimuli. ANN performance is shown for the first 120 trials of task A training, to match the length of human training. For full accuracy trajectories over time, including later stages of training, see Supplementary Fig. 15 (humans: $t(78) = 3.40$, $P < 0.001$, $d = 0.76$, 95% CI 0.03–0.11; ANNs: $t(200) = 3.60$, $d = 0.51$, $P < 0.001$, 95% CI 0.02–0.07). **g**, At the end of the study, participants were asked to recall when they saw each stimulus for the first time (at the beginning of the study, or halfway through). In other words, this reflects the ability to explicitly report the onset of unique task stimuli (humans only: $t(78) = 3.69$, $P < 0.001$, $d = 0.81$, 95% CI 6.5–22.8). **h**, Representational similarity between task A and task B stimuli in ANNs, quantified as the principal angle between their respective hidden layer subspaces after task B training. Rich networks collapse the representations onto the same subspace, while lazy networks retain greater distinction between representations (ANNs only: $t(200) = 125.50$, $P < 0.001$, $d = 17.75$, 95% CI 73.4–75.8). **$P < 0.01$, ***$P < 0.001$ (**c**–**h**). Credit: interference and transfer icons from OnlineWebFonts under a Creative Commons license CC BY 4.0.

$p$(rule B) in Near > Far, one-sided $t$-test; $t(86) = 2.56$, $P = 0.006$, $d = 0.55$, 95% CI 0.04–0.37 (discovery sample); $t(84) = 2.27$, $P = 0.013$, $d = 0.50$, 95% CI 0.02–0.35 (replication sample); see Supplementary Section 4 for further detail on model validation and parameter recoveries, and Supplementary Fig. 7 for effects on retest accuracy).

Taken together, these results support the idea that humans and neural networks show similar patterns of transfer and interference, with the same systematic dependency on task similarity. In neural networks, we can see that learning tasks of intermediate similarity promotes shared representations, manifesting in higher transfer across tasks at the cost of greater interference. By contrast, learning highly dissimilar tasks leads to less transfer but lower interference between

tasks. We find these patterns of trade-offs between the benefits of transfer and avoidance of interference are preserved across the two learning systems.

### Individual differences in transfer and interference

Although participants learning similar tasks generally showed more interference than those learning dissimilar tasks, this pattern was not universal: many individuals in the Near group showed little to no interference. In fact, interference weights in this group were bimodally distributed (Fig. 4a), suggesting the presence of two distinct learning strategies. Some individuals appeared to overwrite rule A with rule B, while others returned to rule A, effectively avoiding interference

(Fig. 4b). In the context of our theory, this naturally leads to the question of whether these individual differences are also characterized by a trade-off between benefitting from transfer and avoiding interference. We predicted this may reflect differences in how participants approached the task structure, with some participants merging across tasks based on shared structure (lumpers), and others focusing on the differences between stimuli (splitters). To study this phenomenon further, we used a model-based approach to classify lumpers and splitters. Participants whose responses during retest of task A were best fit by rule A were categorized as splitters, while those whose responses were best fit by rule B were categorized as lumpers. In our cohort, 47.5% of Near-group participants were lumpers.

If the increased interference observed in lumpers (Fig. 4c, left) arose from a focus on shared structure, we would expect lumpers to demonstrate better transfer compared with splitters. Indeed, we found that lumpers were better at switching to task B, benefitting from similarities between the two tasks (Fig. 4d, left; transfer: splitters: $M = -0.21$, s.e.m. = 0.03, lumpers: $M = -0.06$, s.e.m. = 0.03; two-sided $t$-test: $t(78) = 3.95$, $P < 0.001$, $d = 0.89$, 95% CI 0.08–0.23). In addition, if lumpers were good at capitalizing on shared task structure, we would expect lumpers to successfully generalize the rule to untrained stimuli within task A. To test this, we leveraged a feature of our experimental design: for one 'test' stimulus in task A, feedback was not provided for winter responses, allowing us to measure generalization. We found that lumpers indeed exhibited higher accuracy for the test stimulus, demonstrating greater within-task generalization (Fig. 4e, left; splitters: $M = 0.69$, s.e.m. = 0.04, lumpers: $M = 0.81$, s.e.m. = 0.03; two-sided $t$-test: $t(78) = 2.74$, $P = 0.008$, $d = 0.62$, 95% CI 0.03–0.21). In other words, individuals who experienced more interference were better at extending their knowledge to new situations—both when learning task B as well as when inferring untrained responses within a task. This is consistent with our theory that lumpers are relying more on shared representations during learning.

Could lumpers be performing better on these metrics simply as a result of higher task engagement? If this were the case, we would expect them to show generally higher accuracy across the board. To address this possibility, we assessed participants' accuracy for the 'summer' response during task A—the initial phase of the experiment. Due to the sequential nature of each trial (participants are always probed on summer before winter), summer accuracy reflects the ability to recall the unique, memorized location of each stimulus, whereas winter accuracy can be inferred by applying the rule to the summer location. We found that lumpers—while achieving higher accuracy in transfer and generalization—were significantly worse than splitters at remembering the unique summer positions (Fig. 4f, left; splitters: $M = 0.671$, s.e.m. = 0.015; lumpers: $M = 0.600$, s.e.m. = 0.015; two-sided $t$-test: $t(78) = 3.40$, $P = 0.001$, $d = 0.76$, 95% CI 0.03–0.11). Notably, splitters retained their summer accuracy advantage over lumpers throughout the entire experiment, including during task B and the A retest phases (Supplementary Fig. 15). This indicates that splitters were not merely less engaged in the task, because they performed better than lumpers at remembering summer locations. Instead, one possibility is that they relied on a memorization-based strategy that prioritized memorizing the correct locations, rather than learning to apply the generalizable rule. Consistent with this interpretation, response precision at retest (quantified by the concentration parameter $\kappa$) was significantly lower in splitters than lumpers (Supplementary Fig. 14d; two-sided $t$-test: $t(78) = 2.83$, $P = 0.006$, $d = 0.62$, 95% CI 4.6–29.4; Mann–Whitney: $U = 1,098.0$, $P = 0.004$), but nonetheless splitters achieved higher accuracy at retest (Supplementary Fig. 14e; two-sided $t$-test: $t(78) = -1.80$, $P = 0.076$, $d = 0.40$, 95% CI −0.01 to 0.97; Mann–Whitney: $U = 512.0$, $P = 0.006$). This is consistent with lumpers performing precise but systematically biased responses (applying the task B rule), while splitters relied more on memorized mappings from task A, leading to more variable but less biased responses.

Next, we asked whether these different strategies related to participants' temporal memory of the stimuli. At the end of the study, we presented participants with each stimulus independently and asked them to report whether they had originally seen the stimulus in the first or second half of the study (task A or task B). We found that lumpers were worse at explicitly reporting this temporal separation, consistent with our theory that lumpers merged representations of stimuli across tasks (Fig. 4g; splitters: $M = 90.2\%$, s.e.m. = 1.9, lumpers: $M = 75.5\%$, s.e.m. = 3.6, two-sided $t$-test: $t(78) = 3.69$, $P < 0.001$, $d = 0.81$, 95% CI 6.5–22.8). Given that recall accuracy was clustered near the upper boundary, we verified the group difference using a mixed-effects logistic regression. This robustness check confirmed that lumpers were worse at categorizing the stimuli by their temporal separation ($\beta = 1.35$, s.e.m. = 0.38, $z = 3.57$, $P < 0.001$). Finally, we verified that all results reported above remain unchanged after excluding participants best fit by a model capturing random responding, supporting our claim that these behavioural patterns capture true differences in strategy rather than noise (see control analyses in Supplementary Section 5.2).

Taken together, these results suggest that participants differed in their tendency to learn the task by focusing on generalization of shared structure or memorization of the unique features of stimuli. Crucially, these individual differences in strategy were underpinned by the same fundamental trade-offs: individuals who benefitted more from generalizing shared structure also incurred greater interference.

## Strategy differences can be captured by ANNs trained in rich or lazy regimes

Next, we investigated whether the individual differences in transfer and interference observed in humans could be captured within our connectionist framework. Neural networks can solve the same task with minimal training error while relying on fundamentally different internal representations of stimuli. In the so-called rich regime, networks encode inputs using representations that reflect the task's underlying structure and dimensionality. By contrast, networks in the lazy regime leverage the initial, random projections of inputs, forming high-dimensional representations that facilitate individuation but are independent of the task structure[34–37,43,44]. Because individual differences within the Near group were not driven by differences in task structure or engagement, we hypothesized that lumpers and splitters might be captured by networks trained in the rich and lazy learning regimes, respectively. For example, prior work has shown that rich networks generalize better than lazy networks[43,44]. A well-established method for guiding networks to form rich or lazy representations is by varying the scale of initial weights[37,43,44]. While our previous simulations used small initial weights, promoting the formation of rich structured representations, we reasoned that training a mixture of networks in both the rich and lazy regimes could capture the spectrum of individual differences in transfer, generalization and interference observed in human learners.

As hypothesized, we found that rich and lazy networks mirrored many of the behavioural differences observed in the lumpers and splitters, respectively. First, rich networks exhibited greater interference from learning task B (Fig. 4c, right; lazy: $M = 0.20$, s.e.m. = 0.02, rich: $M = 1.0$, s.e.m. = <0.01, two-sided $t$-test: $t(200) = 32.8$, $P < 0.001$, $d = 4.64$, 95% CI 0.75–0.84). Second, they showed superior transfer performance when task B was introduced (Fig. 4d, right; lazy: $M = -0.43$, s.e.m. = 0.012, rich: $M = -0.17$, s.e.m. = 0.001, two-sided $t$-test: $t(200) = 20.97$, $P < 0.001$, $d = 2.97$, 95% CI 0.24–0.29). Third, rich networks demonstrated better generalization to the held-out test stimulus (Fig. 4e, right; lazy: $M = 0.59$, s.e.m. = 0.03, rich: $M = 0.94$, s.e.m. = 0.003, two-sided $t$-test: $t(200) = 12.72$, $P < 0.001$, $d = 1.80$, 95% CI 0.30–0.40). Fourth, rich networks also showed significantly lower summer accuracy than lazy networks during task A (Fig. 4f, right; lazy: $M = 0.690$, s.e.m. = 0.008; rich: $M = 0.646$, s.e.m. = 0.009; two-sided $t$-test: $t(200) = 3.60$, $d = 0.51$, $P < 0.001$, 95% CI 0.02–0.07). This effect is clearest in the early stages

of learning; specifically, we computed ANN performance over the first 120 trials of task A training, to match the training length experienced by human participants (for a full breakdown of accuracy trajectories over time, including later stages of learning, see Supplementary Fig. 15).

Finally, we turn to the result that splitter participants were substantially better at remembering the onset of each stimulus—that is, whether a stimulus was first encountered in training on task A or task B. This ability to explicitly distinguish when a stimulus was introduced depends on maintaining separable representations of the two tasks. To test whether representational compression differed between rich and lazy ANNs, we measured the degree of overlap between task A and task B stimuli hidden layer representations after task B training, using the principal angle. In rich networks, the task subspaces were nearly aligned (mean principal angle 0.49°, s.e.m. = 0.006), whereas the first two principal components in the lazy networks were near orthogonal for the two tasks (mean principal angle 75.10°, s.e.m. = 0.59), with a highly significant difference between groups (Fig. 4h, two-sided $t$-test: $t(200) = 125.50$, $P < 0.001$, $d = 17.75$, 95% CI 73.4–75.8). These findings provide a representational-level explanation for the temporal memory differences observed in humans: possibly, participants who compressed information across tasks (that is, lumpers) suffered when required to recall temporally specific information about task stimuli, just as rich networks collapsed their task representations.

Overall, these analyses support our hypothesis that patterns of transfer and interference systematically depend on whether learners use shared representations. In particular, in the rich regime, representations of stimuli in the neural networks are structured along a shared low-dimensional manifold, leading to high interference, transfer and generalization. By contrast, the lazy regime utilizes disjoint hidden representations, resulting in lower interference, transfer and generalization. Notably, both learning regimes support successful acquisition of both tasks, suggesting that differences in representation may go unnoticed when assessing only final performance without the inclusion of held-out test stimuli or periods of retest after new learning. These patterns characterizing the different task solutions within the Near group reflect the same fundamental trade-off: sharing representations across tasks brings greater transfer at the cost of higher interference.

While our focus has been on comparing human strategies with rich and lazy learning regimes, several other representational solutions are known to mitigate interference in continual learning. To explore this broader landscape, we conducted simulations using networks trained with three commonly studied interference-mitigation strategies: elastic weight consolidation, replay and modular architectures (Supplementary Section 6). Because splitters avoided interference but failed to generalize within task, their behaviour most closely resembled lazy networks, suggesting a high-dimensional task solution akin to memorization over other forms of interference mitigation such as replay (Supplementary Figs. 17 and 18).

## Discussion

This Article makes three primary contributions. First, we show that humans learning structured tasks in short succession face a trade-off between transferring knowledge across tasks and avoiding interference on previous tasks. This trade-off is shaped both by global properties, such as the similarity between successive tasks, and by individual differences in the solutions people learn. Second, we show this behavioural pattern closely parallels predictions from linear neural networks, which exhibit a trade-off between sharing and separating task representations during continual learning. Following our findings in humans, we find this balance in linear neural networks is influenced both by intertask similarity and the properties of the initial task solution. Finally, we demonstrate that individual differences in the strategies people learn give rise to consistent and complementary performance profiles, favouring either generalization across shared task structures or discrimination of unique task properties.

Across both humans and ANNs, learning more similar tasks led to greater transfer but at the cost of higher interference. Consistent with previous literature[10], we observed that ANNs solved similar tasks by repurposing existing representations, facilitating faster learning but corrupting prior representations. By contrast, learning orthogonal tasks encouraged the formation of separate representations, preventing interference. This aligns with a growing literature in machine learning showing that catastrophic forgetting can be mitigated by encouraging networks to learn in orthogonal subspaces in more complex settings[45–49]. Machine learning methods differ in how they impose subspace separation: for example, Duncker et al.[47] proposed a continual learning algorithm that encourages networks to organize dissimilar task dynamics into orthogonal representational subspaces, while others impose hard task boundaries using precomputed orthogonal projection matrices (for example, ref. 46) or gradient-penalty methods[49]. While our results are grounded in simple linear networks, an important direction for future work is to investigate whether the principles we observe also hold in deeper, nonlinear networks used in modern artificial intelligence systems.

In our setting, linear networks exposed to orthogonal rules in the Far condition naturally developed orthogonal subspaces over the course of learning, without any architectural constraints or additional loss terms. This demonstrates that orthogonal representations can emerge spontaneously when the meta-statistics of the task structure support them. While we observe a similar behavioural pattern in humans, we emphasize that the emergence of orthogonal representations in the human brain remains a theoretical prediction, pending empirical confirmation via neural recordings.

While neural networks provide a useful computational framework for studying learning, they are not biological brains. Our approach treats them as tools to study general principles of continual learning rather than assuming them to be analogues of human cognition. However, the observed trade-offs between sharing and separating representations during learning align with broader theories of task switching in humans[28,29,50,51], and orthogonal neural representations are known to mitigate task interference in biological systems[42,44,52–62]. A recent relevant study in mice[63] demonstrated that individual differences in interference during continual learning were correlated with the degree of orthogonalization in neural representations, underscoring the biological relevance of subspace separation in maintaining memory stability. Such partitioning could, in principle, be implemented biologically via mechanisms such as synaptic consolidation[64] or neural pruning[65], which aim to protect task-relevant parameters by selectively reducing plasticity. While our study remains agnostic about the precise biological substrates, we view these mechanisms as complementary to our representational framework, offering potential routes by which the brain could achieve separation of task representations.

Despite finding a general trend that human learners exposed to two similar tasks experienced more interference, this was not the case for everyone. Closer examination of learners in the Near group who avoided interference revealed two distinct approaches to continual learning. Lumpers leveraged shared structure across tasks, enabling better generalization and transfer but suffering more interference. Splitters, by contrast, exhibited reduced interference, but this came at the cost of poorer generalization within a task and weaker transfer to the second task. This distinction was mirrored in ANNs trained under rich versus lazy regimes, with low-dimensional, task-compressed (rich) solutions resembling lumpers, and high-dimensional, task-agnostic (lazy) solutions resembling splitters. Our simulations show that interference can be mitigated even when learning similar tasks, if the network adopts high-dimensional solutions. This reveals a fundamental trade-off: while compressed, low-dimensional representations support efficient generalization and downstream transfer[36,66–69], they are also more vulnerable to interference when reused across tasks.

Although these patterns suggest similar computational constraints across systems, there are various limitations to our interpretation of these individual differences. First, while rich and lazy networks serve as a computational proxy for these different strategies, their biological relevance remains uncertain. There is no direct biological counterpart for the changes in initial weight scale that lead to these varying representations, and further work would be required to establish that the final representations supporting performance bear similarities across the two groups of humans and ANNs[44].

A second limitation is that, without neural data, we cannot determine how tasks are represented in the brain across the two groups, even if their behavioural patterns suggest underlying differences in strategy. Previous neural recordings have suggested that the brain may simultaneously use both rich and lazy representational schemes in different regions: early sensory areas exhibit high-dimensional, task-agnostic codes resembling lazy learning, while higher-order areas such as posterior parietal cortex contain lower dimensional task-specific representations[44,70]. These findings underscore the challenge of associating human behaviour with a single form of representation, but suggest that comparisons at the level of specific brain regions may also be informative. Future neuroimaging studies could compare the neural representations of lumpers and splitters in our sequential learning paradigm to investigate differences in both dimensionality and localization of task representations across the brain.

A third limitation is that it remains unclear how far the distinction between splitters and lumpers generalizes beyond our task setting. Our classification was based on behavioural patterns of interference, and although this allowed us to predict a rich range of independent behavioural metrics, future work could strengthen the basis of this distinction by introducing a novel task to assess whether the classification generalizes beyond the current setting. Similarly, alternative model parameterizations could potentially provide a closer fit to the splitter and lumper response distributions, offering complementary perspectives on the behavioural distinctions observed in our task. In addition, it would be interesting to explore parametric analyses of individual differences using a continuous measure of network richness. Recent theoretical work[37] provides a principled framework for understanding the variation between rich and lazy learning along a continuous axis. Their analytical characterization of the transition between lazy and rich learning regimes in deep linear networks (driven by the relative scale of initialization) offers a promising basis for predicting behavioural patterns along this continuum, potentially yielding a more nuanced account of human variability in learning strategies beyond the bimodal approach taken in the current study.

While we have argued that humans and ANNs face similar computational trade-offs during continual learning, humans are likely to deploy additional mechanisms to balance these challenges. In particular, the medial temporal lobe supports the rapid acquisition of new tasks before integration into cortical knowledge systems over longer timescales[71–73], a process that our linear networks cannot replicate. Biological systems may also mitigate interference through mechanisms such as replay-based consolidation[71,74,75] and synaptic consolidation[76–78], both of which have inspired continual learning approaches in artificial learning systems[64,65,79–82].

While we explored ANN implementations of these mechanisms—including replay, synaptic consolidation (via elastic weight consolidation) and modular architectures—we found that none recapitulated the behavioural profile of human splitters, which was most closely matched by lazy networks with high-dimensional, task-specific representations (Supplementary Section 6 and Supplementary Figs. 17 and 18). It is possible that such mechanisms operate on timescales beyond those captured by our task. Replay has been shown to support ongoing generalization[83–85], with interesting implications for how knowledge continues to be structured after learning. This parallels recent studies in rodents showing that generalization abilities can continue to

develop even after task performance has plateaued[86]. Synaptic consolidation mechanisms observed in biological systems[76–78] may also require extended consolidation periods to prevent interference. These temporal considerations may help explain why participants in the Near condition who avoid interference (splitters) succeed through strategies that come at the cost of transfer and generalization. Specifically, while participants in the Far condition naturally separate task representations due to their dissimilarity, such separation may be more difficult to establish in the Near condition, where the similarity of rules blurs task boundaries. The strategy observed in splitters may reflect a viable solution for mitigating interference in settings where neither offline consolidation over longer timescales nor straightforward inference of task boundaries is available to learners.

In our study, neither humans nor ANNs were given explicit task labels or cues about when the task changed. This design mimics more naturalistic learning environments, where task boundaries are inferred from environmental structure rather than signalled externally[87]. An important future direction will be to characterize how and when representational separation occurs for different levels and dimensions of task similarity. One promising approach is to draw on meta-learning methods that infer task boundaries from shifts in data structure, such as Bayesian frameworks that jointly segment data and learn task models[88].

Humans are experts at identifying boundaries in the world using attributes that go beyond the notion of rule similarity we use here[6,87,89]. For example, previous work has shown that the temporal proximity of learning episodes is critical for knowledge partitioning in humans, such that events closer in time are more likely to be attributed to the same source[44,90,91]. This general ability to partition knowledge over time has been linked to various aspects of mental health, including anxiety[92] and symptoms in post-traumatic stress disorder[93]. Future work could extend our understanding of these processes in mental health conditions by considering how these behaviours might balance transfer and interference in different environments. This perspective highlights another component of the challenge: while forgetting past knowledge will be maladaptive, there is of course the need for flexibility in revising knowledge structures when the environment has truly changed[6].

Finally, over extended training or in settings involving many tasks, humans appear capable of decomposing tasks into reusable elements[94,95]. Recent work by Driscoll et al.[96] provides a potential computational mechanism for this, showing that recurrent neural networks trained on multiple tasks develop dynamic motifs—such as decision boundaries or attractor states—that are reused across tasks. Extending our paradigm to longer sequences of tasks in humans could reveal whether similar reusable structures emerge to support the decomposition of tasks into shared and distinct components, facilitating transfer without interference.

In conclusion, our results support the theory that patterns of transfer and interference in humans and ANNs reflect a computational trade-off between sharing and separating representations during learning. Across both systems, this balance depends on global trends such as task similarity, as well as properties of the learner's initial task solution. Understanding these constraints may provide new avenues for characterizing individual differences in continual learning and how they relate to stable cognition.

## Methods
### Participants
Participants were recruited on Prolific.co (discovery study: $N = 202$ recruited; $N = 151$ after exclusion; replication study: $N = 215$ recruited; $N = 154$ after exclusion). Prolific inclusion criteria included being between 18 and 40 years old, being an English speaker, being located in the US or the UK, having a minimal approval rate of 90% and having a minimum of five previous submissions on Prolific. Among the 305 participants total remaining after exclusion, the mean age was

31.37 years (s.d. 8.27) and 157 were female. After eligible participants were recruited, we additionally excluded participants who reported using tools (for example, pen and paper) in the debrief, and participants whose accuracy did not significantly exceed chance performance in the final two blocks of learning task A. We preregistered our exclusion criteria, our experimental design, and our main hypotheses about transfer and interference. The discovery study was preregistered at https://osf.io/ps4m9 (preregistered under 'Experiment 1'). The replication study was preregistered at https://osf.io/92dpm. Ethical approval was obtained from the Oxford Central University Research Ethics Committee (Ref: R50750/RE009). All participants gave informed consent before the experiment. The experiment took on average 35 min to complete, and participants were compensated £8 per hour with a performance-dependent bonus of up to £3.

## Task design

The task required participants to map unique stimuli (images of fictional plants) to circular outputs (locations on a planet) in two distinct contexts: summer and winter. On each trial, participants adjusted a dial to indicate the correct position of each stimulus on a circle, always indicating its summer location followed by its winter location (receiving feedback after each response). The experiment consisted of two sequential tasks (task A and task B), each defined by a unique set of six stimuli. Participants were not informed that a second task would follow and received no explicit indication when the task changed (see Supplementary Section 1 for task instructions). Within each task, there was a fixed relational rule that defined the angular offset between the stimuli positions in summer and winter (rule A in task A; rule B in task B). The study was divided into three phases: training on task A (phase 1), training on task B (phase 2) and retesting of task A without feedback for the winter season feature (phase 3).

In total, participants completed 300 trials equivalent to 25 blocks. Each training phase (phases 1 and 2) consisted of 120 trials (10 blocks) where each block included a single presentation of each stimulus in each season (6 stimuli × 2 seasons × 10 blocks). The retest phase (phase 3) consisted of 120 trials (10 blocks) of task A stimuli only (6 stimuli × 2 seasons × 10 blocks). While participants were always probed on the two seasons of a given stimulus in a fixed order (summer then winter), the order of presentation of the different stimuli was randomized within a block. During the retest phase, participants continued to receive feedback for the summer season (always presented first) but not the winter season.

Feedback was presented as a circle indicating the true stimulus location, with Gaussian noise (s.d. 5°) added. Feedback circles were colour-coded green when the response was sufficiently accurate to gain points, and red otherwise. Points were allocated on the basis of the response error (the angular distance between the participant's response and the correct location). Points were calculated as

$$\text{points} = \log\left(\frac{1}{\text{error}^2}\right).$$

Points were capped at a maximum of ten per trial and rounded to the nearest integer (errors ≥30° earned no points). During training (phases 1 and 2), the cumulative score was displayed at the top of the screen. One of the six stimuli in task A was randomly selected as a test stimulus. For the test stimulus, feedback on the winter location was withheld for the entirety of the experiment, requiring participants to infer the correct location using the task rule. After completing the full study, participants were tested on their ability to report the onset of each stimulus. Each stimulus was presented twice in a randomized order, and participants were asked to indicate with a left/right button press whether they had observed each stimulus for the first time near the beginning of the study (corresponding to task A stimuli), or halfway through the study (corresponding to task B stimuli).

## Conditions

Participants were randomly assigned to one of three conditions, defined by the similarity of the rules in tasks A and B. In the Same condition, the rule remained identical across tasks. In the Near condition, the rule changed by 30° (clockwise or counterclockwise) between tasks. In the Far condition, the rule changed by 180° between tasks.

The task A rule was selected randomly and uniformly for each participant. Across both tasks, the angular distance between summer and winter locations was constrained to be greater than 30°. This restriction placed the task A rule in the ranges of 60–150° and 210–300°, and task B rules in the ranges of 30–120° and 240–330°. The summer locations of stimuli within each task were randomized for each participant. Regular spacing between stimuli was enforced by sampling six initial positions at 60° intervals with added Gaussian noise (s.d. 15°), separately for each task.

## Behavioural analyses

We began by analysing participants' accuracy. Response error is defined as the absolute difference between a plant's true location and the participant's response on any trial, in degrees. To compute accuracy, we normalized response error using the maximum possible error (180°):

$$\text{accuracy} = 1 - \left(\frac{\text{error}}{180°}\right).$$

An accuracy of 1 therefore indicates a perfect response, while an accuracy of 0 indicates the maximum possible error. Unless explicitly stated otherwise, we analysed accuracy for the winter probe responses, which invoke the task rule (because winter is a fixed offset from summer and is always preceded by summer).

For our main two hypotheses concerning transfer and interference (preregistered at https://osf.io/92dpm/), we report the results separately for the discovery and replication samples (see Supplementary Fig. 6 for data plotted by sample). To measure transfer, we calculated the change in accuracy for winter response between the final block of task A training (block 10) and the first block of task B training (block 11). Differences in this variable across conditions (Same, Near and Far) were evaluated using a one-way analysis of variance. Post-hoc comparisons were conducted using one-sided t-tests to evaluate the specific predictions that transfer would be lowest in the Far condition, followed by the Near condition and the Same condition.

Interference was measured as the probability of using rule B during retest of task A. We computed participants' rule responses as the offset between their winter season response and the feedback received for the previous summer response. When this metric is computed on the ground-truth winter location, this offset corresponds to the task rule (distance between winter and summer) which is consistent for all stimuli within a task. To quantify interference between the different rules in the Near and Far conditions, we fit a mixture of two von Mises distributions[38] (that is, a circular analogue of the normal distribution) to participants' rule responses, using expectation maximization. We fit a model with predetermined means of rule A ($\theta_A$) and rule B ($\theta_B$) for the two distributions, and two free parameters: a mixing weight ($\pi$) that captures the relative contribution of $\theta_A$ and $\theta_B$, and a single concentration parameter across both distributions ($\kappa$), representing dispersion of the distributions around their respective means. Model fitting was carried out in python using the SciPy package, with custom code adapted from https://framagit.org/fraschelle/mixture-of-von-mises-distributions (ref. 97), based on the method presented in ref. 38. Models were fit separately for each participant, over a range of initial $\pi$ and $\kappa$, with the best-fitting model identified by its log likelihood. Further detail about the the model fitting procedure and model validation is included in Supplementary Section 4, and participant-level model fits with posterior distributions overlaid on responses are shown in Supplementary Figs. 10 and 11.

We used a one-sided *t*-test to evaluate the hypothesis that the contribution of rule B ($\pi$) will be greater in the Near condition than the Far condition during retest of task A, reflecting greater interference from learning task B. As reported in our preregistration, for our interference analysis we excluded participants who failed to learn rule B during task B. Our prediction rests on the assumption that interference when retested on task A is a result of learning task B, so if participants fail to learn task B we did not expect them to show this interference effect at retest of A. We identified these participants by fitting responses during task B (precluding the first block of B where participants have not had the opportunity to learn) to two separate models reflecting use of rule A and rule B (single von Mises distributions centred on $\theta_A$ and $\theta_B$, respectively). Participants who continued to use rule A during task B were excluded in the interference analyses (14%; *n* = 28 excluded out of 202 participants total in the Near and Far groups).

### Individual differences in Near condition

Participants in the Near condition were categorized as either splitters or lumpers based on their susceptibility to interference from task B during the retest of task A. To quantify this, we fit participants' responses during retest to two separate von Mises distributions: one centred on the task A rule ($\theta_A$) and the other on the task B rule ($\theta_B$). Because the models were matched in complexity, the best-fitting model was determined using the minimum negative log likelihood. Participants who returned to using rule A during retest were classified as splitters, while those who updated to using rule B were categorized as lumpers. We continued to exclude those participants who had failed to update to rule B at all during task B (outlined in the previous section). After categorizing participants as splitters or lumpers, we compared a number of behavioural metrics between these two groups.

- Transfer. As previously defined, this is the cost of introducing new task B stimuli. Specifically, this is the change in accuracy for winter responses from the final exposure to task A stimuli to the first exposure to task B stimuli.
- Generalization. Winter accuracy for the test stimulus, for which participants never receive feedback. To infer the correct response, participants had to generalize their knowledge of the task rule to feedback received about the stimulus' summer location. Generalization accuracy is averaged over the second half of task A only, to allow for rule learning.
- Summer accuracy. Average accuracy for summer responses in task A. Unlike winter responses, which could be inferred via the task rule, summer responses relied on participants' memory of the unique stimuli locations. See Supplementary Section 5 for analyses showing that this splitter advantage for summer holds in all three sections of the study (train A, train B, retest A).
- Stimulus onset accuracy. After completing the entire study, participants were asked to report the temporal onset of each stimulus—whether first encountered in the first half of the study (task A) or the second half (task B). In other words, this is their ability to distinguish unique stimuli on the basis of their temporal separation during the study.

We used two-sided *t*-tests to compare the performance of lumpers and splitters for each metric. In addition, because stimulus onset recall accuracy was clustered around the upper boundary, we performed a robustness check by conducting a mixed-effects logistic regression. The model predicted trial-wise binary accuracy during the debrief categorization test from group (lumper/splitter), with a random intercept for participant.

Further analyses explored the nature of interference in both the Near subgroups, and Far condition, focusing on the cognitive mechanisms underlying errors made. In Supplementary Section 4.2, we present a model comparison designed to distinguish whether interference

is better captured by discrete rule swaps or by graded biases. In line with our ANN-inspired predictions, we found that lumpers were best fit by models consistent with graded updating (single distribution), whereas errors in the far group exhibited evidence of swap-like errors (Supplementary Fig. 12). We extend this analysis for the lumper group, by allowing the interference mean to vary freely. The results show strong alignment between the fitted offset and the true rule B direction, further supporting the interpretation that lumpers updated a single internal distribution to reflect the new rule (Supplementary Section 4.3 and Supplementary Fig. 13).

### ANN training procedure

All networks were trained on participant-matched task schedules, meaning each network was paired with a specific human participant and received the exact same sequence of trials—including the precise stimulus order and corresponding target outputs defined by that participant's task rules. Each experimental phase (train task A, train task B and retest task A) consisted of single-batch updates for participant training trials (120 trials per phase, that is, 10 repetitions of 6 unique stimuli probed on each season sequentially). Because neural networks require more training to reach stable performance, we trained each network on its twinned participant schedule repeated 100 times per task phase. New networks were initialized for each participant schedule (*n* = 305 networks total). Network weights were not reset between experimental phases. To match the learning opportunities available to human participants, networks received gradient updates only during trials in which participants received feedback—that is, all trials except the winter trials for the test stimulus in task A and the winter trials during the retest phase of task A.

Neural network simulations were implemented and analysed in Python using the Pytorch, Scikit-learn and Numpy packages. We trained two-layer feed-forward linear networks, mapping discrete one-hot encoded inputs to continuous output coordinates. Layer-wise linear networks provide a tractable framework for analysing internal representations and their role in transfer, generalization and interference[32,98–100]. While our main results are shown in this setting, we replicate key findings in ReLU networks (Supplementary Section 2 and Supplementary Fig. 5) and consider extending to other architectures an important direction for future work.

Inputs to the network were one-hot vectors to match the discrete task stimuli, with six unique inputs per task. Outputs were represented as Cartesian coordinates corresponding to the cosine and sine of the angles for summer and winter, to account for circular wrapping. Thus, the ANNs had 12 input units (representing the 6 stimuli per task) and 4 output units corresponding to cos(summer), sin(summer) and cos(winter), sin(winter). In the results presented, networks had a hidden layer with 50 units, although findings were consistent across different hidden layer sizes (Supplementary Fig. 4). Networks were trained using online stochastic gradient descent (with learning rate $\eta = 0.01$). The model was trained using MSE loss between the true and predicted feature location (Cartesian coordinates) for the probed stimulus season:

$$L = \frac{1}{2}\left[\left(\hat{x}_{s,i} - \sin(\theta_{s,i})\right)^2 + \left(\hat{y}_{s,i} - \cos(\theta_{s,i})\right)^2\right], \tag{1}$$

where $\theta_{s,i}$ is the correct angle for stimulus *s* at the probed season *i* (winter or summer), and $\hat{x}_{s,i}$ and $\hat{y}_{s,i}$ are the outputs corresponding to the network's predicted Cartesian coordinates for stimulus *s* in season *i*. In other words, on each trial, only the summer or winter output pair—depending on which feature was probed—contributed to the loss. The unprobed outputs were not penalized. This output structure avoids forcing the network to resolve competing outputs in a shared space, allowing it instead to learn consistent structure between the outputs. On trials where human participants did not receive feedback (for example, winter responses for the test stimulus or winter responses during retest), no

gradient updates were performed. For interference patterns across a continuum between Near and Far conditions, see Supplementary Fig. 3.

## Analyses of ANN representations

To investigate the task representations supporting transfer in the rich ANNs, we analysed activity at the hidden layer of the network using PCA. First, we assessed the dimensionality of the representations in the different networks. We took each network's activity at the hidden layer when exposed to all inputs, separately after training on task A and task B. We calculated the number of principal components required to explain 99% of the activity variance in each case. This showed that the number of components doubled after training on task B in the Far condition, but remained consistent in the Near and Same condition (Fig. 2d).

To visualize these differences in geometry between conditions (Fig. 2f–h), we plotted the hidden representations of task inputs in the two-dimensional subspace defined by the first two principal components of activity. For illustrative purposes, we trained a network on a randomly selected task A, and then trained it separately on the Same, Near and Far rule version of task B. We then performed PCA ($n = 2$ components) on the network activity when exposed to all inputs (after full training). Projecting all task inputs onto this two-dimensional subspace revealed that, in the Same and Far conditions, the new task B inputs were mapped onto the same subspace, but not in the Far condition. To characterize the representational geometry of the two tasks in the different conditions, we computed the principal angle between task subspaces in all participant-matched networks. The principal angle is a standard measure of the alignment between two subspaces, with smaller angles indicating greater overlap and larger angles indicating orthogonality[40–42]. To compute the principal angle between task subspaces in each network, we extracted the hidden layer activations in response to the set of task A stimuli, and the set of task B stimuli, after training on task B. Using PCA, we identified the two-dimensional subspace for the sets of stimuli belonging to each task, represented by their respective orthonormal bases $\hat{V}_A$ and $\hat{V}_B$. To compute the principal angles between these subspaces, we performed a singular value decomposition on the inner product matrix:

$$\hat{V}_A^T \hat{V}_B^T = P_A \Sigma P_B^T.$$

Here, $\Sigma$ is a diagonal matrix, with diagonal elements corresponding to the cosines of the principal angles between the two subspaces. The first element of $\Sigma$ corresponds to the largest singular value and is the cosine of the first principal angle:

$$\theta_1 = \arccos(\Sigma[1,1]).$$

We display the principal angles ($\theta_1$) between the task A and B subspaces averaged across simulations in each condition in Fig. 2i.

## Rich and lazy regimes in ANNs

We trained ANNs to converge on rich or lazy task solutions by manipulating the scale of initial weights[32–37,43,44,101,102]. All network weights were initialized as random samples from Gaussian distributions with a mean of zero. Following previous work[44], we define the rich regime as networks initialized with small embedding weights ($\sigma = 10^{-3}$) and in the lazy regime large embedding weights ($\sigma = 2$). We present additional information about the performance of networks initialized with variances along this continuum in Supplementary Fig. 16. Because the lazy–rich distinction controls the structure of representations at the hidden layer (formed by the embedding weights), the readout weights must remain flexible to enable learning[44]. During lazy learning, this is what allows the network to extract rules from the high-dimensional representations in the embedding space. We therefore initialized the read-out weights with a fixed rich setting ($\sigma = 10^{-3}$).

## Reporting summary

Further information on research design is available in the Nature Portfolio Reporting Summary linked to this article.

## Data availability

The processed, anonymized human data and simulation data from this study are available via GitHub at https://github.com/eleanorholton/transfer-interference. There are no restrictions on data availability, and all relevant files are provided in CSV format. No data with mandated deposition are included in this study.

## Code availability

The code used for data analysis and modelling in this study is available via GitHub at https://github.com/eleanorholton/transfer-interference.

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

## Acknowledgements

We thank members of the Human Information Processing Lab for useful feedback and discussions of this work. This research was generously funded by a Wellcome Trust Discovery Award (227928/Z/23/Z) to C.S., a Wellcome Studentship (222347/Z/21/Z) awarded to E.H. and funding from the Medical Research Council (MR/N013468/1) for L.B. For the purpose of Open Access, we have applied a CC BY public copyright licence to any author-accepted manuscript version arising from this submission. The funders had no role in the study design, data collection and analysis, or preparation of the paper. Icon and image credits: Figs. 1–4 include third-party icons and images used under their respective licences: Pixabay Content License and OnlineWebFonts CC BY 4.0.

## Author contributions

E.H. contributed to conceptualization, methodology, investigation, formal analysis and writing of the paper. C.S. supervised the project and contributed to conceptualization, methodology and writing. L.B., J.A.T. and J.G. contributed to methodology and paper review and editing.

## Competing interests

The authors declare no competing interests.

## Additional information

**Correspondence and requests for materials** should be addressed to Eleanor Holton.

# Reporting Summary

## Statistics

For all statistical analyses, confirm that the following items are present in the figure legend, table legend, main text, or Methods section.

| n/a | Confirmed | |
|---|---|---|
| ☐ | ☒ | The exact sample size (*n*) for each experimental group/condition, given as a discrete number and unit of measurement |
| ☐ | ☒ | A statement on whether measurements were taken from distinct samples or whether the same sample was measured repeatedly |
| ☐ | ☒ | The statistical test(s) used AND whether they are one- or two-sided *Only common tests should be described solely by name; describe more complex techniques in the Methods section.* |
| ☐ | ☒ | A description of all covariates tested |
| ☐ | ☒ | A description of any assumptions or corrections, such as tests of normality and adjustment for multiple comparisons |
| ☐ | ☒ | A full description of the statistical parameters including central tendency (e.g. means) or other basic estimates (e.g. regression coefficient) AND variation (e.g. standard deviation) or associated estimates of uncertainty (e.g. confidence intervals) |
| ☐ | ☒ | For null hypothesis testing, the test statistic (e.g. *F*, *t*, *r*) with confidence intervals, effect sizes, degrees of freedom and *P* value noted *Give P values as exact values whenever suitable.* |
| ☒ | ☐ | For Bayesian analysis, information on the choice of priors and Markov chain Monte Carlo settings |
| ☒ | ☐ | For hierarchical and complex designs, identification of the appropriate level for tests and full reporting of outcomes |
| ☐ | ☒ | Estimates of effect sizes (e.g. Cohen's *d*, Pearson's *r*), indicating how they were calculated |

*Our web collection on statistics for biologists contains articles on many of the points above.*

## Software and code

Policy information about availability of computer code

| | |
|---|---|
| Data collection | Data were collected using a custom online game developed in JavaScript. Participants were recruited via Prolific. |
| Data analysis | All code is available on GitHub (https://github.com/eleanorholton/transfer-interference). The repository includes custom Python scripts for data analysis and simulations of artificial neural networks (ANNs). All code necessary to reproduce the results of this study is provided. |

For manuscripts utilizing custom algorithms or software that are central to the research but not yet described in published literature, software must be made available to editors and reviewers. We strongly encourage code deposition in a community repository (e.g. GitHub). See the Nature Portfolio guidelines for submitting code & software for further information.

## Data

Policy information about availability of data

All manuscripts must include a data availability statement. This statement should provide the following information, where applicable:
- Accession codes, unique identifiers, or web links for publicly available datasets
- A description of any restrictions on data availability
- For clinical datasets or third party data, please ensure that the statement adheres to our policy

The processed, anonymized human data generated and analyzed in this study are available on GitHub at https://github.com/eleanorholton/transfer-interference. There are no restrictions on data availability, and all relevant files are provided in CSV format. No data with mandated deposition is included in this study.

# Research involving human participants, their data, or biological material

Policy information about studies with human participants or human data. See also policy information about sex, gender (identity/presentation), and sexual orientation and race, ethnicity and racism.

| | |
|---|---|
| Reporting on sex and gender | We asked for participants' self-reported gender at the time of data collection: Out of a total of 305 participants, 157 self-reported as female. We do not include further analysis of gender, as it was not applicable to our research questions. |
| Reporting on race, ethnicity, or other socially relevant groupings | We did not collect data on race or ethnicity, as it was not applicable to our research questions. |
| Population characteristics | See section 'Research sample' in 'Behavioural & social sciences study design' |
| Recruitment | Participants were recruited from Prolific.co, an online platform for participant recruitment. Eligibility criteria included age (18–40 years), English language proficiency, geographical location (US or UK), a minimum approval rate of 90%, and at least 5 previous submissions on Prolific. The use of Prolific may introduce self-selection biases due to familiarity with online studies and limited generalizability to broader populations. However, as our study aims to compare learning mechanisms between humans and artificial neural networks, these biases are unlikely to influence the comparative analysis. |
| Ethics oversight | Ethical approval was obtained from the Oxford Central University Research Ethics Committee (Ref: R50750/RE009). |

Note that full information on the approval of the study protocol must also be provided in the manuscript.

# Field-specific reporting

Please select the one below that is the best fit for your research. If you are not sure, read the appropriate sections before making your selection.

☐ Life sciences  ☒ Behavioural & social sciences  ☐ Ecological, evolutionary & environmental sciences

For a reference copy of the document with all sections, see nature.com/documents/nr-reporting-summary-flat.pdf

# Behavioural & social sciences study design

All studies must disclose on these points even when the disclosure is negative.

| | |
|---|---|
| Study description | Data are quantitative experimental data. The study involves behavioural data collected online from a 45 minute experimental task. |
| Research sample | The research sample consisted of adult participants recruited via Prolific.co, an online platform frequently used for behavioural research. Participants were recruited based on the following inclusion criteria: Aged between 18 and 40 years; Native English speakers; Located in the US or the UK; Holding a minimum approval rate of 90% on Prolific; Having completed at least 5 previous studies on Prolific. A total of 202 participants were recruited for the Discovery study (151 remaining after exclusion) and 215 participants were recruited for the Replication study (154 remaining after exclusion). Of the 305 participants total remaining after exclusion, the mean age was 31.37 years (SD = 8.27) and 157 participants identified as female. |
| Sampling strategy | The sampling strategy was convenience sampling using Prolific.co, an online participant recruitment platform that provides a large and diverse pool of participants. Sample size for the replication study was determined based on a power analysis of the original dataset (discovery sample). The effect sizes detected in the original study suggested that a sample size of 50 participants per condition (same, near, far) would be sufficient to detect the hypothesized effects at a power level of 80%. This sample size also allows us to make direct comparisons with the original study to test the reproducibility of findings. No formal sample size calculation was conducted for the previous study. |
| Data collection | Data were collected using a custom-built online task coded in JavaScript, which was hosted on the lab server. Participants completed the task remotely on their own computers via Prolific.co. The task involved a visual-motor response paradigm where participants moved a dial to indicate their response on a circular display. The experimental setup was fully automated and participants completed the task independently. The experiment was designed to ensure that participants could not receive any feedback or external assistance beyond the visual feedback provided within the task. The researchers were not blind to the experimental conditions, but this was not necessary as no researchers were present during data collection. The study hypotheses were pre-registered and outlined in detail prior to data collection. |
| Timing | Study 1 (discovery sample) was conducted 8th-9th July 2024. Study 2 (replication sample) was conducted 21st October 2025. |
| Data exclusions | Data exclusions were pre-registered and applied according to the following criteria:<br><br>Tool Use Exclusion: Participants who reported using external tools (e.g., pen and paper) to remember correct responses were excluded.<br><br>Accuracy Criterion for Task A: Participants whose accuracy did not significantly exceed chance performance (defined as 90 degrees of error) during the final two blocks of training on Task A were excluded. |

Interference Analysis Exclusion: For the interference analysis, participants who did not adequately learn Task B were excluded. This was assessed by fitting responses during Task B to two von Mises distribution models representing the use of the two rules (A-rule and B-rule). Participants were excluded if the A-rule provided a better fit to their Task B responses than the B-rule.

Incomplete Data: Participants with missing or incomplete data were excluded from the analyses.

These criteria were pre-registered. A total of 51 participants were excluded from the Discovery study (202 recruited, 151 remaining) and 61 participants were excluded from the Replication study (215 recruited, 154 remaining).

**Non-participation**

Participants who began the task could choose to return the study on Prolific.co before completion. Approximately 16% participants returned their submission on Prolific during the study. The decision to return the study was made voluntarily by participants, and no specific reasons for return were provided by Prolific. No participants who completed the task dropped out or declined participation beyond these returns.

**Randomization**

Participants were assigned to one of three experimental conditions (same rule, near rule, far rule) through a process involving three separate recruitment batches on Prolific.co. Each condition was associated with a unique Prolific recruitment link that directed participants to a specific version of the task differing only in the independent variable. The description of the task on the Prolific interface was identical across conditions. Participants self-selected into one of the conditions by responding to the available recruitment link. The recruitment process was run in parallel for all three conditions. Prolific's recruitment interface ensured that participants who participated in one version of the study were not eligible to participate in any of the other conditions.

# Reporting for specific materials, systems and methods

We require information from authors about some types of materials, experimental systems and methods used in many studies. Here, indicate whether each material, system or method listed is relevant to your study. If you are not sure if a list item applies to your research, read the appropriate section before selecting a response.

## Materials & experimental systems

| n/a | Involved in the study |
|-----|----------------------|
| ☒ | Antibodies |
| ☒ | Eukaryotic cell lines |
| ☒ | Palaeontology and archaeology |
| ☒ | Animals and other organisms |
| ☒ | Clinical data |
| ☒ | Dual use research of concern |
| ☒ | Plants |

## Methods

| n/a | Involved in the study |
|-----|----------------------|
| ☒ | ChIP-seq |
| ☒ | Flow cytometry |
| ☒ | MRI-based neuroimaging |

# Plants

**Seed stocks**

NA

**Novel plant genotypes**

NA

**Authentication**

NA

