## [Peer Review File · Nature Human Behaviour]

Humans and neural networks show similar patterns of transfer and interference during continual learning

Corresponding Author: Dr Eleanor Holton

Version 0:

Decision Letter:

22nd April 2025

Dear Ms Holton,

Thank you once again for your manuscript, entitled "Humans and neural networks show similar patterns of transfer and interference during continual learning", and for your patience during the peer review process.

Your Article has now been evaluated by 3 referees. You will see from their comments copied below that, although they find your work of potential interest, they have raised quite substantial concerns. In light of these comments, we cannot accept the manuscript for publication, but would be interested in considering a revised version if you are willing and able to fully address reviewer and editorial concerns.

We hope you will find the referees' comments useful as you decide how to proceed. If you wish to submit a substantially revised manuscript, please bear in mind that we will be reluctant to approach the referees again in the absence of major revisions. We are committed to providing a fair and constructive peer-review process. Do not hesitate to contact us if there are specific requests from the reviewers that you believe are technically impossible or unlikely to yield a meaningful outcome.

In particular, both Reviewers #1 and #3 express concerns about the use of a linear network and recommend comparison with more complex deep nonlinear networks. Additionally, Reviewer #2 requests that you perform additional analyses to rule out alternative mechanisms that could explain the observed behavior. We ask that you take these concerns seriously and address them in full. Also note that Reviewer #1 mentions a "404 error" when trying to access the code. Given the computational nature of your paper, it is important that you make the code and full documentation available to the reviewers for the next round of review.

If you wish to submit a suitably revised manuscript, we would hope to receive it within 4 months. I would be grateful if you could contact us as soon as possible if you foresee difficulties with meeting this target resubmission date.

- Include a "Response to the editors and reviewers" document detailing, point-by-point, how you addressed each editor and referee comment. If no action was taken to address a point, you must provide a compelling argument. When formatting this document, please respond to each reviewer comment individually, including the full text of the reviewer comment verbatim followed by your response to the individual point. This response will be used by the editors to evaluate your revision and sent back to the reviewers along with the revised manuscript.
- Highlight all changes made to your manuscript or provide us with a version that tracks changes.

Link Redacted

Thank you for the opportunity to review your work. Please do not hesitate to contact me if you have any questions or would like to discuss the required revisions further.

Sincerely,

Nature Human Behaviour

Reviewer expertise:

Reviewer #1: Learning, interference, generalization, ANNs

Reviewer #2: Transfer learning, abstract reasoning, computational models

Reviewer #3: Cognitive control, planning, decision-making, ANNs

REVIEWER COMMENTS:

Reviewer #1 (Remarks to the Author):

Summary

This manuscript characterizes the patterns of generalization and interference in both artificial neural networks and human subjects when they learn two tasks in sequence. In the first task, the subject (or network) must learn two radial positions (summer and winter positions) associated with each of six stimuli. For every stimulus, the two positions have equal spacing (eg, 120 degrees clockwise). So, once they learn this rule, they can, in principle, see only one of the two positions (summer is always probed first) and infer the other (winter) position for a new stimulus. In the second task, there are six novel stimuli, each with a novel summer position. However, the rule (i.e., radial distance between summer and winter) can be either the same, nearby to (shifted by 30 degrees), or far (shifted by 180 degrees) from the one in the first task.

The results are interesting and there appears to be good correspondence between learning in the neural networks and human subjects. The demonstration of interference in humans is a nice counterpoint to the often categorical and uncited statement that humans tend to be good at avoiding interference between tasks. The explanation of the individual variability in humans is somewhat unsatisfying, however -- since, as the authors acknowledge, there is no clear mechanism of rich and lazy learning in humans (i.e., the human participants' weights were not just initialized nor do we expect that some humans have their weights initialized to be large, while others have them initialized to be small). Some of the modeling choices are also confusing -- for instance, the choice to use a linear network -- and the description of the networks could be a lot clearer.

I think this is a nice study with interesting results, that would benefit from some additional explanations, discussions, and potentially a few new (but simple) analyses.

Major comments

1. The motivation for using a linear network is not clear. I feel that it should either be justified in a little more detail (the main text mentions that two-layer feed-forward linear networks were used "to study how the representations supporting rule learning emerge through gradient descent," which is possible for nonlinear networks as well), or the main simulations should be run also for a nonlinear network. I would expect that it would give similar results, but any differences between the two would be interesting. The main reason I am aware of for using linear rather than nonlinear networks is the analytical tractability of their learning dynamics. Since no such analysis is performed here, the choice just feels a bit odd -- why use a network at all, instead of a single linear transformation? I suspect the justification could have something to do with the rich and lazy regimes, but this should be spelled out.

2. Do human subjects have knowledge about the overall structure of the experiment (i.e., that they will go through a sequence of task A, B, then A)? More information about how the study is introduced to the subjects would be helpful, since this could be a source of asymmetry between the networks and human participants.

3. The data from both the networks and humans is shown only in a highly processed form. I feel like it would help interpret the data as well as make more clear some of the analyses if the authors showed more, less-processed data. For instance, it would be helpful to see a histogram of behavioral responses around the target, for both summer and winter locations

separately and at the task transitions. Figure 2A, C shows that the difference in accuracy decreases between the same and far conditions, but this measure is a bit hard to wrap your head around. Can we see this effect in the histograms? The same goes for Figure 2B, D: does the distribution of behavioral responses actually look bimodal across the human subjects? The mixture model seems like it would capture this, but one possibility is that there is only a subtle difference and the kappa parameter of the von Mises distributions (which is also fit but never shown) is doing a lot of work (which could happen if the subjects are poor at performing the task)...

4. I feel that more behavioral analysis could yield greater insight into the strategies of especially the human participants. For instance, one possibility could be that the human participants simply guess a winter location that is nearby to the summer location for new stimuli -- and this strategy would end up working reasonably well for the same and near conditions but not the far condition. Is the average first response for the new stimuli actually peaked at the expected location, indicating a more systematic form of inference? More exploration of alternative hypotheses like these would help to convince readers that the human subjects follow the performance of the models, despite the much more mild differences (eg, in Figure 2).

5. The paper focuses on three conditions: Same, near, and far. In the networks, there appears to be a relatively sharp transition (for instance in figure 2A) between the near and far conditions. It would be interesting to use the networks to show that transition in more detail, by interpolating between these two conditions -- is there a sharp transition? Where does it happen? Can this be accounted for by a bifurcation in learning dynamics? I do not view this as necessary for the current paper, but I think it would help to strengthen it, make predictions for future behavioral experiments, and exploit the use of linear networks.

6. The network is given four output units, two for each season. The network then seems to be trained such that, for separate gradient updates, the network is only required to produce the correct output on two of the four output units (eg, only the winter and not the summer outputs) -- is that true? It would be helpful if it were clarified. In any case, this architecture is a significant divergence between the network and the human participants, since the network can always produce its winter and summer outputs simultaneously (even if only one is scored in the loss, activity on the other is not penalized). I understand this simplification is necessary to avoid making the task impossible for a linear network to solve, but it should at least be commented on in the text. A network that is constrained to only produce the output for one of the two seasons at a time would likely have significantly different representations and perhaps different learning, transfer, and interference dynamics.

Minor comments

1. The schematics in the first figure are not really that clear. It would be helpful to really spell out the transfer and interference effects that are expected. Here would also be a helpful place to give a schematic of the neural network and depict the matched training between the two systems.

2. The networks having the learning schedule of the participants is interesting, but I'm not totally sure what it means. Does it mean that each participant had a twin network which was shown the same trials in the same order? Or does it simply mean that the networks were trained with the same overall average structure (eg, 120 trials per phase, with 10 repetitions of each stimulus)? If it's the former, are there any similarities between the networks and their twinned human participant?

3. Related to figure 4 (line numbers would be helpful!), the authors state that lumpers had lower accuracy for summer positions. This is interesting -- and I would wonder if it is also replicated in the ANN. I feel like here some differentiation between the stimuli would be helpful, since this effect could be driven entirely by stimuli from Task A, B, or A re-test depending on how the analysis is designed. Would this effect be consistent with using the same rule for both task A and B stimuli?

4. A comparison with the ANNs for stimulus onset recall would also be interesting (e.g., the rich ANNs likely lump stimuli from the different tasks together while the lazy ANNs do not).

5. In Figure 3 F-H, it would be interesting to see the original representation (after task A training) of the task A stimuli in the same subspace shown for the task A and B stimuli after training on task B.

6. In the intro, the authors write that their findings about lumpers and splitters "imply that a tendency to focus on generalisation of shared features versus individuation of unique features may be a fundamental individual difference in human learning" -- I feel that this is a bit overstated, since, for instance, there's no evidence that someone who is a lumper in this very specific experimental context would also be a lumper in some other context.

Reviewer #1 (Remarks on code availability):

The link gives a 404 error.

Reviewer #2 (Remarks to the Author):

Reviewer's comments

The authors define continual learning as a trade-off between reusing a shared subspace and orthogonalizing the subspace. This trade-off is assessed through transferability and interference, inferred from behavior when participants and networks experience a parametric rule change by progressing through task A, task B, and then returning to task A. They argue that the training regime—rich or lazy—determines whether the network’s representations become more overlapping or orthogonal. Based on this framework, they classify individuals as either "lumpers" or "splitters," and show that their patterns of transfer and interference mirror those of rich and lazy training regimes, respectively.

Their experiment design and results are inspiring, and dividing participants into lumpers and splitters is a creative assessment of individual differences. Thanks to their precise logical flow and concise writing, I broke down their claim into three key components: 1) First, what is the mathematical form of representation that would contribute to or predict the transfer or interference in continual learning? Second, under what network training regime does the subspace become orthogonalized? Third, is there a meaningful behavioral similarity between the proposed network and its biological counterpart?

Upon reviewing the key components, the manuscript appears to present the following line of reasoning: “Networks manage the trade-off between transferability and interference through subspace orthogonalization/reuse, potentially as a result of the rich/lazy training regime. Furthermore, given the observed behavioral similarities between networks and humans, it is proposed that humans may likewise rely on subspace orthogonalization/reuse to address this trade-off (though the underlying representations in humans remain yet undiscovered).” While this argument is intriguing, the central claims may benefit from additional supporting data or further analysis to strengthen their impact and novelty. Specific concerns are outlined below.

Major comments

1. The authors’ claim that subspace orthogonalization prevents interference aligns with ideas previously proposed by other research groups. For example, Xiao Wang and colleagues from Fudan University presented at EMNLP 2023 that subspace representations can support continual learning. Similarly, Philip Torr’s group at the University of Oxford suggested that low-rank orthogonal subspaces provide a mathematical foundation for continual learning (NeurIPS, 2020). These prior works are not currently cited, and it would be both appropriate and constructive to acknowledge and discuss how the present study builds upon or differs from them. Once previous literatures are cited, it may be important to clarify how the current contribution offers additional insight, as the existing literature may otherwise reduce the perceived novelty of the manuscript’s first component.

2. If several groups have already proposed the idea of subspace orthogonalization, then the novelty of the current paper likely hinges on the remaining two aspects. The question of training regime has been a hallmark of the authors’ work over the past few years. Their influential paper (Flesch et al., 2022), which I also enjoyed, demonstrated that a rich training regime leads to low-dimensional and orthogonal manifolds. In that study, the authors noted that rich regimes form orthogonal representations robust to noise across multiple and sequential task presentations—a claim that partially overlaps with the current manuscript. Therefore, their answer to the question, “In what training regime does the subspace become orthogonalized?” seems predictable mainly based on their earlier findings. To make this section more insightful, it would be helpful if the authors could elaborate on what mechanism each training regime may have brought to the current similarity. Indeed, a recent review by Eric Shea-Brown’s group (Farrell et al., 2023) outlines several potential mechanisms by which a rich regime may facilitate orthogonalization, for example, through task-irrelevant information compression or information bottleneck.

3. If the second aspect reiterates the authors’ previous findings from a different perspective, then the novelty of the current paper hinges on the third: the comparison of behavioral patterns between the network and human participants. The observed behavioral similarities are compelling, and prior empirical studies in neuroscience have shown that subspace orthogonalization plays a central role in separating distinct computations, such as movement preparation vs. execution (Kaufman et al., 2014; Elsayed et al., 2016), value computation vs. comparison (Yoo & Hayden, 2020; Johnston et al., 2024), working memory vs. motor planning (Tang et al., 2021), and sequential memory encoding (Xie et al., 2022). More directly relevant, Kim from Nuo Li’s group (2025, Nature) reported the formation of a novel neural repository during continual learning, closely paralleling the result shown in Figure 3E.

However, it is equally plausible that humans employ a variety of mechanisms to manage interference and support transfer. These include machine learning–inspired strategies such as regularization (Serra et al., 2018, PMLR) and teacher model distillation (Duo et al., 2022), as well as biologically inspired mechanisms like memory replay (Rolnick et al., 2019, NeurIPS; Smith et al., 2024, CVPR), neural pruning (Golker et al., 2020, ICLR), synaptic consolidation (Kirkpatrick et al., 2017, PNAS), and dynamic motifs or compositional representations (Driscoll et al., 2024, Nature Neuroscience).

The question, then, is whether the observed behavioral similarity in the current study reflects subspace orthogonalization/reuse or whether it emerges from different underlying mechanisms. Without representational evidence of human support for the authors’ claim, subspace orthogonalization remains just one of several plausible explanations for the observed behavior. Behavioral similarity alone may not be sufficient to conclude that humans rely on orthogonalization/reuse in this task.

4. Relatedly, their previous work examined neural activity in humans and macaque monkeys during task performance and found that early sensory areas represent information in a manner consistent with a lazy regime. In contrast, higher-order areas, such as the posterior parietal cortex, exhibit representations more aligned with a rich regime. This suggests that humans may simultaneously employ both rich and lazy regimes, making direct comparisons between humans and networks challenging. However, comparisons at the level of specific brain regions may be more feasible. Furthermore, the brain

activity of lumpers and splitters while performing this task could be systematically compared.

5. Another way to strengthen the authors' interesting behavioral findings—aside from presenting human neural data—would be to rule out alternative mechanisms that could account for the observed human behavior (e.g., synaptic consolidation or replay-based methods). If the authors' network demonstrates greater similarity to human behavior than networks implementing alternative mechanisms, the argument for subspace orthogonalization/reuse as a contributing factor would be more convincing.

Minor comments

1. Still, Driscoll's paper discusses the emergence and reuse of dynamic motifs—possibly corresponding to subspaces or nonlinear manifolds—which resemble the authors' claims about subspace organization and reuse. If the authors could clarify how their approach differs from Driscoll's, it would provide helpful guidance for a broader readership and enhance the novelty of the manuscript.

2. Mathematically, synaptic consolidation (Kirkpatrick et al., 2017) or neural pruning (Golkar et al., 2020) could act as a synaptic-level mechanism for achieving subspace orthogonalization or reuse, distinct from the rich and lazy training regimes—or perhaps the weight changes observed in the rich regime may yield effects similar to those of synaptic consolidation. Therefore, either ruling out alternative explanations beyond the rich and lazy, or elucidating their relationship to other mechanisms would strengthen the proposed link between training regimes and subspace orthogonalization or reuse.

3. Another concern with relying solely on behavior to validate the results is the possibility of multiple interpretations unless the findings are thoroughly dissected. Unless the authors intend to address major comment #3 directly, they should more comprehensively rule out alternative behavioral explanations through detailed analysis. One such possibility, suggested by Figure 4, is that lumpers generally do not attend to Task A, as indicated by their lower summer accuracy in Figure 4E. As a result, they may exhibit less interference simply because their internal representations were not substantially modified, potentially leading to more overlapping representations. This interpretation is further supported by the fact that lumpers show a lower stimulus recall rate in Figure 4F. Of course, this may not be the correct explanation, but such alternative possibilities should at least be considered and addressed. They should be explicitly tested or rejected with neural data if not ruled out behaviorally.

4. In Figures 2C and 4C, many subjects in both the near and far conditions exhibit a task switch cost above zero. This may suggest that their performance improved more in Task B than in Task A, and the proportion of such subjects does not appear to be small. How can these participants be explained in terms of the training regime or subspace?

5. The classification of splitters and lumpers based on the bimodality of retest accuracy in Figure 4 does not present any significant logical issues. However, since human behavior exists along a spectrum, systematic analysis relating the degree of network richness to that spectrum would offer more insights.

6. In Figure 3I, only two components of the far condition network were used to compare the principal angle. As the far condition has four major PCs, can other components also be compared? The two components can be newly formed, while the remaining two could be reused.

Reviewer #3 (Remarks to the Author):

Overall Assessment

This study investigates continual learning by directly comparing human participants and linear artificial neural networks (ANNs) on the same sequential rule-learning (ABA) task. The findings reveal conserved computational principles across both systems: greater similarity between sequentially learned tasks enhances transfer to the new task, while also increasing retroactive interference with the previously learned task. ANN analyses suggest this trade-off stems from reusing representational subspaces for similar tasks versus forming orthogonal subspaces for dissimilar tasks. A key contribution is the identification of distinct 'lumper' and 'splitter' learning strategies in humans facing intermediate task similarity, which the authors effectively model as analogous to 'rich' and 'lazy' learning regimes in ANNs. Overall, the research provides valuable insights into general computational constraints governing the balance between knowledge integration and segregation during continual learning in humans and machines, and will appeal to a broad audience across fields like machine learning, psychology, and neuroscience. The work represents a significant contribution and, we believe, would be well-suited for publication in *Nature Human Behavior*.

Major Comments:

(1) Exploration of Deep Nonlinear Networks: The study leverages linear ANNs, presumably for analytical tractability and clearer interpretation of representational geometry (as discussed implicitly around Fig 3 and Methods). While insightful, this raises the question of whether the observed principles generalize to more complex, deep nonlinear networks, which are more common in modern AI and potentially closer analogues to biological systems. It would strengthen the work significantly to test whether deep networks exhibit similar representational dynamics (e.g., reuse of subspaces for 'Near' tasks, orthogonalization for 'Far' tasks, rich/lazy regimes) and how their behavior compares to the human data. Alternatively, the

authors could include more justification in the main text for their use of linear ANNs, and further acknowledgments of these limitations. Again, we appreciate the insights that have been gained from the linear network, which does appear to bear similarities to human performance, and do not feel like this omission is 'fatal' for the project. However, we feel that additional control analyses with nonlinear ANNs would further broaden the scope of this work.

(2) Characterizing Interference Errors (Swaps vs. Biases): The analysis of interference relies on a mixture model (Fig 2D, 4A, Methods/SI) that quantifies the probability of using Rule B during Task A re-test. We really appreciate the elegance of this analysis, and . However, it's unclear whether errors reflect 'swaps' (responses centered entirely on Rule B) or graded 'biases' (responses shifted towards Rule B). These possibilities might have different interpretations (e.g., swap models are potentially more consistent with latent causal inference, whereas a bias might emerge from a more traditional [potentially Bayesian] learning process).

(2a) As a general request, the authors could plot more raw(er) response data. For example, they could plot the densities of winter responses during re-test, aligned to the correct response, with the fitted Gaussian Mixture Model (GMM) overlaid. This would serve as a posterior predictive check, visualizing the model fit quality and revealing the nature of participants' errors.

(2b) How do the mixture weight (π) and concentration (κ) parameters contribute distinctly to the behavioral fits and interference patterns? For example, do the concentration parameters (κ) differ between 'lumpers' and 'splitters' — are 'splitters' more precise in their responses?

(2c) Consider comparing the current model to alternatives that might better capture swap vs. bias dynamics. For example, fitting the mean location of the interference component instead of fixing it to Rule B, and/or allowing separate concentration parameters (κ) for the Rule A and Rule B components. These model variants might also help understand the somewhat poorer recovery of mixture weights (i.e., where they may be due to bias), and provide a deeper characterization of the cognitive processes tapped by these experiments (e.g., the formation of new latent contexts).

Minor Comments

(i) Human Use of Orthogonal Representations: The assertion that humans form orthogonal representations for dissimilar ('Far') tasks (Discussion, p11; suggested by ANN results Fig 3H, 3I) is inferred from the ANN analysis rather than direct human neural data. While we find this a highly plausible and theoretically grounded inference, the authors should be a little more circumspect when discussing this point regarding humans, explicitly noting it as a prediction awaiting empirical validation with neural recordings.

(ii) Scope of Continual Learning Mechanisms: The paper focuses on representational overlap/orthogonality as mechanisms underlying the observed continual learning effects. However, human continual learning likely involves additional mechanisms (e.g., hippocampal replay, contextual gating, distinct consolidation processes) not captured by the current linear ANN model. A brief discussion acknowledging these other factors and clarifying how core claims about representational constraints relate/interact with broader consolidation mechanisms would be helpful (related discussion on p11).

(iii) Figure Order (Fig 2 & 3): The presentation jumps from human/ANN behavioral results (Fig 2) to ANN mechanisms (Fig 3) and then back to human results (Fig 2). To improve the narrative flow, consider presenting the ANN model, mechanisms, and results (Fig 3) before the direct human-ANN comparison (Fig 2) to establish the model predictions first.

(iv) Figure 1A-B Diagram: The diagrams illustrating the language learning analogy (Fig 1A-B) are illustrative but potentially redundant given the clear textual explanation. This is a minor stylistic suggestion, not a hard request.

(v) Statistical Analysis of Bounded Outcomes: Accuracy and recall probabilities are bounded between [0, 1]. While the t-tests used for comparisons may be robust in many cases here, they can sometimes be problematic when data clusters near the boundaries (potentially affecting variance estimates). This might be relevant for the stimulus onset recall analysis (Fig 4F), where performance is high. Consider using logistic regression (or beta regression) as a robustness check, especially for this specific analysis.

Reviewer #3 (Remarks on code availability):

Though I only looked through the code briefly, it is very well-documented, and looks like it will make for a good community resource.

Version 1:

Decision Letter:

Our ref: NATHUMBEHAV-25031051A

9th July 2025

Dear Dr Holton,

Thank you for submitting your revised manuscript "Humans and neural networks show similar patterns of transfer and interference during continual learning" (NATHUMBEHAV-25031051A). It has now been seen by the original referees and their comments are below. As you can see, the reviewers find that the paper has improved in revision. We will therefore be happy in principle to publish it in Nature Human Behaviour, pending minor revisions to satisfy the referees' final requests and to comply with our editorial and formatting guidelines.

We are now performing detailed checks on your paper and will send you a checklist detailing our editorial and formatting requirements within two weeks. Please do not upload the final materials and make any revisions until you receive this additional information from us.

Sincerely,

Nature Human Behaviour

Reviewer #1 (Remarks to the Author):

Summary

I thank the authors for their very detailed response to my comments. They have addressed all of my major concerns. I feel that the revised manuscript will make an excellent contribution to the field. I have just a few minor comments.

Minor comments

1. I appreciate the authors' inclusion of histograms showing error distributions in Figure 3 A, F. I'm just a bit unclear on what "mean error" means as the x-axis label. What is the mean of? Maybe participants? When I first saw the new plot, I expected that it would simply be a histogram of all trials pooled across participants (which is what I think is done in Figure 4B), or something along those lines, but this seems not to be the case. Clarifying it would be helpful!

2. On line 220 (tracked changes version), the authors state that "the pattern of switch costs that we observe can only be explained by..." -- this seems too broad, since the authors only compared to a handful of alternatives. I feel this statement would require some kind of more exhaustive search or mathematical proof; I'd suggest moderating it.

3. The y-axes for Figure 4A and B were confusing since the label is the same, but the two plots count different things -- as I understand it, 4A counts participants and 4B counts trials. I would suggest making it more explicit (though the figure legend does resolve my confusion!).

Reviewer #1 (Remarks on code availability):

The code appears well-documented and usable.

Reviewer #2 (Remarks to the Author):

I have read the revised manuscript multiple times, and although my comments in this round are minimal, it has become one of the most extensive and engaging review experiences I've had. I was highly impressed by the depth and clarity of the authors' responses to the previous concerns. The revision is thoughtful, comprehensive, and carefully executed. In particular, Figures S2 and S17 were highly informative and contributed meaningfully to the interpretability of the core claims. I even found that some of the supplementary figures might merit inclusion in the main text, given that supplementary materials often receive less attention. Overall, I sincerely appreciate the authors' attention to detail throughout the revision, and I found the process of reading this updated version both rewarding and enjoyable.

Reviewer #2 (Remarks on code availability):

I did not run the entire code again, but it was well-documented, easy to read, and interpretable.

Reviewer #3 (Remarks to the Author):

We commend the authors on their thoughtful and comprehensive review. Their response answers all of our major questions, and the resulting paper has even better clarity, depth, and breadth than the initial manuscript. We have one more clarifying question, but we are happy to trust the authors to decide how to incorporate it into their manuscript, even without another round of review. We fully endorse this exemplary manuscript.

Question: The posterior predictive checks in 4B, S10, S11 confirm that the authors have captured the unimodality and the distribution means quite well. However, they don't appear to capture the shape of the distribution very well.

I see two plausible reasons for this.

(1) scale-matching: It seems that the authors overlaid 'counts' for histograms and (presumably) the PDF of the VM model. If the authors plotted the data and model in consistent units (e.g., using a gaussian KDE instead a histogram for the data) then the quality of their model fit would be clearer.

(2) model misspecification: It may be that the authors' VM model doesn't capture the variance of the data distribution well. In this case they may prefer to use another parameterization, e.g., a VM-uniform mixture or a circular t-distribution.

If this is all just scale-matching, then re-plotting should be a straightforward fix. However, the data distributions look more peaked than the authors' model. Even though there may be model misspecification, I am happy to defer to the authors given that I think it is unlikely that alternative models will substantially change the authors' core results.

I think you might find that the left tail of the splitter error distribution is better fit by including a uniform component ('guessers'/'forgetters'). This could be asymmetric because uniform density on the right tail of splitters would make someone a lump (though it is surprising that there isn't an obvious uniform density on the right tail of the lump distribution). Perhaps the authors have their own explanation for the left tail of the splitter distribution.

An unfortunate outcome may be that the authors have included more 'guessers' in the splitter group somehow, and that's contributing to why splitters show bigger switch costs, worse generalization performance, and (most plausibly) lower kappa. However, I think it is quite unlikely that 'guessers' could entirely drive these 'lumper advantage' effects. Moreover, (1) accounting for guessers might actually enhance 'splitter advantage' findings, (2) the authors' control analyses that already account for some differences in overall performance (to the extent that 'guessers' merely have lower task engagement), and (3) and the ANNs capture a broad set of behavioral patterns that I think are quite unlikely under this 'guesser' concern.

So, we leave it up to the authors to decide how to handle this. Our hierarchy of recommendations are (1) include posterior predictive checks in Fig 3, (2) find a better way to match the plotting scales between model and data, and (3) decide whether and how to accommodate any remaining model mis-match. We think this should either include additional modelling or mentioning this caveat in the discussion. We want to reiterate again that we think this has limited ability to alter the paper's core conclusions; the existing work is of an extremely high calibre.

Responses to Reviewers

Humans and neural networks show similar patterns of transfer and interference during continual learning

Eleanor Holton, Lukas Braun, Jess Thompson, Jan Grohn and Christopher Summerfield

We are grateful to the three reviewers for their insightful comments, which we hope have helped us improve the manuscript considerably. We are pleased to submit a substantially revised version of the paper. Below, we provide point-by-point responses to each comment. Reviewer comments are shown in *italicized text*. Our responses follow each comment in standard black font. New additions to the manuscript are indicated in **bold** and new text is referenced by page and line number.

Reviewer #1:

This manuscript characterizes the patterns of generalization and interference in both artificial neural networks and human subjects when they learn two tasks in sequence. In the first task, the subject (or network) must learn two radial positions (summer and winter positions) associated with each of six stimuli. For every stimulus, the two positions have equal spacing (eg, 120 degrees clockwise). So, once they learn this rule, they can, in principle, see only one of the two positions (summer is always probed first) and infer the other (winter) position for a new stimulus. In the second task, there are six novel stimuli, each with a novel summer position. However, the rule (i.e., radial distance between summer and winter) can be either the same, nearby to (shifted by 30 degrees), or far (shifted by 180 degrees) from the one in the first task.

The results are interesting and there appears to be good correspondence between learning in the neural networks and human subjects. The demonstration of interference in humans is a nice counterpoint to the often categorical and uncited statement that humans tend to be good at avoiding interference between tasks. The explanation of the individual variability in humans is somewhat unsatisfying, however -- since, as the authors acknowledge, there is no clear mechanism of rich and lazy learning in humans (i.e., the human participants' weights were not just initialized nor do we expect that some humans have their weights initialized to be large, while others have them initialized to be small). Some of the modeling choices are also confusing -- for instance, the choice to use a linear network -- and the description of the networks could be a lot clearer.

I think this is a nice study with interesting results, that would benefit from some additional explanations, discussions, and potentially a few new (but simple) analyses.

Major comments

1. *The motivation for using a linear network is not clear. I feel that it should either be justified in a little more detail (the main text mentions that two-layer feed-forward linear networks were used "to study how the representations supporting rule learning emerge through gradient descent," which is possible for nonlinear networks as well), or the main simulations should be run also for a nonlinear network. I would expect that it would give similar results, but any differences between the two would be interesting. The main reason I am aware of for using linear rather than nonlinear networks is the analytical tractability of their learning dynamics. Since no such analysis is performed here, the choice just feels a bit odd -- why use a network at all, instead of a single linear transformation? I suspect the justification could have something to do with the rich and lazy regimes, but this should be spelled out.*

Thank you for raising this important point. We now include additional discussion in the manuscript clarifying our focus on linear networks (**Lines 133–139, p. 4**), as well as supplementary analyses demonstrating that our key findings hold in a non-linear setting (**Supplementary Methods 1, Fig. S5**). Finally, we expand our discussion to connect with existing literature on orthogonalisation in non-linear networks and emphasise the importance of testing whether the principles observed here extend to other deeper, non-linear architectures (**Lines 335–345, p. 11**). We hope these additions provide a clearer rationale for our modelling choices and address the reviewer's concerns. With regards to the reviewer's specific questions:

1. Linear networks over regression. Single-layer models cannot transfer prior knowledge to new sets of unique stimuli in our framework because, with one-hot inputs, no regression weights would be shared across stimuli. This means that a regression model would not show any generalisation properties. Since our central question concerns how different internal representations support generalisation, transfer and interference, this makes a two-layer linear network the minimal model for probing studying these dynamics. In our framework, transfer and generalisation refer to distinct abilities: *generalisation* is the ability to apply a learned rule to an untrained stimulus within the same task, while *transfer* refers to the ability to carry over prior knowledge from Task A to Task B. We contrast the behaviour of a regression model and rich ANN at generalisation (top panel) and transfer (bottom panel). Rich ANNs can generalise to untrained data, while the linear regression cannot. We also show Transfer for both types of model (bottom panel), defined as the change in accuracy from Task A to Task B. In the linear ANN, this is condition dependent based on the similarity between task rules, while error in the regression model is flat and at chance across all conditions.

2. Extension to reLU networks. We initially conducted our analyses using layerwise linear models, as our task is linearly solvable so these models offer the simplest interpretable tools for the learning dynamics we aim to capture. However, we fully agree with the reviewer that it is an important direction to explore whether our main findings extend to non-linear models. To address this, we conducted additional analyses using a two-layer network with ReLU activations. These analyses confirm that the main findings—namely, reduced transfer and reduced interference in Far versus Near networks—also hold in the non-linear model, supporting the robustness of our conclusions beyond linear networks. We have now included this analysis in our supplementary materials, where we also provide full detail about our methods for the (**Supplementary Methods 1, Fig. S5**).

Figure S5. Patterns of transfer and interference in a two-layer non-linear (ReLU) network. (A) Schematic of the non-linear network architecture, consisting of two fully connected layers with a ReLU activation applied to the hidden layer. (B) Loss curves during training of the ReLU network under the Same, Near, and Far conditions, showing mean-squared error reduction across tasks. Each network is trained sequentially on Task A followed by Task B, and then re-tested on Task A (weights not updated during re-test). (C) Transfer to Task B (change in winter accuracy between the final block of Task A and the first exposure to Task B stimuli). Far networks exhibited worse transfer than Near networks. (D) Interference (probability of applying Rule B when retested on Task A) is reduced in Far compared to Near networks. (E) Principal angles between Task A and Task B subspaces after training. Larger angles indicate greater orthogonality; Far networks exhibited angles close to 90°, consistent with reduced interference. (F-H) Two-dimensional PCA projections of hidden layer activity (pre-ReLU) for networks trained in the Same, Near, and Far conditions, respectively (Task A inputs shown in purple, Task B inputs shown in green). (I) Dimensionality of the hidden layer representation, quantified as the number of principal components required to explain 95% of hidden activity variance before applying the ReLU activation. Dimensionality was on average higher in the ReLU network compared to linear networks, reflecting richer representational complexity. However, consistent with the linear networks, ReLU networks trained in the Far regime increased complexity the most after training on Task B.

2. Do human subjects have knowledge about the overall structure of the experiment (i.e., that they will go through a sequence of task A, B, then A)? More information about how the study is introduced to the subjects would be helpful, since this could be a source of asymmetry between the networks and human participants.

Thanks for raising this key point. The human subjects did not receive any information that suggested there would be a sequence of tasks (Task A followed by Task B). At the transition point between tasks, new stimuli began to appear, but there was no additional explicit indication that a new task had begun. For networks, Task B also corresponded to a different set of inputs

which networks were only exposed to after training on Task A. We made these design choices specifically to minimise the asymmetry between information available to human participants and to the networks, probing the question of how people and networks self-structure their learning without clear boundaries between tasks. To make clarify this, we have made two changes:

1. Within the manuscript, we have added additional information to clarify the information received by the participants both within the main text (**Lines 96–97, p. 4**) and in the methods (**Lines 473–474, p. 13**)
2. To provide complete information about what participants were told, we now include the full set of introductory task instructions, retest instructions, and debrief instructions in the (**Supplementary Text S1**).

3. The data from both the networks and humans is shown only in a highly processed form. I feel like it would help interpret the data as well as make more clear some of the analyses if the authors showed more, less-processed data. For instance, it would be helpful to see a histogram of behavioral responses around the target, for both summer and winter locations separately and at the task transitions. Figure 2A, C shows that the difference in accuracy decreases between the same and far conditions, but this measure is a bit hard to wrap your head around. Can we see this effect in the histograms? The same goes for Figure 2B, D: does the distribution of behavioral responses actually look bimodal across the human subjects? The mixture model seems like it would capture this, but one possibility is that there is only a subtle difference and the kappa parameter of the von Mises distributions (which is also fit but never shown) is doing a lot of work (which could happen if the subjects are poor at performing the task)...

We appreciate the reviewer’s suggestion to include less-processed behavioral data for clearer interpretation and to establish the basis for our model-based analyses. In response, we have added several new visualizations that clarify key aspects of participant behavior across conditions and task phases. The new additions to the manuscript and supplements include:

1. **Figure 3, A-C** – Histograms of Task B error. We include histograms of average participant winter error early in Task B (immediately after the switch) and late in Task B. In the Same condition, the early Task B distribution is tightly clustered around zero, indicating successful transfer of the Task A rule. In the Near condition, the distribution is slightly biased toward the Task A rule, consistent with participants applying the previous learned rule. In the Far condition, errors are broadly distributed, suggesting participants are in general not applying the prior Task B rule but are instead learning from scratch. Note that *late* in training of Task B (final block), all groups show low-error distributions centered around zero, indicating successful learning of the new rule.

2. **Figure 3, F-H** – Histograms of Task A re-test error. We also have added a second set of histograms overlaying the average participant Task A error at re-test with their Task A error during training. In the Near condition, these distributions show a clear shift toward the Task B rule and have peaks at both Rule A and Rule B, illustrating the interference effect and supporting our model-based finding that participants were bimodally distributed (some who responded with Rule A and some with Rule B). In the far condition, re-test error remains centered around zero with some outliers.

3. **Supplementary Figure S14** – Kappa parameter. We have also added a supplementary figure comparing kappa values between ‘lumpers’ and ‘splitters’ in the Near condition. While lumpers show higher kappa values during retest (indicating more precise responses), their accuracy is in fact lower, supporting our interpretation that lumpers have updated to the Task B rule and are applying it consistently but inappropriately during re-test. By contrast, splitters show lower kappa values (more variable responses), but higher accuracy, suggesting a different strategy that better preserves the original mappings even during learning of Task B.

Supplementary Figure S4. Comparing response distributions in lumpers and splitters. **(A–B)** Distribution of all retest response errors pooled across participants for lumpers (left) and splitters (right), with the average posterior of the fitted mixture of von Mises model overlaid (black line). Purple and green notches mark the expected error if applying the Task A or Task B rule, respectively. Lumpers cluster around the Task B rule, while splitters are centered on the Task A rule. The distribution for splitters is broader, indicating lower precision. **(C)** Mixture model weight (π) for splitters versus lumpers. Circles show group means, dots represent individual participants, and error bars indicate s.e.m. **(D)** Same as (C), but for fitted κ (kappa) of the von Mises distribution. Lumpers show higher κ , reflecting greater response precision ($t(78) = 2.83$, $p = .0059$; Mann–Whitney $U = 1098.0$, $p = .0039$). **(E)** Average retest winter accuracy for each group. Despite higher κ values, lumpers show lower accuracy—consistent with the interpretation that they apply the Task B rule consistently but inappropriately during re-test. Splitters, by contrast, show more variable responses (lower κ) but higher accuracy, suggesting a strategy that better preserves the original Task A mapping during Task B learning ($t(78) = -1.80$, $p = .076$; Mann–Whitney $U = 512.0$, $p = .0059$).

4. **Supplementary Figure S8, C–D** – distributions of summer and winter error. As can be seen below, summer locations are effectively learned across all groups, and error decreases across each training phase (notably because participants continue to get feedback about summer responses during A re-test though they stop receiving feedback about winter responses).

5. **Supplementary Figures S10 and S11**– Histograms of individual responses with posterior model fits (in Near and Far condition). We overlaid participant-level histograms of raw responses with posterior fits of the mixture of von Mises models during re-test of Task A (shown for near and far participants). These illustrate that the model accurately captures both the central tendencies and spread of participant behavior via the fitted π and κ parameters. Owing to space constraints, we refer the reviewer to Supplementary Figures S10 and S11

We hope these additions make the behavioral patterns clearer and further validate the modeling results by grounding them in trends that are observable in less processed data.

4. I feel that more behavioral analysis could yield greater insight into the strategies of especially the human participants. For instance, one possibility could be that the human participants simply guess a winter location that is nearby to the summer location for new stimuli -- and this strategy would end up working reasonably well for the same and near conditions but not the far

condition. Is the average first response for the new stimuli actually peaked at the expected location, indicating a more systematic form of inference? More exploration of alternative hypotheses like these would help to convince readers that the human subjects follow the performance of the models, despite the much more mild differences (eg, in Figure 2).

We agree that further behavioral analyses can help clarify the strategies employed by human participants, and we thank the reviewer for this suggestion.

1. **Histograms of responses in Figure 3.** We believe our updated visualizations described in response to Question 3 already speak to this point, showing that participants in the Same condition generalize the Task A rule to new winter stimuli in early Task B, and participants in the Near condition exhibit systematic bias toward the Task A rule (suggesting participants are applying prior task rules in the Same and Near condition). In contrast, participants in the Far condition show highly varied Task B responses early in Task B, consistent with learning a new rule.
2. **Ruling out heuristic strategies (Supplementary Fig. S9).** We thank the reviewer for suggesting an alternative explanation based on participants using the summer location to infer the winter response. We agree this is an important possibility to consider. To evaluate this, we simulated behaviour if participants used this heuristic, as well as if they responded randomly. We examined behaviour from these two simple response criteria:
 - a. Participants respond randomly
 - b. Participants repeat the the summer feedback location for the winter responses

And compared these simulated simple strategies to our hypothesis:

- c. Participants apply the Task A rule during Task B

Crucially, we find that only the use of Task A rule (i.e. c), reproduces the condition-specific switch cost patterns observed in the behavioural data – namely, better transfer in the Same and Near conditions than in the Far condition.

Supplementary Figure S9. Simulated transfer under alternative strategies. Transfer (i.e. switch cost) is defined as the drop in accuracy from the final set of winter responses for Task A stimuli to the first set of winter responses for the Task B stimuli (A-C) Simulated switch costs for three alternative strategies. (A) Random responding – winter responses in Task B are drawn from a uniform distribution. (B) Summer-repeat strategy – winter responses in Task B repeat the preceding summer feedback. This strategy predicts lower accuracy in summer due to control constraints that matched the Task A rules across all conditions, while the rules for Near and Far were also matched in Task B (see Figure S1). (C) Rule-based strategy – participants continue applying the Task A rule to the feedback they receive on summer for the new Task B stimuli. To capture participant-specific variability, we fitted a von Mises distribution to each participant’s winter responses in the second half of Task A (blocks 5–10), using the known Task A rule as the fixed mean and estimating individual κ . This precision was used to simulate variable rule-based responses in Task B. (D) Human behavioural data showing switch costs across task conditions, defined as the drop in accuracy from the final set of winter responses for Task A stimuli to the first set of winter responses for the Task B stimuli. Error bars denote s.e.m. across participants. Only the rule-based strategy (C) reproduces the condition-specific switch cost pattern observed in the human data—namely, greater transfer in the Same and Near conditions than in the Far condition.

3. **Clarification about rule matching across tasks (Supplementary Fig. S1).** To further clarify why the heuristic strategy of using the summer feedback location to generate winter responses cannot account for our results, we include an additional Supplementary Fig. S1 that depicts the distribution of Task A and Task B rules across conditions (Same/Near/Far). Our condition manipulation (Same, Near, Far) was based on the angular *relationship* between Task A and Task B rules, not the spatial proximity of summer and winter locations. Crucially, we matched the distributions of task rules across the Near and Far conditions to avoid confounding rule similarity with the rule itself. As shown in Supplementary Fig.1B-C, if participants had relied on summer feedback for winter, errors should have been highest in the Same condition and comparable in Near and Far. Instead, we observe the opposite pattern, consistent with rule-based generalisation.

Supplementary Figure S1. Matched task rules across conditions. (A) Histograms of Task A rules (i.e., the angular offset between summer and winter locations during Task A) for participants in the Same (yellow), Near (blue), and Far (red) conditions. Task A rules are matched across conditions. (B) Histograms of Task B rules across the same three conditions. Task B rules in the Near and Far conditions were identically distributed and offset by $\pm 30^\circ$ or 180° from their corresponding Task A rules. Task B rules in the Same condition remain within the original Task A range. (C) Mean expected error across conditions if participants had responded to winter trials by simply selecting the location where they had received summer feedback (during Task B). This simple strategy would produce larger errors in the Same condition (due to a wider average angular separation between summer and winter) and similar errors in Near and Far conditions. However, empirical results show the opposite pattern—suggesting that participants rely on learned task rules rather than a spatial proximity heuristic. Error bars indicate standard deviation.

- As a final additional analysis of participant strategy during the retest period, we refer the reviewer to a new model comparison we implemented to distinguish between different types of interference errors at re-test. Specifically, we analyse whether errors reflected gradual biases toward the new rule or trial-wise swaps to the Task B rule (described in response to reviewer 3 question 2, and presented in **Supplementary Figure S12**). We

find that in the far condition, errors represent trial-wise rule swaps (consistent with representing two separate task rules), while in the near condition, errors are more likely to represent a graded bias (consistent with learning a single rule).

5. The paper focuses on three conditions: Same, near, and far. In the networks, there appears to be a relatively sharp transition (for instance in figure 2A) between the near and far conditions. It would be interesting to use the networks to show that transition in more detail, by interpolating between these two conditions -- is there a sharp transition? Where does it happen? Can this be accounted for by a bifurcation in learning dynamics? I do not view this as necessary for the current paper; but I think it would help to strengthen it, make predictions for future behavioral experiments, and exploit the use of linear networks.

Thanks for this suggestion, we have made two additions that address this question:

1. **Supplementary Fig. S3** – analysis of interference across a continuum of rule shifts. We now include a finer-grained sweep of rule shift angles between Task A and Task B to explore the transition between Near and Far conditions in more detail. In the lazy regime, the network maintains high-dimensional, task-agnostic representations, resulting in relatively stable interference across all rule shifts. This insensitivity reflects the fact that the network learns little structure that can be generalised between tasks. In contrast, in rich networks, we observe a sharp nonlinearity: as rule distance increases, interference drops abruptly when tasks are orthogonal (at 180°), where networks switch to encoding tasks in orthogonal subspaces (consistent with the reviewer’s prediction of a bifurcation in learning dynamics).

Supplementary Figure S3. Interference across a continuum of rule shifts between Task A and Task B. To explore the transition between Near (30°) and Far (180°) conditions, we trained neural networks on a sweep of rule shift angles in 15° increments. For each of 10 participant-matched training schedules, networks were trained under three regimes: rich (red), lazy (blue), and rich + modular (green). The y-axis shows re-test interference (quantified as the

probability of applying Rule B during re-test of Task A) as a function of the angular distance between task rules (x-axis). Rich networks exhibit a sharp nonlinearity in interference, consistent with a shift in representational strategy towards orthogonal representations only when the rules themselves are orthogonal. Lazy networks show low, stable interference throughout. The rich + modular network—identical to the rich network but with non-overlapping hidden units for each task—imposes orthogonal representations by design, preventing interference regardless of rule distance.

2. **Supplementary Methods S6, Fig. S17-S18** – analyses in modular, replay and EWC networks. In addition, we also performed additional analyses exploring how interference develops in networks using alternative strategies to mitigate forgetting (i.e. modular, replay and elastic weight consolidation), which could inspire further behavioural investigations. In the plot above, interference behaviour has been added for a modular rich network which is forced to learn the two tasks using separate sets of hidden units.

6. The network is given four output units, two for each season. The network then seems to be trained such that, for separate gradient updates, the network is only required to produce the correct output on two of the four output units (eg, only the winter and not the summer outputs) -- is that true? It would be helpful if it were clarified. In any case, this architecture is a significant divergence between the network and the human participants, since the network can always produce its winter and summer outputs simultaneously (even if only one is scored in the loss, activity on the other is not penalized). I understand this simplification is necessary to avoid making the task impossible for a linear network to solve, but it should at least be commented on in the text. A network that is constrained to only produce the output for one of the two seasons at a time would likely have significantly different representations and perhaps different learning, transfer, and interference dynamics.

Thanks for making this observation. It is correct that the network has four output units (two per season), and during training, only the relevant season-specific outputs (summer or winter) contribute to the loss on a given trial. The non-probed season outputs are not penalized. We have added additional text to the revised manuscript to clarify both the design of this training procedure, and the rationale behind it (**Lines 607–611, pg. 16**). We expand on the rationale below.

This architecture enables the network to learn the transformation between summer and winter locations as a relational mapping in the output space, while keeping the input representation shared. In contrast, forcing the network to use shared outputs for summer and winter would fundamentally alter the problem studied. Tasks that require context-dependent responding with a shared output space resemble XOR problems, where solving the task requires nonlinearities (Flesch et al., 2022; Mante et al., 2013). Such problems have been widely studied in

context-dependent classification tasks (e.g., Roy et al., 2010; Brincat et al., 2018; Takagi et al., 2020), which can be solved by non-linear networks that partition hidden units via anti-correlated inputs gated by context. Adopting that architecture would shift the focus from generalisation of relational transformations to resolving interference between competing responses.

We believe our architecture choice mirrors the structure of the human task: while viewing the stimulus image, participants are always asked to report both locations, first summer, and then winter. This sequential design emphasises learning a consistent mapping between outputs, rather than selecting between competing responses (which might pose a challenge if the order of stimuli and seasons were randomised).

Minor comments

1. The schematics in the first figure are not really that clear. It would be helpful to really spell out the transfer and interference effects that are expected. Here would also be a helpful place to give a schematic of the neural network and depict the matched training between the two systems.

Thanks for this suggestion. To improve clarity, we have revised Figure 1 to address these issues:

1. The new schematic in **Fig.1D** shows the **windows for the expected transfer and interference effects**.
2. The new schematic in Fig.1D also illustrates an **example training regime** for a learner (i.e. spelling out the stimuli ordering). We also depict that each participant schedule is twinned with a network and explain this in the caption.
3. We have also added a **schematic of the neural network in Fig.1E**.

Figure 1. Task Design. (A) The main task consisted of mapping plant stimuli to their locations on a circular dial, across two contexts (summer and winter). Participants always responded with the probed plant's summer location first, and then its winter location, receiving feedback after each response during training. (B) Within Task A, the relationship between each plant's location in summer (white circle) and winter (black circle) corresponded to a fixed angular rule (e.g. 120° clockwise), that was randomised across participants. (C) In Task B, all participants learned to map a new set of stimuli to their respective summer and winter locations. However, the rule defining the relationship between seasons differed across groups of participants. In the Same condition, the seasons for Task B were related by the same rule previously learned in Task A; in the Near condition the rule shifted by 30°; in the Far condition, the rule shifted by 180°. (D) All learners were trained on Task A (120 trials), then Task B (120 trials), and then re-tested on Task A without feedback for winter. Transfer is defined as the change in winter accuracy from the final block (i.e. one full stimuli cycle) of Task A, to the first block of Task B. If participants learn the rule, transfer should be better when the Task B rule is more similar. Interference is defined as the probability of updating to the Task B rule during re-test of Task A. For each participant, we trained a twinned neural network on the same stimuli sequence order and task rules. (E) Networks consisted of feed-forward two-layer ANNs trained to associate sets of unique inputs (one-hot vectors; separate sets for each task) with the cartesian coordinates of the winter and summer locations.

2. The networks having the learning schedule of the participants is interesting, but I'm not totally sure what it means. Does it mean that each participant had a twin network which was shown the same trials in the same order? Or does it simply mean that the networks were trained with the same overall average structure (eg, 120 trials per phase, with 10 repetitions of each stimulus)? If it's the former, are there any similarities between the networks and their twinned human participant?

Thank you for this question - we could have made this clearer in the original manuscript. We have clarified this in the updated manuscript and performed an additional analysis to address the reviewer’s question about network-human correlations.

1. **Clarification of twinned networks (Lines 123–125 p. 4 and Lines 582–584, p. 16).**

Each participant was indeed paired with a “twin” network that received the exact same trial sequence – i.e., the same rules experienced in each task, and the same randomised ordering of stimuli. However, since neural networks require more training to reach stable performance, we trained each network on their twinned participant schedule repeated 100 times per task phase (Task A, Task B, and A re-test). This ensured the network had sufficient opportunity to converge on a task solution before moving to the next training section, while preserving the structure of the human training experience. We also add a schematic to indicate this in the new **Figure 1D** (see response to Minor Comment #1).

2. We performed additional analyses to address your question of whether there are similarities between individual participants and their “twinned” networks. We do not observe any trends in the relationships between participant behaviours of transfer and interference, and the behaviour of their matched network within each condition. This partly reflects the fact that networks trained in the rich regime converge on a narrow range of behaviours, whereas human participants exhibit greater inter-individual variability (related to our later analyses varying network training regime along the rich–lazy continuum).

3. Related to figure 4 (line numbers would be helpful!), the authors state that lumpers had lower accuracy for summer positions. This is interesting -- and I would wonder if it is also replicated in the ANN. I feel like here some differentiation between the stimuli would be helpful, since this effect could be driven entirely by stimuli from Task A, B, or A re-test depending on how the analysis is designed. Would this effect be consistent with using the same rule for both task A and B stimuli?

Thank you for this thoughtful observation. To clarify our summer accuracy analysis and address your point, we have now included additional results across all task phases and confirmed that the effect replicates in the ANN models, with full details provided in Supplementary Methods S5 and Fig. S15.

1. **Human summer accuracy differences across different task sections – Supplementary Methods S5 + Fig. S15A.** Thanks for encouraging us to clarify our analyses of summer accuracy in human participants. To clarify, in the main text analysis, we focused on summer accuracy during Task A and found that splitters outperformed lumpers (two-sample t-test: $t(78) = 3.40$, $p = 0.001$). However, we now confirm that this effect – i.e. higher summer accuracy for splitters than lumpers – is robust across all phases of the task (Task B: $t(78) = 4.11$, $p < 0.001$; A Re-test: $t(78) = 3.95$, $p < 0.001$; all phases combined: $t(78) = 4.44$, $p < 0.001$). We clarify the focus on Task A in the main manuscript (**Lines 241–250, p. 8** and **Lines 293–296, p. 9**) and include the extension of results to other task sections in the supplementary materials.
2. **Replication of summer advantage in lazy ANNs, Fig. 4F, + Fig. S15A.** The reviewer’s intuition is correct that this effect is replicated in the neural networks (although only in initial learning of Task A). Specifically, we observe that lazy networks – which resemble splitters in their behavioural profiles – show significantly higher summer accuracy than rich networks within the matched window of Task A ($t(200) = 3.60$, $p < 0.001$; Fig.S15A). This pattern arises because the lazy networks are initialized with larger weights, leading them to form high-dimensional random projections that approximate input–output mappings without requiring substantial weight change. As a result, they converge more quickly during training (Chizat et al., 2018; Flesch et al., 2022). We now include this matched comparison in the main Figure 4, but refer readers to the supplementary figure that shows development of accuracy over time to see how this changes across learning.

Figure S15. Accuracy across training for splitters and lumpers compared to rich and lazy networks. (A) Summer accuracy for human participants in the Near condition, averaged across blocks and separated by group: lumpers (orange) and splitters (blue). Shaded areas represent ± 1 SEM. Vertical dotted lines indicate transitions between Task A, Task B, and A re-test. **(B)** Same as (A), but for ANNs trained in the lazy (blue) or rich (orange) regime. Here we show ANN performance on the first 50 repetitions of their participant-matched schedule per task phase. While networks were trained for 100 repetitions, only the initial 50 are shown to highlight the learning trajectories prior to convergence, as performance in the final 50 repetitions had already plateaued. **(C)** Winter accuracy for the same human participants shown in (A) **(D)** Winter accuracy for the corresponding ANNs shown in (B). **(E)** Winter accuracy for the test stimulus, where participants receive summer feedback only and must infer the correct winter location. **(F)** Winter accuracy for the test stimulus in corresponding ANNs. No gradient updates are performed for the test stimulus winter trials.

Fig. 4F. Average accuracy for summer responses, which must be remembered for each stimulus separately (in contrast to winter responses, which can be inferred by applying the rule to the summer feedback). This requires participants to discriminate the unique stimuli. ANN performance is shown for the first 120 trials of Task A training, to match the length of human training. For full accuracy trajectories over time, including later stages of training, see Supplementary Fig. S15.

3. Regarding the reviewer’s final question—whether the summer advantage in splitters might be due to applying the same rule to both Task A and Task B stimuli—we note that this interpretation seems not to apply given the task structure. Specifically, summer locations are always probed first, and no stimuli are shared across tasks. As a result, participants cannot use the previous task rule to infer the summer location of a novel Task B stimulus; such a rule can only be applied when the corresponding winter location is presented.

4. A comparison with the ANNs for stimulus onset recall would also be interesting (e.g., the rich ANNs likely lump stimuli from the different tasks together while the lazy ANNs do not).

Thanks for this interesting suggestion, which we have implemented in **Fig.4H** (alongside further explanation in **Lines 297–307, p. 9**). To test whether rich ANNs “lump” stimuli across tasks more than lazy ANNs, we measured the representational overlap between Task A and Task B after learning, by computing the principal angle between their respective PCA subspaces in hidden layer space—smaller angles reflect greater overlap. In rich networks, task subspaces were aligned (mean principal angle = 0.49° , SE = 0.006), whereas in lazy networks, the angle was closer to orthogonal (mean principal angle = 75.10° , SE = 0.59). A two-sample t-test confirmed this difference was highly significant: $t(200) = 125.50$, $p < 0.001$, $d = 17.75$. This analysis relates to the behavioural stimulus onset recall test in participants, where retaining temporal distinctions between task stimuli would depend on maintaining separable task representations.

(H) Representational similarity between Task A and Task B stimuli in ANNs, quantified as the principal angle between their respective hidden layer subspaces after Task B training. Rich networks collapse the representations onto the same subspace, while lazy networks retain greater distinction between representations.

In lines 297–307 (p. 9), we explain this:

Finally, we turn to the result that ‘splitter’ participants were significantly better at remembering the onset of each stimulus — that is, whether a stimulus was first encountered in training on Task A or Task B. This ability to explicitly distinguish when a stimulus was introduced likely depends on maintaining separable representations of the two tasks. To test whether representational compression differed between rich and lazy ANNs, we measured the degree of overlap between Task A and Task B stimuli hidden layer representations after Task B training, using the principal angle. In rich networks, the task subspaces were nearly aligned (mean principal angle = 0.49° , SE = 0.006), whereas the first two principal components in the lazy networks were near orthogonal for the two tasks (mean principal angle = 75.10° , SE = 0.59), with a highly significant difference between groups (Fig.4H, $t(200) = 125.50$, $p < 0.001$, $d = 17.75$). These findings provide a representational-level explanation for the temporal memory differences observed in humans: possibly, participants who compressed information across tasks (i.e. lumpers) suffered when required to recall temporally specific information about task stimuli, just as rich networks collapsed their task representations.

5. In Figure 3 F-H, it would be interesting to see the original representation (after task A training) of the task A stimuli in the same subspace shown for the task A and B stimuli after training on task B.

Thanks, to address this suggestion, we have added the following:

1. **Fig. S2** – We have added a new supplementary figure to visualise the neural representations of Task A and Task B stimuli in both subspaces, enabling clearer insight into how these representations evolve throughout training.
2. **Fig. 2E** – We have also added the post Task A subspace to the main text.

Fig.S7. Evolution of ANN hidden layer subspaces. (A) Task A and Task B stimuli projected into the subspace defined by the first two principal components obtained after training on Task A. As expected, Task A stimuli are well-separated in this subspace immediately after training on Task A. Notably, for networks in the Same and Near conditions, this subspace continues to be used after Task B training. However, the Task B stimuli remain concentrated near the origin, corresponding to the fact they have not yet been trained at the end of Task A. (B) Projection of hidden layer activity onto the same subspace as in (A) after the network was trained in the Far regime. Task B stimuli remain entirely collapsed at the centre even after Task B training. This indicates that learning in the Far condition evolves in a separate, orthogonal subspace, consistent with our principal angle analyses. (C) Projection of hidden layer activity onto the same subspace as in (A) after the network was trained in the Near regime. Task B stimuli are mapped onto the same subspace, reflecting reuse of Task A representations. (D) Same as

(C), but for the Same condition, similarly showing reuse of the original Task A subspace. (E–H) In the Far regime, Task A and Task B stimuli occupy two orthogonal two-dimensional subspaces, with a principal angle of 90° . Accordingly, we visualise hidden activity projected onto each task’s subspace, before and after Task B training. (E) Post-Task A hidden activity projected onto the A stimuli subspace. (F) Post-Task B hidden activity projected onto the A stimuli subspace. (G) Post-Task A hidden activity projected onto the B stimuli subspace. (H) Post-Task B hidden activity projected onto the B stimuli subspace. (I–J) Complementary subspace analysis in the Far regime, fitting the four principal components to hidden layer activity across all inputs after Task B training. Since Task A and Task B occupy orthogonal 2D subspaces, their union forms a 4D subspace. The top four principal components span this space, capturing linear combinations of both task representations rather than aligning strictly with either.

6. In the intro, the authors write that their findings about lumpers and splitters "imply that a tendency to focus on generalisation of shared features versus individuation of unique features may be a fundamental individual difference in human learning" -- I feel that this is a bit overstated, since, for instance, there's no evidence that someone who is a lumper in this very specific experimental context would also be a lumper in some other context.

Thanks for pointing this out – we have revised the sentence to emphasise the need for future validation across contexts:

These findings suggest that a tendency to focus on generalisation of shared features versus individuation of unique features may reflect a meaningful axis of variation in human learning, though further work is needed to determine the stability of these tendencies across contexts.

Reviewer #1 (Remarks on code availability):

The link gives a 404 error.

We apologize for any confusion—the code was initially provided as a separate zip file for anonymous review. We have now made the code publicly available on GitHub for full transparency: <https://github.com/eleanorholton/transfer-interference>.

Reviewer #2

Reviewer’s comments

The authors define continual learning as a trade-off between reusing a shared subspace and orthogonalizing the subspace. This trade-off is assessed through transferability and interference, inferred from behavior when participants and networks experience a parametric rule change by progressing through task A, task B, and then returning to task A. They argue that the training

regime—rich or lazy—determines whether the network’s representations become more overlapping or orthogonal. Based on this framework, they classify individuals as either "lumpers" or "splitters," and show that their patterns of transfer and interference mirror those of rich and lazy training regimes, respectively.

Their experiment design and results are inspiring, and dividing participants into lumpers and splitters is a creative assessment of individual differences. Thanks to their precise logical flow and concise writing, I broke down their claim into three key components: 1) First, what is the mathematical form of representation that would contribute to or predict the transfer or interference in continual learning? Second, under what network training regime does the subspace become orthogonalized? Third, is there a meaningful behavioral similarity between the proposed network and its biological counterpart?

Upon reviewing the key components, the manuscript appears to present the following line of reasoning: “Networks manage the trade-off between transferability and interference through subspace orthogonalization/reuse, potentially as a result of the rich/lazy training regime. Furthermore, given the observed behavioral similarities between networks and humans, it is proposed that humans may likewise rely on subspace orthogonalization/reuse to address this trade-off (though the underlying representations in humans remain yet undiscovered).” While this argument is intriguing, the central claims may benefit from additional supporting data or further analysis to strengthen their impact and novelty. Specific concerns are outlined below.

Major comments

1. The authors’ claim that subspace orthogonalization prevents interference aligns with ideas previously proposed by other research groups. For example, Xiao Wang and colleagues from Fudan University presented at EMNLP 2023 that subspace representations can support continual learning. Similarly, Philip Torr’s group at the University of Oxford suggested that low-rank orthogonal subspaces provide a mathematical foundation for continual learning (NeurIPS, 2020). These prior works are not currently cited, and it would be both appropriate and constructive to acknowledge and discuss how the present study builds upon or differs from them. Once previous literatures are cited, it may be important to clarify how the current contribution offers additional insight, as the existing literature may otherwise reduce the perceived novelty of the manuscript’s first component.

Thanks for highlighting these important contributions. We now cite and discuss relevant prior work on subspace orthogonalisation methods for mitigating catastrophic forgetting, including the studies mentioned by Wang et al. (2023) and Chaudry, Torr et al. (2020), in the revised Discussion. We agree that these methods offer a valuable foundation for understanding how task-specific representations can be separated to prevent interference. Our study complements this literature by showing that orthogonal subspaces can emerge spontaneously even in the

absence of explicit task segmentation or architectural constraints, when task statistics are suitably structured. The revised passage (**Lines 334–345, p. 11**) reads:

Across both humans and ANNs, learning more similar tasks led to greater transfer but at the cost of higher interference. Consistent with previous literature (Lee et al. 2022), we observed that ANNs solved similar tasks by repurposing existing representations, facilitating faster learning but corrupting prior representations. In contrast, learning orthogonal tasks encouraged the formation of separate representations, preventing interference. **This aligns with a growing literature in machine learning that proposes to mitigate catastrophic forgetting by encouraging networks to learn in orthogonal subspaces (Zheng et al. 2019, Chaudry et al. 2020, Duncker et al. 2020, Farajtabar et al. 2020, Wang et al. 2023). These methods differ in how they impose subspace separation: for example, Duncker et al. (2020) proposed a continual learning algorithm that encourages networks to organise dissimilar task dynamics into orthogonal representational subspaces, while others impose hard task boundaries using precomputed orthogonal projection matrices (e.g., Chaudhry et al. 2020) or gradient-penalty methods (e.g. Wang et al. 2023). In our setting, linear networks exposed to orthogonal rules in the ‘Far’ condition naturally developed orthogonal subspaces over the course of learning, without any architectural constraints or additional loss terms, demonstrating that orthogonal representations can emerge spontaneously when the meta-statistics of the task structure support them. Notably, neither humans nor ANNs in our study were given explicit task labels or cues about when the task changed. This design mimics more naturalistic learning environments, where task changes are inferred from environmental structure rather than externally signalled (Gershman & Niv, 2010). An important future direction will be to characterise how and when task segmentation occurs at intermediate levels of similarity. One promising approach is to draw on meta-learning methods that infer task boundaries from shifts in data structure, such as Bayesian frameworks that jointly segment data and learn task models (Milan et al. 2019).**

In addition, in the latter section of the paper we discuss how lazy representations can also mitigate catastrophic forgetting (**Lines 370–376, p. 11**). In this case, the mechanism is not through low-dimensional subspace orthogonalisation for different tasks, but rather by learning the original task using a high-dimensional solution that cannot be repurposed for subsequent tasks, and therefore avoids being corrupted. We further discuss this mechanism in response to your question (3), concerning behavioural patterns compared across humans and neural networks.

2. If several groups have already proposed the idea of subspace orthogonalization, then the novelty of the current paper likely hinges on the remaining two aspects. The question of training

regime has been a hallmark of the authors’ work over the past few years. Their influential paper (Flesch et al., 2022), which I also enjoyed, demonstrated that a rich training regime leads to low-dimensional and orthogonal manifolds. In that study, the authors noted that rich regimes form orthogonal representations robust to noise across multiple and sequential task presentations—a claim that partially overlaps with the current manuscript. Therefore, their answer to the question, “In what training regime does the subspace become orthogonalized?” seems predictable mainly based on their earlier findings. To make this section more insightful, it would be helpful if the authors could elaborate on what mechanism each training regime may have brought to the current similarity. Indeed, a recent review by Eric Shea-Brown’s group (Farrell et al., 2023) outlines several potential mechanisms by which a rich regime may facilitate orthogonalization, for example, through task-irrelevant information compression or information bottleneck.

Thanks for these thoughtful comments regarding the novelty of our findings in relation to previous work from our lab, particularly Flesch et al. (2022). We have added a paragraph introducing the discussion that lay-out the novelty of our findings (**Lines 325–333, p.11**), as well as elaborating in other sections of the discussion. Our current study builds on Flesch et al. in several important ways.

1. **Task similarity.** While Flesch et al. 2022 demonstrated that rich training regimes can lead to low-dimensional, orthogonal manifolds across sequential tasks (when training is interleaved), the different tasks in that study were not systematically varied in structural similarity. As such, the task was not designed to investigate how both transfer and interference depend on the *relationship* between tasks. In contrast, the current study is explicitly designed to probe how varying the similarity between tasks influences both representational structure and behavioural outcomes such as generalisation, transfer, and interference. This allows us to look at how orthogonalization depends on the structural statistics of the tasks themselves, rather than the training regime (interleaved or blocked).
2. **Uncued learning.** In our study, neither humans nor networks are given any explicit cues that signal a change in context or task. Unlike Flesch et al. (2022), which used cued contexts, here learners must perform uncued learning, inferring without external guidance where the task boundaries are. In this setting, we find that networks exposed to orthogonal rules (in the ‘Far’ condition) naturally form orthogonal subspaces as a consequence of the global statistics. By contrast, in the Near condition, where tasks share more structural overlap, networks tend to share representations, resulting in interference. This general trend where learners show more interference for more similar tasks is also demonstrated by human learners. This is striking particularly in light of the prevailing view that humans are largely immune to the kind of catastrophic interference typically observed in neural networks (discussed in **Lines 426–441, p. 13**).

- 3. Individual differences and dimensionality of task solutions.** We extend our exploration of interference beyond both training regime (Flesch et al., 2022) and global task statistics (i.e., task similarity) to examine how the dimensionality of task solutions influence interference. We show that in networks, interference can also be avoided when learning two similar tasks (Near) by adopting high-dimensional task solutions. This reveals a critical trade-off: the compressed, low-dimensional “rich” representations that support generalisation to new stimuli and transfer to new tasks are the very representations that become vulnerable to interference when similar tasks are learned in sequence. These ideas build on the mentioned recent theoretical work by Farrell et al. (2023), which links rich learning regimes to compression mechanisms. In our framework, transfer and generalisation refer to distinct abilities: *generalisation* is the ability to apply a learned rule to an untrained stimulus within the same task, while *transfer* refers to the ability to carry over prior knowledge from Task A to Task B. Compressed representations can support both of these abilities (efficient generalisation and downstream transfer; Fusi et al. 2016, Tishby et al. 2018, Canatar et al. 2021, Bernardi et al. 2020, Johnston et al. 2023), but our findings show they also introduce a potential liability: when compressed representations are reused for new tasks, they are more easily disrupted than high-dimensional task-agnostic solutions.

We believe this in particular, our analysis of how interference depends on both (a) the meta-statistics of task similarity and (b) the dimensionality of the task solutions learned, provides a novel and complementary contribution beyond Flesch et al. 2022. We have updated the Discussion section to clarify these distinctions and to situate our findings more explicitly in relation to prior work, including Farrell et al. 2023 (**Lines 372–376, p. 12**).

3. If the second aspect reiterates the authors’ previous findings from a different perspective, then the novelty of the current paper hinges on the third: the comparison of behavioral patterns between the network and human participants. The observed behavioral similarities are compelling, and prior empirical studies in neuroscience have shown that subspace orthogonalization plays a central role in separating distinct computations, such as movement preparation vs. execution (Kaufman et al., 2014; Elsayed et al., 2016), value computation vs. comparison (Yoo & Hayden, 2020; Johnston et al., 2024), working memory vs. motor planning (Tang et al., 2021), and sequential memory encoding (Xie et al., 2022). More directly relevant, Kim from Nuo Li’s group (2025, Nature) reported the formation of a novel neural repository during continual learning, closely paralleling the result shown in Figure 3E.

However, it is equally plausible that humans employ a variety of mechanisms to manage interference and support transfer. These include machine learning–inspired strategies such as regularization (Serra et al., 2018, PMLR) and teacher model distillation (Duo et al., 2022), as

well as biologically inspired mechanisms like memory replay (Rolnick et al., 2019, NeurIPS; Smith et al., 2024, CVPR), neural pruning (Golkar et al., 2020, ICLR), synaptic consolidation (Kirkpatrick et al., 2017, PNAS), and dynamic motifs or compositional representations (Driscoll et al., 2024, Nature Neuroscience).

The question, then, is whether the observed behavioral similarity in the current study reflects subspace orthogonalization/reuse or whether it emerges from different underlying mechanisms. Without representational evidence of human support for the authors' claim, subspace orthogonalization remains just one of several plausible explanations for the observed behavior. Behavioral similarity alone may not be sufficient to conclude that humans rely on orthogonalization/reuse in this task.

Thank you for these insightful comments. To directly address the reviewer's point, we have added (1) Network simulations to probe these different representational structures in networks with replay, EWC, and modular architectures (**Supplementary Methods S6. + Fig. S17**); (2) A comparison of the behavioural predictions that emerge from different representational structures to the behaviour of our "splitters", to support our claim that "splitters" best resembles high-dimensional task-agnostic solutions observed in networks trained in a lazy regime (**Fig. S18**); (3) An extensive discussion of how these findings relate to potential biological mechanisms, beyond the scope of this paper (**Lines 402–425, p. 12**). We describe these in detail below.

1. **Comparing different representations enabling mitigation of interference (Supplementary Methods S6. + Fig. S17).** We strongly agree with the reviewer's point that different methods of mitigating interference can be supported by different representational structures beyond the subspace orthogonalisation. In particular, our claim that "splitters" (those with reduced interference in the Near group) are behaviourally analogous to networks trained in a "lazy" regime is directly contrasted to other forms of representation that could mitigate interference, including learning the two tasks in two low-dimensional orthogonal subspaces. Specifically, "lazy" solutions are supported by high-dimensional, task-agnostic representations that do not reflect shared structure across stimuli.

We identify four different types of representations that could be formed in our continual learning setting:

- (a) **Single low-rank subspace re-used across tasks.** This is what we witness forming in rich linear networks trained on two similar tasks.

- (b) **Orthogonal low-rank subspaces separated by tasks.** This emerges naturally in networks trained on Far tasks in the rich regime. This can also be implemented in networks trained on similar tasks, via a simple modular network architecture that uses separate hidden units for each task. This approach echoes architecture-based strategies in continual learning that isolate task-specific subnetworks to prevent interference (e.g., Lee et al., 2017; Rusu et al., 2016).
- (c) **Overlapping low-rank subspaces.** We find this representational structure emerges in networks where interference is mitigated through additional mechanisms, specifically (i) experience replay, a memory-based approach that maintains a buffer of previous examples to interleave with new data during learning (Robins, 1995; Lopez-Paz & Ranzato, 2017; Rolnick et al., 2019; Van den Van et al., 2020, Smith et al., 2024), and (ii) synaptic consolidation, a regularization-based strategy in which parameter updates are constrained based on their estimated importance to previous tasks, in this case via Elastic Weight Consolidation (Kirkpatrick et al., 2017; Smith et al., 2023).
- (d) **High-dimensional “lazy” subspaces.** Networks initialised with small weights form high-dimensional task-agnostic representations, that do not generalise to untrained structure, but also result in lower interference since the representations cannot be re-used.

Critically, representational types b, c, and d all lead to mitigation of interference but have different generalisation and transfer properties. These behavioural properties allow us to systematically characterise how our results relate to distinct families of representations in the broader continual learning literature. To study this, we investigated different techniques for mitigating interference – EWC, replay, and modular networks. Full methods describing our implementation can be found in **Supplementary Methods S6**. Below, we compare the representational properties of five models—rich learning, lazy learning, modular, EWC, and replay –highlighting their respective trade-offs in transfer, generalisation, and interference.

Supplementary Figure S17. Representational solutions in networks trained on similar tasks (Near condition). Each panel illustrates how different network regimes or interference-mitigation strategies shape internal representations and behavioural outcomes in a continual learning setting: **(1) Rich networks** (small initial weights, no additional constraints) form a single low-dimensional subspace shared across tasks. This supports strong within-task generalisation and transfer but causes interference due to subspace reuse. **(2) Lazy networks** (large initial weights, no additional constraints) develop high-dimensional, task-agnostic representations. These reduce interference but impair generalisation and transfer. **(3) EWC** mitigates interference by constraining parameter updates based on their importance for Task A, using the Fisher Information Matrix. **(4) Replay** reduces forgetting by interleaving training on Task A memories during Task B learning. Both EWC and Replay preserve shared low-dimensional structure while reducing interference through regularisation or joint optimisation. **(5) Modular**

networks assign non-overlapping hidden units to each task, enforcing orthogonal subspaces. This eliminates interference and preserves task-specific generalisation but limits transfer due to representational separation.

2. **Supplementary Fig. S18 and section ‘Comparing human profiles to model behaviours.’ (p. 23, supplementary materials).** Given this range of behavioural profiles from different network types, we can explicitly ask what patterns our “lumpers” and “splitters” exhibit.

- **Splitters** (i.e. participants in the Near condition who avoid catastrophic forgetting) tend to show poor within-task generalisation and poor transfer, resembling lazy networks but not replay, EWC, or modular networks.
- **Lumpers** (i.e. participants in the Near condition who suffer from forgetting) tend to show strong within-task generalisation and strong transfer, matching the profile of rich networks with no additional mechanisms for mitigating interference.

Supplementary Figure S18. Extension of Figure 4 to networks trained with EWC, replay, and modular architectures. This figure compares the behavioural profiles of human participants (“lumpers” and “splitters”) to

networks trained with three additional interference-mitigation strategies: EWC (Elastic Weight Consolidation) penalises changes to parameters important for previous tasks using the Fisher Information Matrix; Replay interleaves Task A trials from memory during Task B learning to preserve performance; Modular: allocates separate hidden units to each task, enforcing orthogonal subspaces. **(A)** Histogram showing distribution of interference weights used to characterise lumpers and splitters across human participants. **(B)** Interference across groups. Splitters, lazy networks, EWC, replay, and modular networks all show reduced interference. **(C)** Transfer behaviour, measured as the change in accuracy from Task A to Task B. Splitters, lazy networks, and rich + modular networks all show reduced transfer. **(D)** Generalisation behaviour, measured as accuracy on an untrained stimulus in Task A. Splitters and lazy networks show the lowest generalisation. Together, these results show that the behavioural profile of human splitters most closely resembles that of lazy networks and that the behavioural profile of human lumpers most closely resembles that of rich networks.

- 3. Biological mechanisms.** We fully agree that humans could, in principle, achieve reduced interference through a variety of biological mechanisms. However, the focus of our study is not to infer the specific neural mechanisms, but rather to characterize the resulting representational properties that lead to different patterns of behaviour. We clarify this in the discussion (**Lines 351–365, pg. 11**). However, our finding discussed in (1) that human participants in the Near condition most closely resemble lazy networks—exhibiting poor transfer and poor within-task generalisation, but reduced interference—raises the question of why more structured, low-dimensional representations that support generalisation do not emerge for this group. One possibility is that biological mechanisms that support other forms of representation mitigating interference for similar tasks operate on longer timescales than our task allows. Replay-based mechanisms potentially support generalisation only after learning, as suggested by work on offline replay and abstraction (Schuck & Niv, 2019; Liu et al., 2019; Shearer et al., 2024). Synaptic consolidation mechanisms observed biologically (e.g., Cichon & Gan, 2015; Hayashi-Takagi et al., 2015; Yang et al., 2009) could, in theory, mitigate interference by reducing plasticity in synapses important for previously learned tasks, that might lead to shared subspaces supported by computational analogues like elastic weight consolidation (Kirkpatrick et al., 2017). However, these processes may require extended consolidation periods to effectively protect prior knowledge. Finally, orthogonalization-based strategies, as implemented in our modular network, likely depend on the distinctiveness of tasks: while participants in the Far condition naturally separate tasks due to their dissimilarity, such separation may be harder to establish in the Near condition, where the similar task rules may blur boundaries and interfere with rapid reorganization. Thus, the lazy strategy observed in “splitters” may reflect a solution for mitigating interference in cases where neither offline consolidation nor straightforward inference of task boundaries are available. We now add a full discussion of this in the Discussion section (**Lines 402–425, p. 12**).

We hope these new analyses help to address the reviewer's insightful questions by systematically comparing the representational solutions that mitigate interference, clarifying the behavioural analogues of these solutions in human data, and situating our findings within the broader landscape of computational and biological mechanisms for continual learning.

4. Relatedly, their previous work examined neural activity in humans and macaque monkeys during task performance and found that early sensory areas represent information in a manner consistent with a lazy regime. In contrast, higher-order areas, such as the posterior parietal cortex, exhibit representations more aligned with a rich regime. This suggests that humans may simultaneously employ both rich and lazy regimes, making direct comparisons between humans and networks challenging. However, comparisons at the level of specific brain regions may be more feasible. Furthermore, the brain activity of lumpers and splitters while performing this task could be systematically compared.

Thank you, we now highlight in the revised discussion that humans may simultaneously employ both rich and lazy regimes across brain regions (**Lines 383–391, p. 12**):

A second limitation is that, without neural data, we cannot determine how tasks are represented in the brain across the two groups, even if their behavioural patterns suggest underlying differences in strategy. Previous neural recordings have suggested that the brain may simultaneously employ both “rich” and “lazy” representational schemes in different regions: early sensory areas exhibit high-dimensional, task-agnostic codes resembling lazy learning, while higher-order areas such as posterior parietal cortex contain lower dimensional task-specific representations (Freedman & Assad, 2006; Flesch et al., 2022). These findings underscore the challenge of associating human behaviour with a single form of representation, but suggest that comparisons at the level of specific brain regions may also be informative. Future neuroimaging studies could compare the neural representations of “lumpers” and “splitters” in our sequential learning paradigm to investigate differences in both dimensionality and localisation of task representations across the brain.

5. Another way to strengthen the authors’ interesting behavioral findings—aside from presenting human neural data—would be to rule out alternative mechanisms that could account for the observed human behavior (e.g., synaptic consolidation or replay-based methods). If the authors’ network demonstrates greater similarity to human behavior than networks implementing alternative mechanisms, the argument for subspace orthogonalization/reuse as a contributing factor would be more convincing.

We believe this point substantially overlaps with the analyses outlined in comment (3), and we thank the reviewer for emphasizing their importance, which lead us to conduct a set of additional analyses in networks with replay, EWC, and modularity (**Supplementary Methods S6. + Fig. S17**). These analyses support our claim that the solutions formed through lazy initialization—rather than the representational solutions formed by replay or consolidation—best accounts for the observed behavioural patterns in splitters (as opposed to lumpers who experience interference). We also reference these new results in the Results section of the main text (**Lines 317–323, p. 11**).

Minor comments

1. Still, Driscoll’s paper discusses the emergence and reuse of dynamic motifs—possibly corresponding to subspaces or nonlinear manifolds—which resemble the authors’ claims about subspace organization and reuse. If the authors could clarify how their approach differs from Driscoll’s, it would provide helpful guidance for a broader readership and enhance the novelty of the manuscript.

We thank the reviewer for highlighting this important connection. We agree that Driscoll et al. (2024) present compelling evidence for the emergence and reuse of dynamic motifs in recurrent networks trained on multiple tasks. An interesting future direction would be to extend our approach to multi-task settings in humans, to investigate whether similarly reusable representations emerge over longer timescales or with broader task repertoires (**Lines 442–447, p. 13**).

Over extended training or in settings involving many tasks, humans appear capable of decomposing tasks into reusable elements (Ferguson, 1956, Schulz, 2017). Recent work by Driscoll et al. (2024) provides a potential mechanism for this, showing that recurrent neural networks trained on multiple tasks develop dynamic motifs—such as decision boundaries or attractor states—that are reused across tasks. Extending our paradigm to longer sequences of tasks in humans could reveal whether similar reusable structures emerge to support the decomposition of tasks into shared and distinct components, facilitating transfer without interference.

2. Mathematically, synaptic consolidation (Kirkpatrick et al., 2017) or neural pruning (Golkar et al., 2020) could act as a synaptic-level mechanism for achieving subspace orthogonalization or reuse, distinct from the rich and lazy training regimes—or perhaps the weight changes observed in the rich regime may yield effects similar to those of synaptic consolidation. Therefore, either ruling out alternative explanations beyond the rich and lazy, or elucidating their relationship to other mechanisms would strengthen the proposed link between training regimes and subspace orthogonalization or reuse.

Thanks for this suggestion. We fully agree that biological mechanisms such as synaptic consolidation (Kirkpatrick et al., 2017) and neural pruning (Golkar et al., 2019) could support subspace orthogonalization or reuse. Indeed, it is plausible that participants in the *Far* condition achieve separation of task representations via such mechanisms. As noted in our response to Comment 3, in contrast we have argued that “splitters” (those who avoid interference in the *Near* condition) are not forming two low-dimensional orthogonal task representations akin to those seen in modular networks, but instead rely on high-dimensional, task-agnostic solutions (akin to “lazy learning”) that avoid interference without supporting generalisation or transfer.

To fully address the relevance of these biological mechanisms for learners in the *Far* condition who may be using orthogonal low-dimensional subspaces, we now explicitly incorporate a broader discussion of biological mechanisms for orthogonalisation into the revised manuscript (Lines 358–365, p. 11):

A recent relevant study in mice (Kim et al., 2025) demonstrated that individual differences in interference during continual learning were correlated with the degree of orthogonalization in neural representations, underscoring the biological relevance of subspace separation in maintaining memory stability. Such partitioning could, in principle, be implemented biologically via mechanisms like synaptic consolidation (Kirkpatrick et al., 2017) or neural pruning (Golkar et al., 2019), which aim to protect task-relevant parameters by selectively reducing plasticity. While our study remains agnostic about the precise biological substrates, we view these mechanisms as complementary to our representational framework, offering potential routes by which the brain could achieve separation of task representations.

3. Another concern with relying solely on behavior to validate the results is the possibility of multiple interpretations unless the findings are thoroughly dissected. Unless the authors intend to address major comment #3 directly, they should more comprehensively rule out alternative behavioral explanations through detailed analysis. One such possibility, suggested by Figure 4, is that lumpers generally do not attend to Task A, as indicated by their lower summer accuracy in Figure 4E. As a result, they may exhibit less interference simply because their internal representations were not substantially modified, potentially leading to more overlapping representations. This interpretation is further supported by the fact that lumpers show a lower stimulus recall rate in Figure 4F. Of course, this may not be the correct explanation, but such alternative possibilities should at least be considered and addressed. They should be explicitly tested or rejected with neural data if not ruled out behaviorally.

Thanks for pointing out that our characterisation of lumpers and splitters would benefit from considering more alternative possibilities.

1. In **Supplementary Methods S6. + Fig. S17**, we addressed major comment #3 directly by providing a detailed set of analyses comparing human behavior to five ANN variants (lazy, rich, modular, EWC, and replay networks). These analyses, presented in the revised manuscript, clarify why lumpers are best characterized as rich learners and splitters as lazy learners, and justify our focus on these comparisons.
2. We also agree that it is essential to rule out alternative behavioral interpretations—such as attentional differences—as explanations for the observed group differences. To this end, we now include additional behavioral analyses in the supplementary material that track performance for lumpers and splitters over the course of the entire experiment. Specifically, we show the full trajectories of accuracy for summer trials, winter trials, and generalisation trials (test stimuli) across all training and test blocks. These data provide strong evidence against the possibility that lumpers simply attended less during training on Task A:
 1. **Test stimulus accuracy in Task A (Supplementary Fig. 15E)**: Lumpers perform better than splitters on the untrained test stimulus during Task A, suggesting superior generalisation of the rule. This pattern would not be expected if they were inattentive.
 2. **Winter accuracy in Task A (Supplementary Fig. 15C)**: Lumpers and splitters show comparable winter accuracy during initial training on Task A, indicating similar engagement with the generalisable feature. Specifically, accuracy on trained winter stimuli did not differ significantly between groups ($t(78) = -0.98$, $p = 0.33$; $U = 707.0$, $p = 0.38$), arguing against the interpretation that lumpers were less attentive across the board during this phase of learning.
 3. **Response precision in Task A**. Lumpers also show comparable precision in their use of Rule A during training on Task A. To assess this, we fit a von Mises distribution (with mean fixed at the Task A rule) to each participant's winter responses in the second half of Task A (blocks 5–10), specifically modelling the circular distance between winter responses and previous summer feedback. We found no significant difference in precision between lumpers and splitters ($t(32) = 0.117$, $p = 0.907$; $U = 638.000$, $p = 0.922$), providing more evidence that lumpers were not less attentive.

Figure S15. Accuracy across training for splitters and lumpers compared to rich and lazy networks.

(A) Summer accuracy for human participants in the Near condition, averaged across blocks and separated by group: lumpers (orange) and splitters (blue). Shaded areas represent ± 1 SEM. Vertical dotted lines indicate transitions between Task A, Task B, and A re-test. (B) Same as (A), but for ANNs trained in the lazy (blue) or rich (orange) regime. Here we show ANN performance on the first 50 repetitions of their participant-matched schedule per task phase. While networks were trained for 100 repetitions, only the initial 50 are shown to highlight the learning trajectories prior to convergence, as performance in the final 50 repetitions had already plateaued. (C) Winter accuracy for the same human participants shown in (A) (D) Winter accuracy for the corresponding ANNs shown in (B). (E) Winter accuracy for the test stimulus, where participants receive summer feedback only and must infer the correct winter location. (F) Winter accuracy for the test stimulus in corresponding ANNs. No gradient updates are performed for the test stimulus winter trials.

Together, these results strongly suggest that lumpers are not simply less attentive to Task A, but rather adopt a different strategy—one that prioritizes shared structure across stimuli that supports better generalisation and transfer. In contrast, splitters appear to rely on more stimulus-specific representations, enhancing summer accuracy, but reducing generalisation. We have added these analyses as supplementary figures listed above.

4. In Figures 2C and 4C, many subjects in both the near and far conditions exhibit a task switch cost above zero. This may suggest that their performance improved more in Task B than in Task A, and the proportion of such subjects does not appear to be small. How can these participants be explained in terms of the training regime or subspace?

Thanks for pointing this out. First, human participants are inherently noisier than neural networks, and greater variability in responding will naturally lead to a subset of cases where performance appears better in Task B than in Task A. However, as the reviewer suggests, more interestingly, the prevalence of such effects should not be uniform across conditions if participants are learning the task structure, in which case the similarity between Task A and Task B should systematically influence the likelihood of positive switch costs.

To address this, we added **Supplementary Figure S9**. Here, we simulate the expected switch costs under three plausible strategies that participants could use when encountering the novel Task B stimuli:

1. Random responding
2. Using the observed summer feedback location as the winter response
3. Applying the Task A rule to the summer feedback location to infer winter (i.e. learning the task *structure*)

These simulated outcomes are shown in panels A, B, and C of Figure S9 respectively (the human data are shown in D for comparison). To capture strategy (3), we fitted a von Mises distribution to each participant's winter responses during Task A, using Rule A as the fixed mean of the distribution and estimating each participant's individual kappa value in order to quantify the empirical precision (or noise) in their responses. This approach allows us to simulate behaviour that reflects each participant's actual level of variability, rather than assuming a uniform noise profile across the group. We then used these kappas to simulate the Transfer effect for participants under the assumption they continued applying the Task A rule during Task B.

Supplementary Figure S9. Simulated transfer under alternative strategies. Transfer (i.e. switch cost) is defined as the drop in accuracy from the final set of winter responses for Task A stimuli to the first set of winter responses for the Task B stimuli (A-C) Simulated switch costs for three alternative strategies. (A) Random responding – winter responses in Task B are drawn from a uniform distribution. (B) Summer-repeat strategy – winter responses in Task B repeat the preceding summer feedback. This strategy predicts lower accuracy in summer due to control constraints that matched the Task A rules across all conditions, while the rules for Near and Far were also matched in Task B (see Figure S1). (C) Rule-based strategy – participants continue applying the Task A rule to the feedback they receive on summer for the new Task B stimuli. To capture participant-specific variability, we fitted a von Mises distribution to each participant’s winter responses in the second half of Task A (blocks 5–10), using the known Task A rule as the fixed mean and estimating individual κ . This precision was used to simulate variable rule-based responses in Task B. (D) Human behavioural data showing switch costs across task conditions, defined as the drop in accuracy from the final set of winter responses for Task A stimuli to the first set of winter responses for the Task B stimuli. Error bars denote s.e.m. across participants. Only the rule-based strategy (C) reproduces the condition-specific switch cost pattern observed in the human data—namely, greater transfer in the Same and Near conditions than in the Far condition. *Note that (E-G) are not included in the Supplementary Figure, but have been added here to illustrate the expected accuracy increase highlighted by the reviewer.*

The results (panels E–H) show the percentage of simulated participants with switch costs above 0 under each strategy. As shown in panel G, even under strategy (3), approximately 50% of participants in the same condition exhibit positive switch costs due to noisy responding (reflecting empirical levels of response variability). Notably, only strategy (3)—continued use of the Task A rule—reproduces the condition-specific pattern observed in the experimental data: a

higher proportion of positive switch costs in the Near and Same conditions than in the Far condition. This makes intuitive sense: when the rules are similar or identical, reapplying the Task A rule to infer the winter response (from summer feedback) is beneficial. This aligns with the predictions of our model, suggesting that participants reuse prior structure when tasks remain similar, just as the networks do.

5. The classification of splitters and lumpers based on the bimodality of retest accuracy in Figure 4 does not present any significant logical issues. However, since human behavior exists along a spectrum, systematic analysis relating the degree of network richness to that spectrum would offer more insights.

We agree that analysing behaviour on a continuum between lumper and splitters could yield valuable insights. Our reason for splitting the data is because in the Near condition, participants appear to cluster into two distinct subgroups of low and high interference. To formally test this, we performed a Hartigan's Dip Test which indicated that the distribution of mixing weights in the Near condition deviated significantly from unimodality (dip = 0.145, $p < .001$), consistent with a categorical split between participants applying Rule A and those applying Rule B. In contrast, the Far condition did not deviate from unimodality (dip = 0.037, $p = .45$), suggesting more uniform reversion to the previous rule across participants.

However, we strongly agree that capturing the full spectrum of learning dynamics is an important future direction, and this connects closely to recent theoretical work by Dominé et al. (2025), who analytically derive the continuum between lazy and rich learning regimes in deep linear networks by varying the key initialization parameter (λ). Future work could explicitly model this by fitting individual differences in richness (e.g., via parameterised λ in neural simulations), capturing a more continuous transition between lumper and splitter strategies. We thank the reviewer for the suggestion, and now include this interesting future direction in the discussion (Lines 395–401 p.12):

In addition, it would be interesting to explore parametric analyses of individual differences using a continuous measure of network richness. Recent theoretical work (Dominé et al., 2025) provides a principled framework for understanding the variation between rich and lazy learning along a continuous axis. Their analytical characterisation of the transition between lazy and rich learning regimes in deep linear networks (driven by the relative scale of initialization) offers a promising basis for predicting behavioural patterns along this continuum, potentially yielding a more nuanced account of human variability in learning strategies beyond the bimodal approach taken in the current study.

6. In Figure 3I, only two components of the far condition network were used to compare the principal angle. As the far condition has four major PCs, can other components also be compared? The two components can be newly formed, while the remaining two could be reused.

Thanks for this suggestion – we have added a visualisation of the geometry of these different components in **Supplementary Fig. S2, E-J** (pasted below). As the reviewer pointed out, after Task B training in the Far condition, the network develops four significant principal components (PCs), reflecting the union of two orthogonal 2D task-specific subspaces. The reviewer’s intuition is correct – that the first components from Task A continue to be used for Task A, while two completely new components are formed for Task B, such that each task retains a distinct, non-overlapping representational basis in the Far regime.

Fig.S7. Evolution of ANN hidden layer subspaces (A) Task A and Task B stimuli projected into the subspace defined by the first two principal components obtained after training on Task A. As expected, Task A stimuli are well-separated in this subspace immediately after training on Task A. Notably, for networks in the Same and Near conditions, this subspace continues to be used after Task B training. However, the Task B stimuli remain concentrated near the origin, corresponding to the fact they have not yet been trained at the end of Task A. (B) Projection of hidden layer activity onto the same subspace as in (A) after the network was trained in the Far regime. Task B stimuli remain entirely collapsed at the centre

even after Task B training. This indicates that learning in the Far condition evolves in a separate, orthogonal subspace, consistent with our principal angle analyses. **(C)** Projection of hidden layer activity onto the same subspace as in **(A)** after the network was trained in the Near regime. Task B stimuli are mapped onto the same subspace, reflecting reuse of Task A representations. **(D)** Same as **(C)**, but for the Same condition, similarly showing reuse of the original Task A subspace. **(E–H)** In the Far regime, Task A and Task B stimuli occupy two orthogonal two-dimensional subspaces, with a principal angle of 90° . Accordingly, we visualise hidden activity projected onto each task's subspace, before and after Task B training. **(E)** Post-Task A hidden activity projected onto the A stimuli subspace. **(F)** Post-Task B hidden activity projected onto the A stimuli subspace. **(G)** Post-Task A hidden activity projected onto the B stimuli subspace. **(H)** Post-Task B hidden activity projected onto the B stimuli subspace. **(I–J)** Complementary subspace analysis in the Far regime, fitting the four principal components to hidden layer activity across all inputs after Task B training. Since Task A and Task B occupy orthogonal 2D subspaces, their union forms a 4D subspace. The top four principal components span this space, capturing linear combinations of both task representations rather than aligning strictly with either 2D subspace.

Reviewer #3 (Remarks to the Author):

Overall Assessment

This study investigates continual learning by directly comparing human participants and linear artificial neural networks (ANNs) on the same sequential rule-learning (ABA) task. The findings reveal conserved computational principles across both systems: greater similarity between sequentially learned tasks enhances transfer to the new task, while also increasing retroactive interference with the previously learned task. ANN analyses suggest this trade-off stems from reusing representational subspaces for similar tasks versus forming orthogonal subspaces for dissimilar tasks. A key contribution is the identification of distinct 'lumper' and 'splitter' learning strategies in humans facing intermediate task similarity, which the authors effectively model as analogous to 'rich' and 'lazy' learning regimes in ANNs. Overall, the research provides valuable insights into general computational constraints governing the balance between knowledge integration and segregation during continual learning in humans and machines, and will appeal to a broad audience across fields like machine learning, psychology, and neuroscience. The work represents a significant contribution and, we believe, would be well-suited for publication in Nature Human Behavior.

Major Comments:

(1) Exploration of Deep Nonlinear Networks: The study leverages linear ANNs, presumably for analytical tractability and clearer interpretation of representational geometry (as discussed implicitly around Fig 3 and Methods). While insightful, this raises the question of whether the observed principles generalize to more complex, deep nonlinear networks, which are more common in modern AI and potentially closer analogues to biological systems. It would

strengthen the work significantly to test whether deep networks exhibit similar representational dynamics (e.g., reuse of subspaces for 'Near' tasks, orthogonalization for 'Far' tasks, rich/lazy regimes) and how their behavior compares to the human data. Alternatively, the authors could include more justification in the main text for their use of linear ANNs, and further acknowledgments of these limitations. Again, we appreciate the insights that have been gained from the linear network, which does appear to bear similarities to human performance, and do not feel like this omission is 'fatal' for the project. However, we feel that additional control analyses with nonlinear ANNs would further broaden the scope of this work.

Thanks for these comments about the potential for extending these findings to deep non-linear networks. To address this, we now include additional discussion in the manuscript clarifying our focus on linear networks (**Lines 133–139, p. 4**), as well as supplementary analyses demonstrating that our key findings hold in a non-linear setting i.e. networks with ReLU (**Supplementary Methods 1, Fig. S5**). Finally, we expand our discussion to connect with existing literature on orthogonalisation in non-linear networks and emphasise the importance of testing whether the principles observed here extend to other deeper, non-linear architectures (**Lines 335–345, p. 11**). We hope these additions provide a clearer rationale for our modelling choices and address the reviewer's concerns.

1. Extension to non-linear ReLU networks. We initially conducted our analyses using layerwise linear models, as our task is linearly solvable so these models offer the simplest interpretable tools for the learning dynamics we aim to capture. However, we fully agree with the reviewer that it is an important direction to explore whether our main findings extend to non-linear models. To address this, we conducted additional analyses using a two-layer network with ReLU activations. These analyses confirm that the main findings—namely, reduced transfer and reduced interference in Far versus Near networks—also hold in the non-linear model, supporting the robustness of our conclusions beyond linear networks. We have now included this analysis in our supplementary materials, where we also provide full detail about our methods for the (**Supplementary Methods 1, Fig. S5**).

Figure S5. Patterns of transfer and interference in a two-layer non-linear (ReLU) network. (A) Schematic of the non-linear network architecture, consisting of two fully connected layers with a ReLU activation applied to the hidden layer. (B) Loss curves during training of the ReLU network under the Same, Near, and Far conditions, showing mean-squared error reduction across tasks. Each network is trained sequentially on Task A followed by Task B, and then re-tested on Task A (weights not updated during re-test). (C) Transfer to Task B (change in winter accuracy between the final block of Task A and the first exposure to Task B stimuli). Far networks exhibited worse transfer than Near networks. (D) Interference (probability of applying Rule B when retested on Task A) is reduced in Far compared to Near networks. (E) Principal angles between Task A and Task B subspaces after training. Larger angles indicate greater orthogonality; Far networks exhibited angles close to 90°, consistent with reduced interference. (F-H) Two-dimensional PCA projections of hidden layer activity (pre-ReLU) for networks trained in the Same, Near, and Far conditions, respectively (Task A inputs shown in purple, Task B inputs shown in green). (I) Dimensionality of the hidden layer representation, quantified as the number of principal components required to explain 95% of hidden activity variance before applying the ReLU activation. Dimensionality was on average higher in the ReLU network compared to linear networks, reflecting richer representational complexity. However, consistent with the linear networks, ReLU networks trained in the Far regime increased complexity the most after training on Task B.

2. **Clarification of choice of linear networks and acknowledgement of limitation.** We now include additional discussion in the manuscript clarifying our focus on linear networks (Lines 133–139, p. 4):

“We chose two-layer linear networks as the simplest architecture capable of learning transferable shared structure in this task. In contrast, single-layer regression models trained on unique (one-hot) inputs cannot share weights across stimuli and are therefore incapable of transfer. Since the task is linearly solvable, linear networks are the most parsimonious choice for studying the representational dynamics supporting transfer and interference. However, we also confirm in

supplementary analyses (Supplementary Methods S1, Fig. S5) that the key behavioural effects also hold in ReLU networks, supporting the robustness of our findings beyond linear networks.”

We also acknowledge that extension to more complex networks would be a valuable future direction (**Lines 596–598 p. 16**), and in the discussion, pasted below (**Lines 337–345, p. 11**):

“Consistent with previous literature (Lee et al., 2022), we observed that ANNs solved similar tasks by repurposing existing representations, facilitating faster learning but corrupting prior representations. In contrast, learning orthogonal tasks encouraged the formation of separate representations, preventing interference. **This aligns with a growing literature in machine learning showing that catastrophic forgetting can be mitigated by encouraging networks to learn in orthogonal subspaces in more complex networks (Zeng et al., 2019; Chaudhry et al., 2020; Duncker et al., 2020; Farajtabar et al., 2020; Wang et al., 2023). Machine learning methods differ in how they impose subspace separation: for example, Duncker et al. (2020) proposed a continual learning algorithm that encourages networks to organise dissimilar task dynamics into orthogonal representational subspaces, while others impose hard task boundaries using precomputed orthogonal projection matrices (e.g., Chaudhry et al., 2020) or gradient-penalty methods (Wang et al., 2023). While our results are grounded in simple linear networks, an important direction for future work is to investigate whether the principles we observe also hold in deeper, nonlinear networks used in modern AI systems.**”

(2) Characterizing Interference Errors (Swaps vs. Biases): The analysis of interference relies on a mixture model (Fig 2D, 4A, Methods/SI) that quantifies the probability of using Rule B during Task A re-test. We really appreciate the elegance of this analysis, and . However, it's unclear whether errors reflect 'swaps' (responses centered entirely on Rule B) or graded 'biases' (responses shifted towards Rule B). These possibilities might have different interpretations (e.g., swap models are potentially more consistent with latent causal inference, whereas a bias might emerge from a more traditional [potentially Bayesian] learning process).

Thank you for raising this insightful point. We fully agree that distinguishing between swap errors and biases offers a useful lens for interpreting participant strategies, and appreciate the link made to the latent cause literature. We have implemented this analysis as a new section **Supplementary Methods S3**. (‘Characterizing Interference Errors: Swaps vs. Biases’), with the results shown in **Figure S12** and described below.

To formally test this, we implemented a model comparison designed to disambiguate these possibilities. Each participant’s distribution of responses at re-test (on Task A stimuli) was fit using five models:

1. Uniform: control model (no free parameters)
2. Rule A: responses drawn from a single distribution centered on the original rule (fitted κ)
3. Rule B: responses drawn from a single distribution centred on the new Task B rule. This is consistent with fully updating to the Task B rule (fitted κ)
4. Interpolated: responses drawn from a weighted average of Rule A and Rule B means. This model would capture participants who are updating a single distribution, but may not have fully updated to the Task B rule (1 interpolation weight, 1 κ)
5. Mixture: a mixture of von Mises distributions centered on Rule A and Rule B. This model would capture rule swap errors within a participant (1 mixture weight, 1 κ)

We compared these models using Bayesian model selection for group studies (Stephan et al., 2009; Rigoux et al., 2014).

If humans rely on similar representational structures, we would expect their error patterns to mirror this geometry: in the near condition, errors (among lumpers) should arise from updating a single hypothesis about a shared subspace (i.e., a gradual shift or updating the full distribution); whereas in the far condition, errors should reflect misclassification—mistaking which representational space (i.e., which task) a given stimulus belongs to, consistent with swap-like behaviour.

In the Far condition, the mixture model had the highest protected exceedance probability, consistent with the presence of swap-like errors. This is consistent with the geometry of representations observed in our artificial neural networks: training in the Far condition leads to orthogonal representations for the two tasks, so if participants make errors, we would expect errors to reflect swaps rather than an interpolation between the task rules. In the Far condition, a subset of participants was best fit by the Rule A model, reflecting participants who make minimal swap errors but are able to return to the rule. A negligible proportion of participants are best fit by Rule B (i.e. full updating). Overall, this is consistent with Far participants generally inferring separate latent causes for the tasks – reflected in the fact that their errors reflect ‘swaps’ between the tasks rather than a graded bias.

In Near (lumpers), the best-fitting model was Rule B, suggesting these participants updated a single distribution to the Task B rule. In Near (splitters), the best-fitting model was Rule A, indicating resistance to interference. Notably, in both Near subgroups, a proportion of participants were better explained by the interpolation model, consistent with graded updating of

a single model or ‘latent cause’. Crucially, we found minimal evidence that participants in the Near group made swap errors—the mixture model was rarely preferred. This is also consistent with the representational geometry seen in our networks. In the Same and Near conditions, networks encode the tasks using the same representational subspace (i.e. shared principal components), such that both tasks are embedded within a common latent space. This would predict fewer errors ‘swap-like’ errors, because the networks do not represent the tasks separately. Overall, this is consistent with Near participants generally inferring a single latent cause for the tasks. We thank the reviewer for suggesting this interesting analysis, which we describe in full detail in **Supplementary Methods S3**.

Figure S12. Model comparison results distinguishing between swap-like and bias-like interference errors. Each participant’s responses at re-test on Task A stimuli were fit using five candidate models: a uniform model (control), a Rule A model (original rule), a Rule B model (fully updated to the new rule), an interpolated rule model (shifted mean), and a mixture model (trial-level swaps between rules). Bars indicate the expected model frequency within each group (Far, Near (lumpers), Near (splitters)), as estimated by Bayesian model selection for group studies (Stephan et al., 2009; Rigoux et al., 2014), implemented via the *bms* package (Huang, 2024). Error bars reflect the standard deviation of the Dirichlet distribution over model frequencies. Horizontal lines and annotations indicate the winning model in each group, with the associated protected exceedance probability.

(2a) As a general request, the authors could plot more raw(er) response data. For example, they could plot the densities of winter responses during re-test, aligned to the correct response, with

the fitted Gaussian Mixture Model (GMM) overlaid. This would serve as a posterior predictive check, visualizing the model fit quality and revealing the nature of participants' errors.

Thanks, we fully agree this is an important addition.

1. **Aggregated histograms (Figure 3 A-C and F-H)** – In Figure 3, we now include histograms of mean error in early B and in re-test of A that illustrate the transfer and interference effects.
2. **Individual participant plots and posterior fits – Supplementary Figures S10 (Far) and S11 (Near)**, we now include plots for individual participants of the winter re-test response densities aligned to the correct (Task A) rule, overlaid with the posterior fits from the von Mises mixture model. These serve as posterior predictive checks and allow visual inspection of how well the model captures participant behavior. Participants were selected to cover a range of kappa and pi values, to check model fits. These visualizations show good correspondence between observed data and the model, and help clarify the nature of participants' interference errors across conditions.

Supplementary Figure S10 (Far Participants).

Supplementary Figure S11 (Near Participants).

- We also include histograms of raw response errors for both winter and summer seasons in **Supplementary Figure S8**.

(2b) How do the mixture weight (π) and concentration (κ) parameters contribute distinctly to the behavioral fits and interference patterns? For example, do the concentration parameters (κ) differ between 'lumpers' and 'splitters' — are 'splitters' more precise in their responses?

Thanks for this question regarding the distinct contributions of the mixture weight (π) and concentration (κ) parameters to behavioral fits and interference patterns, which we address in **Supplementary Figure S8**. To address this, we added a new supplementary figure comparing κ values between 'lumpers' and 'splitters' in the Near condition. **Supplementary Figure S8A-B** show average re-test responses aligned to the correct Task A rule, overlaid with fitted von Mises distributions. Visually, it is clear that splitters exhibit broader response distributions, indicating lower precision (lower κ) compared to lumpers. This difference is confirmed statistically in **Supplementary Figure S8D** where lumpers show significantly higher κ values than splitters

($t(78) = 2.83$, $p = 0.006$; Mann–Whitney $U = 1098.0$, $p = 0.004$). However, as shown in **Supplementary Fig.8E**, splitters nonetheless achieve higher accuracy at re-test than lumpers ($t(78) = -1.80$, $p = 0.076$ (n.s.); Mann–Whitney $U = 512.0$, $p = 0.006$).

This dissociation supports our interpretation that lumpers have updated to the Task B rule and apply it consistently (although inappropriately) during re-test, resulting in precise but systematically biased responses. In contrast, splitters adopt a different strategy that preserves the original Task A mappings better with more variable responses. This would be consistent with using more memorisation, rather than applying a shared rule.

Supplementary Figure 8. Comparing response distributions in lumpers and splitters. **(A–B)** Distribution of all retest response errors pooled across participants for lumpers (left) and splitters (right), with the average posterior of the fitted mixture of von Mises model overlaid (black line). Purple and green notches mark the expected error if applying the Task A or Task B rule, respectively. Lumpers cluster around the Task B rule, while splitters are centered on the Task A rule. The distribution for splitters is broader, indicating lower precision. **(C)** Mixture model weight (π) for splitters versus lumpers. Circles show group means, dots represent individual participants, and error bars indicate s.e.m. **(D)** Same as (C), but for fitted κ (kappa) of the von Mises distribution. Lumpers show higher κ , reflecting greater response precision ($t(78) = 2.83$, $p = .0059$; Mann–Whitney $U = 1098.0$, $p = .0039$). **(E)** Average retest winter accuracy for each group. Despite higher κ values, lumpers show lower accuracy—consistent with the interpretation that they apply the Task B rule consistently but inappropriately during re-test. Splitters, by contrast, show

more variable responses (lower κ) but higher accuracy, suggesting a strategy that better preserves the original Task A mapping during Task B learning ($t(78) = -1.80$, $p = .076$; Mann–Whitney $U = 512.0$, $p = .0059$).

(2c) Consider comparing the current model to alternatives that might better capture swap vs. bias dynamics. For example, fitting the mean location of the interference component instead of fixing it to Rule B, and/or allowing separate concentration parameters (κ) for the Rule A and Rule B components. These model variants might also help understand the somewhat poorer recovery of mixture weights (i.e., where they may be due to bias), and provide a deeper characterization of the cognitive processes tapped by these experiments (e.g., the formation of new latent contexts).

Thanks for these additional ideas for further analysis. Below we describe the analyses we have added that address this.

1. **Supplementary Methods 3, Fig. S12 – Bias vs. swap.** We believe the model comparison analysis described in our response to (2) provides the clearest operationalisation of swap versus bias dynamics. In that framework, swap errors are captured by a mixture model, where responses are generated from a weighted combination of Rule A and Rule B distributions. In contrast, biases are captured by the interpolation model, where responses are centered at a weighted average of the two rules. This approach allows us to distinguish between these two error types and compare them directly using Bayesian model selection.
2. **Supplementary Methods 4, Fig. S13 – Fitting the model offset.** We appreciated the elegance of the reviewer’s suggestion of fitting the mean location of the interference component instead of fixing it to Rule B – specifically for validating interference in our ‘lumpers’ (Near group participants who experience interference). We tested whether the interference component of lumpers’ response distribution—rather than being fixed at Rule B—might be better captured by allowing its mean direction to be freely estimated.

Specifically, we modelled responses ($\theta_1, \theta_2, \dots, \theta_n$) as samples from a von Mises distribution centered on a shifted version of the Rule A mean:

$$\theta_i \sim \text{von Mises}(\mu = \mu_A + \Delta, \kappa)$$

Where μ_A is the known mean direction of the original Rule A, Δ is a free offset parameter, capturing the shift of responses away from Rule A, and κ is the concentration parameter.

Below we plot the fitted Δ (**Fig. S13**), colour coded by the true Rule B offset from Rule A. The results showed a strong correspondence between the fitted offset and the true offset direction, consistent with Near group lumpers fully updating to Rule B, rather than a bias toward an intermediate direction. This strengthens our interpretation that these participants updated that representation to reflect the more recently learned rule.

Figure S13. Fitted offset of responses from the original Task A rule for participants in the Near (lumpers) group. Each bar represents a participant’s fitted Δ value from a von Mises model allowing a free mean shift from Rule A. Bars are colour-coded according to the true angular offset between Rule A and Rule B (either -30° or $+30^\circ$).

We’ve added the methods and results of this additional analysis to **Supplementary Methods 4, and Fig. S13** and we thank the reviewer for prompting this useful extension.

Minor Comments

(i) Human Use of Orthogonal Representations: The assertion that humans form orthogonal representations for dissimilar ('Far') tasks (Discussion, p11; suggested by ANN results Fig 3H, 3I) is inferred from the ANN analysis rather than direct human neural data. While we find this a highly plausible and theoretically grounded inference, the authors should be a little more circumspect when discussing this point regarding humans, explicitly noting it as a prediction awaiting empirical validation with neural recordings.

We fully agree we need to be more careful in clarifying the interpretation of our findings. We have revised the manuscript to avoid making strong claims about the neural representations underlying human behaviour. For example, in the discussion (**Lines 346–350, p. 11**) we now write:

In our setting, linear networks exposed to orthogonal rules in the ‘Far’ condition naturally developed orthogonal subspaces over the course of learning, without any architectural

constraints or additional loss terms. This demonstrates that orthogonal representations can emerge spontaneously when the meta-statistics of the task structure support them. While we observe similar behavioural patterns of reduced transfer alongside lower interference in humans, we emphasise that the emergence of orthogonal representations in the human brain remains a theoretical prediction, pending empirical confirmation via neural recordings.

(ii) Scope of Continual Learning Mechanisms: The paper focuses on representational overlap/orthogonality as mechanisms underlying the observed continual learning effects. However, human continual learning likely involves additional mechanisms (e.g., hippocampal replay, contextual gating, distinct consolidation processes) not captured by the current linear ANN model. A brief discussion acknowledging these other factors and clarifying how core claims about representational constraints relate/interact with broader consolidation mechanisms would be helpful (related discussion on p11).

Thanks for highlighting these other mechanisms involved in mitigating interference during continual learning. We agree they are important to address.

1. We have now added an extensive supplementary section implementing replay- and consolidation-based mechanisms in ANNs (see **Supplementary Methods 6 and Fig. S17 and S18**; and response to Reviewer 2, Comment 3), and compared their behavioural profiles to our human data. These analyses support our core claims about the representational structures that predict our human behavioural patterns. Specifically, we find that “splitters” best resemble high-dimensional “lazy” networks in their pattern of transfer, generalisation and interference behaviours. We contrast that with the behavioural profile of networks trained through other alternative mitigation techniques such as replay or consolidation.
2. We have also expanded the discussion to acknowledge that humans likely deploy additional mechanisms beyond those captured in our linear models, and a discussion of how these relate to our findings (**Lines 402–431, p. 12**):

While we have argued that humans and ANNs face similar computational trade-offs during continual learning, humans are likely to deploy additional mechanisms to balance these challenges. In particular, the medial temporal lobe (MTL) supports the rapid acquisition of new tasks before integration into cortical knowledge systems over longer timescales (McClelland et al., 1995; O’Reilly et al., 2014; Schapiro et al., 2017), a process that our linear networks cannot replicate. **Biological systems may also mitigate interference through mechanisms such as replay-based consolidation (Wilson & McNaughton, 1994; McClelland et al., 1995; Diekelmann & Born, 2010), and synaptic consolidation (Cichon & Gan, 2015; Hayashi-Takagi et al., 2015; Yang et al., 2009), both of which have inspired continual**

learning approaches in artificial learning systems (Masse et al., 2018; Shin et al., 2017; Hadsell et al., 2020; Lesort et al., 2020, Kirkpatrick et al., 2017, Golkar et al., 2019).

Possibly, such mechanisms may operate on timescales beyond those captured by our task. Replay has been shown to support ongoing generalisation (Schuck & Niv, 2019; Liu et al., 2019; Shearer et al., 2024), with interesting implications for how knowledge continues to be structured after learning. This parallels recent studies in rodents showing that generalisation abilities can continue to develop even after task performance has plateaued (Kumar et al. 2024). Synaptic consolidation mechanisms observed in biological systems (Cichon & Gan, 2015; Hayashi-Takagi et al., 2015; Yang et al., 2009) may also require extended consolidation periods to prevent interference. These temporal considerations may help explain why participants in the Near condition who avoid interference (“splitters”) succeed through strategies that come at the cost of transfer and generalisation. Specifically, while participants in the Far condition naturally separate task representations due to their dissimilarity, such separation may be harder to establish in the Near condition, where the similarity of rules blurs boundaries. The strategy observed in “splitters” may reflect a viable solution for mitigating interference in a setting where neither offline consolidation on longer timescales, nor straightforward inference of task boundaries, are available to learners.

(iii) Figure Order (Fig 2 & 3): The presentation jumps from human/ANN behavioral results (Fig 2) to ANN mechanisms (Fig 3) and then back to human results (Fig 2). To improve the narrative flow, consider presenting the ANN model, mechanisms, and results (Fig 3) before the direct human-ANN comparison (Fig 2) to establish the model predictions first.

Thanks for this suggestion – we have revised the manuscript to invert the order of Figure 2 and Figure 3, as the reviewer suggests.

(iv) Figure 1A-B Diagram: The diagrams illustrating the language learning analogy (Fig 1A-B) are illustrative but potentially redundant given the clear textual explanation. This is a minor stylistic suggestion, not a hard request.

We appreciate this observation and agree that the analogy in Figure 1A-B is redundant given the textual explanation, and have removed it in the revised manuscript.

(v) Statistical Analysis of Bounded Outcomes: Accuracy and recall probabilities are bounded between $[0, 1]$. While the t -tests used for comparisons may be robust in many cases here, they can sometimes be problematic when data clusters near the boundaries (potentially affecting variance estimates). This might be relevant for the stimulus onset recall analysis (Fig 4F), where

performance is high. Consider using logistic regression (or beta regression) as a robustness check, especially for this specific analysis.

Following this suggestion, we performed a robustness check for the stimulus onset recall analysis using logistic regression. We ran a mixed-effects logistic regression predicting trial-wise correct categorisations during the debrief phase, with group (lumper vs. splitter) as a fixed effect and a random intercept for participant. The model confirmed a significant group difference ($\beta = 1.35$, $SE = 0.38$, $z = 3.57$, $p < .001$), indicating lower accuracy for lumpers compared to splitters. We have included this additional robustness check in the main results section (**Lines 262–264, p. 8**).

Reviewer #3 (Remarks on code availability):

Though I only looked through the code briefly, it is very well-documented, and looks like it will make for a good community resource.

Reviewer Responses (Round 2)

Humans and neural networks show similar patterns of transfer and interference during continual learning

Eleanor Holton, Lukas Braun, Jess Thompson, Jan Grohn and Christopher Summerfield

We thank the reviewers for their final round of feedback and have incorporated the suggestions from Reviewers 1 and 3 as detailed below.

Reviewer #1:

Summary

I thank the authors for their very detailed response to my comments. They have addressed all of my major concerns. I feel that the revised manuscript will make an excellent contribution to the field. I have just a few minor comments.

Minor comments

1. I appreciate the authors' inclusion of histograms showing error distributions in Figure 3 A, F. I'm just a bit unclear on what "mean error" means as the x-axis label. What is the mean of? Maybe participants? When I first saw the new plot, I expected that it would simply be a histogram of all trials pooled across participants (which is what I think is done in Figure 4B), or something along those lines, but this seems not to be the case. Clarifying it would be helpful!

We re-labelled the axes of **Figure 3 A, B, C, F, G, H** as 'Mean Participant Error' to clarify that the mean was taken across participants. We also updated the figure legend to explicitly state: "histograms of the mean winter error across participants".

2. On line 220 (tracked changes version), the authors state that "the pattern of switch costs that we observe can only be explained by..." -- this seems too broad, since the authors only compared to a handful of alternatives. I feel this statement would require some kind of more exhaustive search or mathematical proof; I'd suggest moderating it.

To avoid making too strong a claim, we changed this sentence to: "Importantly, the pattern of switch costs that we observe is better explained by participants transferring their previous rule to the new Task B stimuli, rather than alternative behavioural strategies such as responding randomly or repeating their summer location feedback."

3. The y-axes for Figure 4A and B were confusing since the label is the same, but the two plots count different things -- as I understand it, 4A counts participants and 4B counts trials. I would suggest making it more explicit (though the figure legend does resolve my confusion!).

We implemented the reviewer's suggestion and changed the y-axes labels of **4A** to 'Count (Near Group Participants)' and of **4B** to 'Count (Near Group Trials)'.

Reviewer #1 (Remarks on code availability):

The code appears well-documented and usable.

Reviewer #2:

I have read the revised manuscript multiple times, and although my comments in this round are minimal, it has become one of the most extensive and engaging review experiences I've had. I was highly impressed by the depth and clarity of the authors' responses to the previous concerns. The revision is thoughtful, comprehensive, and carefully executed. In particular, Figures S2 and S17 were highly informative and contributed meaningfully to the interpretability of the core claims. I even found that some of the supplementary figures might merit inclusion in the main text, given that supplementary materials often receive less attention. Overall, I sincerely appreciate the authors' attention to detail throughout the revision, and I found the process of reading this updated version both rewarding and enjoyable.

Thanks for these kind words, we really appreciated the depth of thought you put into these reviews and enjoyed thinking about how to implement your insights!

Reviewer #2 (Remarks on code availability):

I did not run the entire code again, but it was well-documented, easy to read, and interpretable.

Reviewer #3:

We commend the authors on their thoughtful and comprehensive review. Their response answers all of our major questions, and the resulting paper has even better clarity, depth, and breadth than the initial manuscript. We have one more clarifying question, but we are happy to trust the authors to decide how to incorporate it into their manuscript, even without another round of review. We fully endorse this exemplary manuscript.

Question: The posterior predictive checks in 4B, S10, S11 confirm that the authors have captured the unimodality and the distribution means quite well. However, they don't appear to capture the shape of the distribution very well.

I see two plausible reasons for this.

(1) scale-matching: It seems that the authors overlaid 'counts' for histograms and (presumably) the PDF of the VM model. If the authors plotted the data and model in consistent units (e.g., using a gaussian KDE instead a histogram for the data) then the quality of their model fit would be clearer.

(2) model misspecification: It may be that the authors' VM model doesn't capture the variance of the data distribution well. In this case they may prefer to use another parameterization, e.g., a VM-uniform mixture or a circular t-distribution.

If this is all just scale-matching, then re-plotting should be a straightforward fix. However,

the data distributions look more peaked than the authors' model. Even though there may be model misspecification, I am happy to defer to the authors given that I think it is unlikely that alternative models will substantially change the authors' core results.

I think you might find that the left tail of the splitter error distribution is better fit by including a uniform component ('guessers'/'forgetters'). This could be asymmetric because uniform density on the right tail of splitters would make someone a lumper (though it is surprising that there isn't an obvious uniform density on the right tail of the lumper distribution). Perhaps the authors have their own explanation for the left tail of the splitter distribution.

An unfortunate outcome may be that the authors have included more 'guessers' in the splitter group somehow, and that's contributing to why splitters show bigger switch costs, worse generalization performance, and (most plausibly) lower kappa. However, I think it is quite unlikely that 'guessers' could entirely drive these 'lumper advantage' effects. Moreover, (1) accounting for guessers might actually enhance 'splitter advantage' findings, (2) the authors' control analyses that already account for some differences in overall performance (to the extent that 'guessers' merely have lower task engagement), and (3) and the ANNs capture a broad set of behavioral patterns that I think are quite unlikely under this 'guesser' concern.

So, we leave it up to the authors to decide how to handle this. Our hierarchy of recommendations are (1) include posterior predictive checks in Fig 3, (2) find a better way to match the plotting scales between model and data, and (3) decide whether and how to accommodate any remaining model mis-match. We think this should either include additional modelling or mentioning this caveat in the discussion. We want to reiterate again that we think this has limited ability to alter the paper's core conclusions; the existing work is of an extremely high calibre.

Thanks for this feedback and for highlighting the scale-matching and potential model-misspecification concerns. We have addressed these points as follows:

1. **Scale-matching.** We are grateful for the reviewer highlighting the difference in scale in the posterior plots (Fig. 4B and Supplementary Figs. S10–S11). Upon review, we realized that the posterior was plotted over participants, whereas the histograms were calculated over all data points. We corrected this by plotting the posterior over individual data points in Fig. 4B and in the aggregated data plots of S10–S11. Additionally, as suggested, we included KDE plots over the posterior in S10. These changes improve the apparent fit of the model to the data. There remains a left-hand tail in the splitter distribution, which we investigated further as described below.

2. **Model-misspecification / random responders.** Following the reviewer's suggestion, we conducted a control analysis to identify participants whose responses appeared random. We implemented a leave-one-out cross-validation (LOO-CV) procedure comparing each participant's re-test responses in the Near condition to three candidate models: a von Mises distribution centred on Rule A, a von Mises distribution centred on Rule B, and a uniform distribution. Participants were flagged as random responders if the uniform model had a higher LOO-CV log-likelihood than both von Mises models. We then repeated all Splitter/Lumper analyses excluding these participants.

Importantly, all main findings remained unchanged after this exclusion. In the Near condition, we initially identified 42 Splitters and 38 Lumpers. After excluding participants classified as random responders, 40 Splitters and 35 Lumpers remained. Lumpers continued to show superior transfer ($t(73) = -3.81$, $p < 0.001$, $d = -0.89$) and generalisation accuracy ($t(73) = -3.02$, $p = 0.003$, $d = -0.71$), whereas splitters maintained higher summer accuracy ($t(73) = 2.99$, $p = 0.004$, $d = 0.69$). At re-test, splitters again showed lower response precision than lumpers (κ ; $t(73) = -3.00$, $p = 0.004$, $d = -0.68$), but higher accuracy ($t(73) = 2.16$, $p = 0.034$, $d = 0.51$). Finally, lumpers remained worse at end-of-study temporal categorisation of stimuli ($t(73) = 3.56$, $p = 0.001$, $d = 0.81$). This control analysis is now reported in the new section **Supplementary Methods 7**.

3. **Acknowledging other models.** We also added a note in the discussion acknowledging that alternative model parameterizations could potentially provide a closer fit to the splitter/lumper distributions. Nevertheless, as the reviewer notes, these adjustments are unlikely to alter the core conclusions of the manuscript:

...it remains unclear how far the distinction between splitters and lumpers generalises beyond our task setting. Our classification was based on behavioural patterns of interference, and although this allowed us to predict a rich range of independent behavioural metrics, future work could strengthen the basis of this distinction by introducing a novel task to assess whether the classification generalises beyond the current setting. **Similarly, alternative model parameterizations could potentially provide a closer fit to the splitter and lumper response distributions, offering complementary perspectives on the behavioural distinctions observed in our task.**

We thank the reviewer for their thoughtful engagement with this project in both the first and second rounds of reviews – we really enjoyed thinking about the implications of their suggestions, and implementing the new analyses.